



# Hydraulic fracturing in thick shale basins: problems in identifying faults in the Bowland and Weald Basins, UK

**David K. Smythe**[1, *]

[1]{College of Science and Engineering, University of Glasgow, Scotland}

[*]{now at: La Fontenille, 1, rue du Couchant, 11120 Ventenac en Minervois, France}

Correspondence to: David Smythe (david.smythe@glasgow.ac.uk)

**Abstract.** North American shale basins differ from their European counterparts in that the latter are one
to two orders of magnitude smaller in area, but correspondingly thicker, and are cut or bounded by
normal faults penetrating from the shale to the surface. There is thus an inherent risk of groundwater
resource contamination *via* these faults during or after unconventional resource appraisal and
development. US shale exploration experience cannot simply be transferred to the UK. The Bowland
Basin, with 1900 m of Lower Carboniferous shale, is in the vanguard of UK shale gas development. A
vertical appraisal well to test the shale by hydraulic fracturing (fracking), the first such in the UK,
triggered earthquakes. Re-interpretation of the 3D seismic reflection data, and independently the well
casing deformation data, both show that the well was drilled through the earthquake fault, and did not
avoid it, as concluded by the exploration operator. Faulting in this thick shale is evidently difficult to
recognise. The Weald Basin is a shallower Upper Jurassic unconventional oil play with stratigraphic
similarities to the Bakken play of the Williston Basin, USA. Two Weald licensees have drilled, or have
applied to drill, horizontal appraisal wells based on inadequate 2D seismic reflection data coverage. I
show, using the data from the one horizontal well drilled to date, that one operator failed identify two
small but significant through-going normal faults. The other operator portrayed a seismic line as an
example of fault-free structure, but faulting had been smeared out by reprocessing. The case histories
presented show that: (1) UK shale exploration to date is characterised by a low degree of technical
competence, and (2) regulation, which is divided between four separate authorities, is not up to the task.
If UK shale is to be exploited safely: (1) more sophisticated seismic imaging methods need to be
developed and applied to both basins, to identify faults in shale with throws as small as 4-5 m, and (2)
the current lax and inadequate regulatory regime must be overhauled, unified, and tightened up.

## 1   Introduction

The progress of unconventional hydrocarbon development in the USA cannot be emulated in the UK
for many reasons, not least because the origin and structure of the shale basins are very different. I have



reviewed the structure of these shale basins. The English shale basins are two to one hundred times smaller in area than their US counterparts, but hold a shale target two to one hundred times thicker (Fig. 1, Table 1). The US basins are of foreland or intracratonic type, except for the Eagle Ford and Haynesville-Bossier shale plays of Texas and Louisiana, which are extensional basins on the distal

flanks of the Gulf of Mexico. The data source for the US basin data is the Energy Information Administration, except for the Permian Midland and Permian Delaware Basins, for which the data come from Matador Resources Company and other local industry sources. These last two plays are stacked multiple target plays, not necessarily shale, but requiring unconventional exploration methods. Development of these two plays is at an early stage, and local formation names may vary or be

inconsistent. In Fig. 1 the data for the Bowland-Hodder shale of northern and central England are from Andrews (2013). Only the onshore portion of the Bowland Basin is included. The Kimmeridge Clay (Andrews, 2014) shale play area of the Weald, SE England, is defined as the mature shale area.

The English basins are of extensional origin, often developed during discrete episodes, and sometimes with a local overprint of compression. They are cut by faults, which are predominantly

normal, but sometimes re-activated compressively and/or by strike-slip, and many of which extend upwards from the shale to outcrop. In contrast, the US shale basins locally show thrust or reverse faulting at  the target shale depth, or, in the case of the Gulf plays, minor growth extensional faults, but it is extremely rare for any of these faults to extend up to outcrop. My desk study of the US shale basins (for publication elsewhere) demonstrates that there are fewer than two dozen unconventional wells in all

the major shale plays (totalling in excess of 50,000 horizontal fracked wells) where the drill pad lies within 5 km of an outcropping fault. So in the USA there is not the potential problem of contamination of groundwater by upwards migration of fluids *via* faults as there may be in the UK, where the mapped density of faults linking the shale to the surface outcrop can be extremely high.

## 1.1  Review of faulting in relation to fracking

A joint review of fracking for shale gas by two UK academic societies (Royal Society and Royal Academy of Engineering, 2012) failed to address the problem of through-penetrating faults in the UK shale basins. Much of the report concentrated on the risk of induced seismicity. The problem of pre-existing faults was barely discussed at all, even though it was introduced as a subject for concern by a

submission to the expert committee by the Geological Society of London. Instead, the report accepted uncritically the conclusions of a Halliburton study (Fisher and Warpinski, 2012), as did Green et al. (2012) in their report commissioned by the Department of Energy and Climate Change (DECC).

This uncritical attitude towards an industry publication is surprising, as well as naïve, given that:

- Halliburton's database remains confidential.



- Wells are located only to county level.
- Individual wells cannot be identified on the four main graphs presented.
- We do not know whether inconvenient results have been omitted.
- We do not know how complete is the database.
- There are no wells in areas where complex geology (faults or tight folds) at the shale horizon extends to the surface.

**Table 1.** Maximum Thickness (m) *vs.* Shale Play Area (km²) of US and UK Shale Basins.
Key: AR – Arkansas; LA – Louisiana; MD – Maryland; MT – Montana; ND – North Dakota;
NM – New Mexico; OH – Ohio; OK – Oklahoma; PA – Pennsylvania; TX – Texas.

| Basin Play | Location | Area | Shale Name | Thickness |
|---|---|---|---|---|
| Anadarko Basin | OK, TX | 8711 | Woodford | 200 |
| Appalachian Plateau | PA, OK, OH, MD | 149573 | Marcellus | 270 |
| Ardmore Basin | OK | 3116 | Woodford | 200 |
| Arkoma Basin | OK, AR | 7520 | Woodford-Caney | 100 |
| Williston Basin, US only | ND, MT | 80711 | Bakken + Three Forks | 30 |
| Salt Basin | TX, LA | 29051 | Haynesville-Bossier | 100 |
| Fort Worth Basin | TX | 68489 | Barnett | 300 |
| Permian/Delaware Basin | TX, NM | 20901 | Delaware, Bone Spring | 350 |
| Permian/Midland Basin | TX, NM | 111900 | Wolfcamp-Cline | 400 |
| Western Gulf Basin | South TX | 38028 | Eagle Ford | 100 |
| Bowland Basin | NW England | 1000 | Bowland-Hodder | 1900 |
| Widmerpool Trough | E. Midlands, England | 990 | Bowland-Hodder | 2900 |
| Gainsborough Trough | E. Midlands, England | 1500 | Bowland-Hodder | 3000 |
| Edale Basin | Pennines, England | 995 | Bowland-Hodder | 3500 |
| Weald Basin | SE England | 1088 | Kimmeridge Clay | 550 |

The study covers just four plays, the Barnett shale of Texas, the Woodford shale of Oklahoma, the Marcellus shale in the northeastern US, and the Eagle Ford shale in south Texas. The Bossier-
15 Haynesville shale play of Texas and Louisiana, the Utica shale of Ohio, the Niobrara of Wyoming and Colorado, and the Bakken of North Dakota and Montana are all excluded from the study, even though Halliburton claims leading expertise and experience in all these plays. It would have been useful to know why.





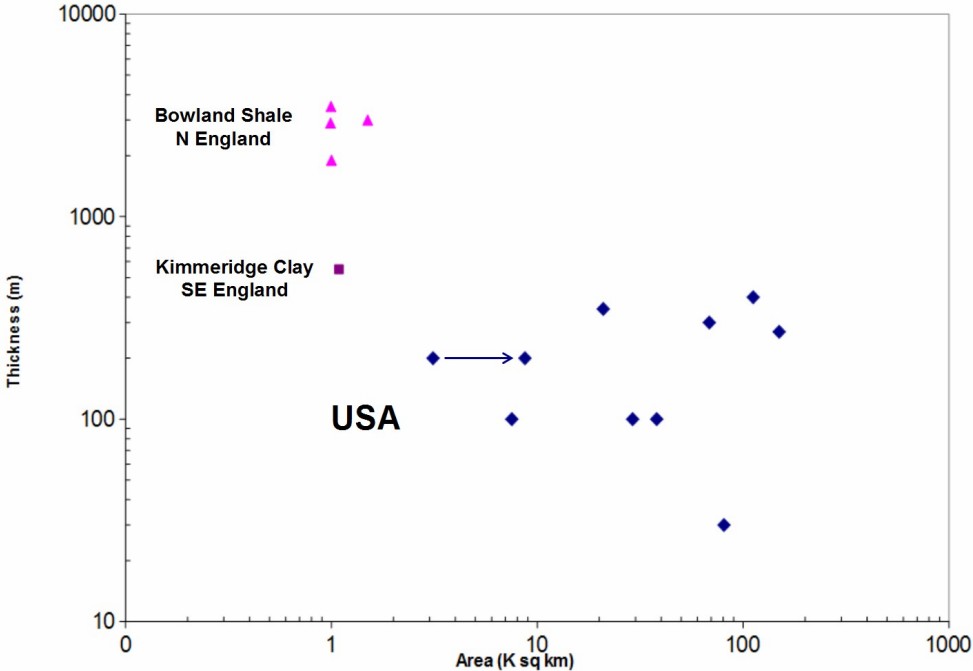

**Figure 1.** Maximum thickness (m) vs. shale play area ($1000 \cdot km^2$) of five UK and ten US shale basins. Data are tabulated in Table 1. The arrow indicates that the small Ardmore Basin is an extension of the larger Anadarko Basin.

There are some surprising facets to the database; for example Cleveland County, Oklahoma, has just one fracked well, but is listed in the graph for the Woodford Shale, whereas several other counties in the Anadarko/Arkoma Basin of Oklahoma, with dozens of wells apiece, have been omitted. The answer may simply be that Halliburton did not have contracts with the operators in these counties, but the problem remains that we simply cannot know, based on the evidence presented.

Even if we accept Halliburton's main thesis at face value – that creation of new fractures by fracking has a natural upward limit above the horizontal wellbore of around 500 m, perhaps 1000 m at the most – the account is erroneous at several places:

    1. Plotting fractures by microseismic monitoring is incomplete. Pettitt et al. (2009) show that a sequence of microseismic events can jump 'silently' up a fault plane to another level, in their

example about 100 m higher. Microseismic activity does not record the passage of fracking fluid up a pre-existing fault.

    2. Such leakage up faults could be a slow process, not necessarily occurring at the time of fracking.



3. The authors argue that if faults were conduits they would have leaked all the gas away by now. This is clearly false; the whole point of fracking is to release gas which is trapped and therefore unable to migrate.

In conclusion the Halliburton study is severely flawed, even when considered on its own terrain of
US geology. It is certainly inapplicable to the UK.

## 2    Shale fracking in the UK

Just under 2200 hydrocarbon exploration and production wells have been drilled to date onshore in the UK. This figure includes deviated wells, noteworthy among which are the extended-reach laterals,
up to 10 km long, drilled from onshore by BP in the Wytch Farm Field under Bournemouth Bay (Cocking et al., 1997, Hogg et al., 1999). Several UK academic experts have maintained that around 200 onshore oil or gas wells in the UK had already been fracked over the last 50 years (Styles, 2013; Verdon, 2013; Younger, 2014). These wells allegedly included the 'extended reach' horizontal wells drilled out out under Bournemouth Bay from Poole into the Wytch Farm oilfield (Verdon, op. cit.,
Younger, op. cit.). The implication is that there is nothing especially novel about fracking shale; but the quoted figures of 200 wells and 50 years are both misleading. Some vertical wells have indeed been stimulated by fracking and other treatments in pursuit of conventional hydrocarbon development, but it is the sandstone or limestone *reservoirs* that have been fracked. No horizontal (lateral) wells in shale, which in conventional exploration terms is a source rock and not a reservoir, have yet been fracked in
the UK using high volumes of low-viscosity water under extremely high pressures, a treatment termed 'super-fracking' (Turcotte et al., 2014). Here is part of the response from the Department of Energy and Climate Change (DECC) to an enquiry about how many wells have been fracked:

*"DECC has records of some kind of the drilling of 2159 onshore wells … we believe that at least 200 did have hydraulic fracturing treatments of some kind, but we would emphasise that these non-*
*shale fracs are not comparable, in the volumes of fluid employed, to Cuadrilla's operations at Preese Hall in 2011 – the non-shale fracs are much smaller."* (Toni Harvey, senior geoscientist, onshore exploration and development, DECC, August 2013; http://www.refracktion.com/index.php/tag/peter-styles/).

The message here from this authoritative source is clear – that the non-shale fracking operations are
of a different and much smaller order. Styles (2013) in a briefing presentation to the South Downs National Park Authority, ignored this important distinction. Concerning the alleged fracking at Wytch Farm (Verdon, 2013; Younger, 2014), the initial publication on the Wytch Farm oil field, Dorset (Colter and Havard, 1981) makes no mention of hydraulic fracturing in the field. Subsequent field development is described by Hogg et al. (1999). The extended reach wells attain a maximum horizontal distance of



10 km from the drill pad, but they have never been hydraulically fractured. They were designed to inject water low down into the oil-bearing aquifer (a 'bottom waterflood') to help the oil flow. This process has no relevance to fracking of any sort.

The British Geological Survey (BGS) has compiled and analysed the unconventional hydrocarbon prospects in three UK regions on behalf of the Department of Energy and Climate Change (DECC). Two of these shale plays, the Carboniferous Bowland Shale of northern England (Andrews, 2013) and the Mesozoic shale plays of southern England (Andrews, 2014), are discussed below. The BGS curates the core and other well data from all the oil and gas operations both on the UK continental shelf and the onshore. It also holds copies of all seismic reflection and other data acquired by industry operating

under an exploration and development licence issued by DECC on behalf of the Crown. The onshore seismic and well data are released through agents after four years from acquisition or well completion, and the cores are also made available for public study on BGS premises.

### 3    The Bowland Basin

The Bowland Basin is in the vanguard of the UK government's drive to promote unconventional oil and gas development. The only UK shale well to have been hydraulically fractured ('fracked'), Preese Hall-1, was drilled here in 2011, but the fracking stages triggered a series of minor earthquakes. Following a moratorium on fracking while the problem of induced seismicity was investigated, the drilling of appraisal wells is due to resume. The three-year moratorium has permitted new insights to be

developed, new geophysical data to be obtained, some existing data to be released under UK regulations for the release of industry data, and new shale wells to be drilled, but not, as yet, fracked.

### 3.1  Structure

The area of interest, called the Fylde, shown in Fig. 2, is flat-lying, and covered with 10-30 m of

post-glacial deposits. Surface geological mapping by the BGS has identified the major faults, but only in a generalised way, due to the poor solid rock exposure. In the Fylde the seismic data mostly date from the 1970s and 1980s, are frequently post-stack migrated, and are adequate for mapping the post-Hercynian sediments. However, imaging of the underlying Carboniferous is poor. These publicly available data were used by the BGS for its structural mapping, together with the well data.

The BGS has compiled a report into the hydrocarbon prospectivity of the Bowland-Hodder shale play of central and northern England (Andrews, 2013). The Bowland-Hodder unit is an informal name given to shales of Viséan to early Namurian age, including contemporaneous limestones developed on shelf areas between depocentres. The BGS study area covers six distinct basins, of which the Bowland is the largest in area, if its offshore portion in the East Irish Sea is included.





The most striking feature is the complex and pervasive faulting (Figs. 2, 3), in contrast to the shale plays of the USA. In the Bowland Basin the shale is 1900 m thick, whereas in contrast, target shales in North America are frequently thinner than is clearly resolvable with surface seismic reflection data (about 30 m). Above the shale (Fig. 3) lie the Upper Carboniferous Millstone Grit (MG) and the

5    Permian Collyhurst Sandstone Group (CS). These are both permeable arenaceous units. The operator is relying on the Manchester Marl Formation, highlighted in pink in Fig. 3, to act as an aquitard (Griffiths et al., 2003) between the aquifers of the Collyhurst Sandstone Formation below and the Sherwood Sandstone Group above. Clearly this is only possible if the numerous faults all act as barriers to fluid flow, including the fault segments in which sandstone is juxtaposed against sandstone.

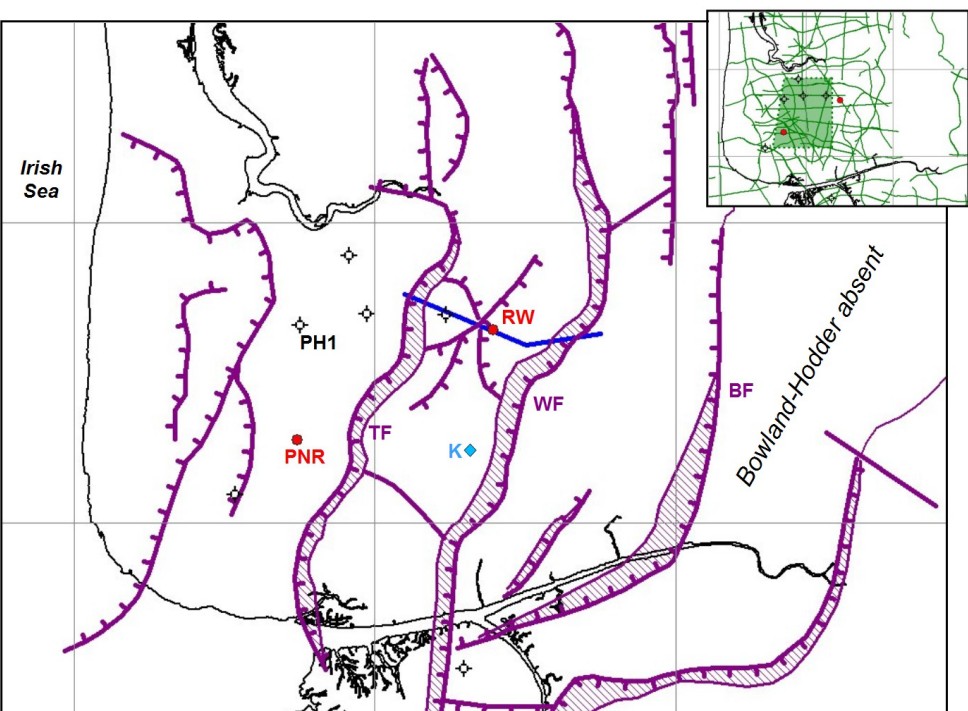

**Figure 2.** Faulting at top Bowland-Hodder (purple lines, comb on downthrown side), from BGS mapping (Andrews, 2013). Coastline in black. Major faults are: TF – Thistleford-Larbreck; WF – Woodsfold; BF – Bilsborrow. Well locations shown by 'dry hole' symbol (PH1 – Preese Hall-1).

15    Cuadrilla proposed wells shown as red circles (RW – Roseacre Wood; PNR – Preston New Road). K – Kirkham geothermal borehole. Blue line – location of cross-section of Fig. 3. Inset map shows 2D seismic line coverage (green lines) and subsurface coverage of Cuadrilla 2012 3D survey (green-shaded quadilateral). Grid - UK national grid at 10 km interval.



East of the Bilsborrow Fault (BF; Fig. 2) the Bowland-Hodder unit is absent, having been removed by pre-Permian erosion. Between the Bilsborrow and Woodsfold Faults (WF) there is a terrace with the Bowland-Hodder at around 1 km deep. West of the Woodsfold Fault, in the area of existing and proposed wells, the top of the Bowland-Hodder is at around 2 km depth.

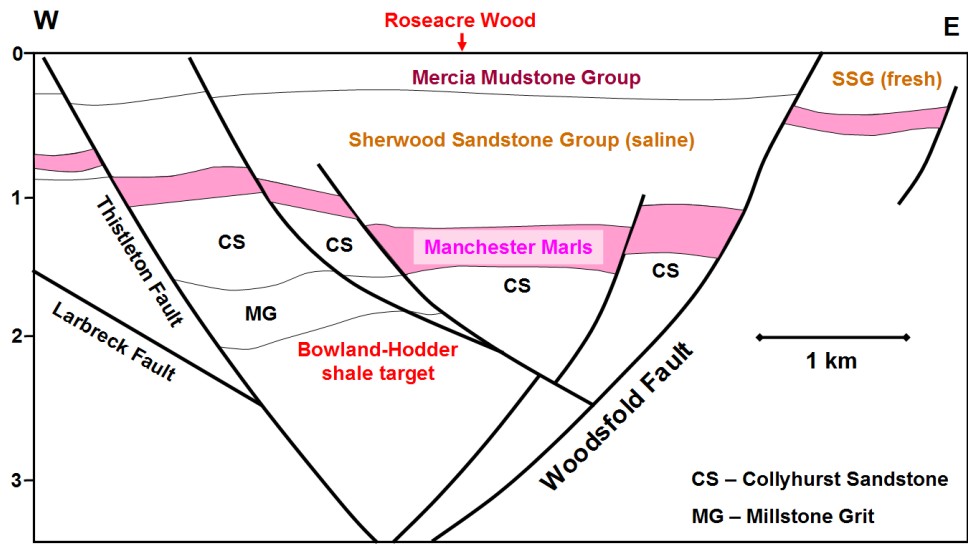

**Figure 3.** Cross-section through the proposed Roseacre Wood shale gas well (labelled RW in Fig. 2), compiled from Cuadrilla and BGS data. Upper Carboniferous comprises Millstone Grit and Collyhurst Sandstone. The Sherwood Sandstone Group at outcrop east of the Woodsfold Fault (SSG) is the most important groundwater aquifer in England after the Chalk. However, below the Mercia Mudstone Group

the SSG is alleged by the UK Environment Agency to be hypersaline, but may in fact be fresh down to 500 m depth. The whole Bowland-Hodder unit fills the central target zone down to the bottom of the diagram, although only the upper Bowland Shale is the current target, labelled in red. The outcrop of the Woodsfold Fault is uncertain, by 650 m to the NW or by 1100 m to the SE of the position shown.

**3.1.1   Available seismic reflection data**

The seismic reflection database comprises 2D lines at a one to two kilometre spacing (Fig. 2, inset). High-quality images of the released seismic data can be freely inspected at the government agency's website, the UK Onshore Geophysical Library (UKOGL, http://www.ukogl.org.uk). The images are offered in colour and greyscale variable area format, at a typical resolution of around 4 pixels per trace

horizontally and 0.4 pixels per millisecond vertically. The sidelabel (the information panel attached to hard-copy displays of seismic sections in the pre-workstation era) is also available as an image, and the navigation data may be downloaded as a shapefile. In short, this excellent resource gives a very good



idea of the data quality. The green area in the inset map of Fig. 2 is a 3D seismic survey carried out by CGGVeritas for Cuadrilla Resources (2012) in Lancashire in 2012, and which is still confidential. It has the following properties:

- Mixed source, 91% explosive, 9% vibroseis
- E-W receiver lines 250 m spacing
- N-S source lines 250 m spacing
- Surface coverage ~100 km²
- Inferred full-fold subsurface coverage ~50 km²

The subsurface coverage figure calculated above assumes a maximum useful offset of 3000 m. A
small-scale field layout map has been published (Cuadrilla Elswick Ltd, 2014), but no further details have been made available. If a source interval of 50 m for the shots (Cuadrilla Resources Ltd, 2011) and a receiver interval of 25 m is assumed, the survey yields a theoretical full fold of 60, which is respectable; but, as can be seen from the figures above, half of the subsurface coverage is low fold, with inadequate offset ranges, and in addition there are many source and receiver gaps which will have
further degraded the coverage.

The commissioning of 3D seismic surveys in areas of unconventional shale exploration is becoming common in the USA, not least because such a survey, say covering 100 km², currently costs around $5M, the same order of cost as one horizontal fracked well ($5-10M). Many articles in oil industry society magazines attest to the close link between 3D seismic surveys and measurement while drilling
(MWD) techniques for drilling horizontal wells (e.g. Durham, 2011, 2012; Roth, 2010; Usher, 2012). If a prior 3D survey exists, the progress of the drill bit can be tracked in real time on a 3D image of the geology. But Cuadrilla only obtained its 3D seismic survey, described as "*feasibility survey*" (Cuadrilla Resources Ltd, 2011), one year after the Preese Hall-1 well was drilled (Cuadrilla Resources Ltd, 2012).

**3.1.2   Revised geological interpretations**

The Bowland-Hodder unit is estimated from seismic interpretation to be 1900 m thick below the Fylde district of the Bowland Basin (Fig. 2), but the whole unit could be up to 4000 m thick in the north of England (Andrews, 2013). It is overlain by arenaceous Upper Carboniferous rocks and post-Hercynian sediments, including the Sherwood Sandstone Group. The Sherwood is noteworthy in this
region as being England's second-most important groundwater aquifer, after the Cretaceous Chalk aquifer of SE England; in the Wessex Basin it is the reservoir of Europe's largest onshore oil field, Wytch Farm, discussed above.

The exploration and earthquake investigations were undertaken in the following order:

1.  Reprocessing of 2D seismic reflection lines (1980-83 vintage).





2.  Geological interpretation.

3.  Drilling of the vertical appraisal well Preese Hall-1 (2010-2011).

4.  Fracking stages 1 and 2 carried out (28-31 Mar 2011).

5.  First events triggered (30 Mar 2011 – 5 Apr 2011), incl. $M_L$ = 2.3 (1 Apr 2011).

6.  Wellbore deformation identified (4 Apr 2011).

7.  Installation of two local seismometers from Keele University (7 Apr 2011).

8.  Fracking stage 3 carried out (8 Apr 2011).

9.  Installation of two extra seismometers (11 Apr 2011).

10. Removal of two Keele seismometers (20 and 28 Apr 2011).

11. Fracking stages 4 and 5 (26-28 May 2011); more events induced, incl. $M_L$ = 2.3 (27 May 2011).

12. Well operations suspended; de Pater and Baisch report commissioned (June 2011).

13.  De Pater and Baisch (2011) report completed (2 November 2011).

14. Acquisition of 3D seismic reflection survey (March – June 2012).

15. 3D seismic data processing (July – October 2012).

16. Geological re-interpretation.

Figure 3 is a cross-section through the proposed Roseacre Wood shale gas well (located by the blue line in Fig. 2), compiled from Cuadrilla and BGS data. Upper Carboniferous comprises Millstone Grit (MG) and Collyhurst Sandstone (CS). The Sherwood Sandstone Group (SSG) aquifer is at outcrop east of the Woodsfold Fault. However, below the Mercia Mudstone Group the SSG is allegedly hypersaline, according to the Environment Agency. The whole Bowland-Hodder unit fills the central target zone down to the bottom of the diagram, although only the upper Bowland Shale is the current target, labelled in red.

Figure 3 has been compiled from the licensee's application for planning permission to drill appraisal wells at Roseacre Wood (Cuadrilla Elswick Ltd, 2014), combined at the SE end with BGS information, being essentially the cross-section from the Garstang solid geology sheet (British Geological Survey, 1990). The operator seeks permission to drill an initial vertical well, then up to four stacked lateral wells to the west in the target shale zone. Cuadrilla has also made a contemporaneous application to drill at Preston New Road, 7.5 km to the SW (Fig. 2). Planning permission to drill and frack the wells at both sites has not, at the time of writing (January 2016), been granted.

The Woodsfold Fault is the most important fault in the region, because it forms the boundary between the Principal Aquifer of the SSG, to the east, from the downthrown Fylde area to the west which will be subject to fracking at around 2000 m depth if the planning applications are granted. But the location of the Woodsfold Fault is not even known to an accuracy of better than hundreds of metres. Figure 3 shows its position, taken from the Garstang map cross-section down to the Manchester Marl,



but then curved concave-upwards below that depth to make it match the Cuadrilla interpretation. But this near-surface fault location is at variance with the fault as mapped at Top Bowland-Hodder (Fig. 2) which, if extrapolated upwards to the surface, would put the fault about 800 m further SE (1100 m to the SE in the plane of the section of Fig. 3). To compound the problem further, a confidential contract

study by the BGS in 1995-96 (British Geological Survey, 1996) places the fault outcrop 500 m to the NW of the position shown in Fig. 3 (650 m to the NW in the plane of the section). The 1996 report was only released under a Freedom of Information request in August 2015. So we do not know, within a range of 1300 m, where exactly this important fault lies; is it 2.1 km or is it 3.3 km SE of the proposed Roseacre Wood wellsite, or at some intermediate distance?

In conclusion, there are four possible interpretations of the location of the Woodsfold Fault to the east of the shale gas area to be exploited; three by the BGS and (at least) one by Cuadrilla.

### 3.2    Hydrogeology of the Fylde

    The UK Environment Agency (EA) is responsible for protecting public groundwater supplies. Let us

examine the evidence for the EA's view that the SSG aquifer west of the Woodsfold Fault is saline and therefore undrinkable. The EA states that this water is stagnant and never recharged, and that over a long period the water has dissolved minerals from the rocks. But the top of this aquifer lies at only around 300 m depth below the relatively impermeable Mercia Mudstone Group (MMG), and groundwater is normally fairly fresh at such a depth. Groundwater usually only becomes saline rather

deeper, say at depths of greater than 500 m.

    The Principal Aquifer lies east of the Woodsfold Fault, and the abstraction wells here are generally drilled to 100 m or shallower depth. Potable water is defined as having a chloride content of less than 250 mg $l^{-1}$. Seawater has a typical salinity of 35,000 mg $l^{-1}$ (i.e. 35 parts per thousand, or 3.5%) . The water sampled in the Kirkham geothermal test borehole (blue diamond in Fig. 2) is hypersaline, with

salinity of up to 91,000 mg $l^{-1}$. There were three such hypersaline samples taken at Kirkham. Two of these were measured at around 250 m depth, within the MMG; the third was measured at an unrecorded depth. Sixteen other samples, five of which come from shallow depths (around 17 m) have low salinity (224 mg $l^{-1}$ or less). The top of the SSG here is at 366 m depth. Therefore it would appear that the groundwater within the SSG has not been sampled. The hypersalinity of the three samples can be

explained by dissolution of thin perched halite layers which exist within the MMG.

    Several other deep wells west of the Woodsfold Fault suggest that potable water was formerly exploited within the SSG. In conclusion, none of the above evidence can justify the EA's over-simplified claim that the SSG aquifer below the Fylde can be discounted as a resource. It has clearly been used as such in the past.





### 3.3 Earthquake and fault location near Preese Hall-1

The Preese Hall-1 vertical well was drilled into the Bowland-Hodder unit in 2011 (Fig. 2), and fracking of the shale was started. It transpired that the basin is near-critically stressed, and the fracking triggered a series of small earthquakes. The shale is gas-prone and over-pressured. Three separate

studies of the earthquake triggering problem have since been undertaken; de Pater and Baisch (2011), Green et al. (2012), and Clarke et al. (2014). The last authors have mapped the locations of the seismic events induced by the fracking, and conclude that they all occurred on a single pre-existing fault favourably aligned to the regional horizontal principal compressive stress component. I do not dispute the conclusions of these authors concerning the locations of the events and the left-lateral sense of slip

assigned to the movement, but I question the validity of their mapping of this fault, its relation to the well, and the wider conclusions that they have drawn regarding a safe distance for fracking in the vicinity of faults.

Preese Hall-1 is noteworthy for being the first UK shale well to be fracked, as noted above (DECC, 2014). Given its importance as the first well of a potential new era of unconventional exploration in the

UK, the sequence of  work listed above shows that the well was drilled and fracked prematurely. UK regulatory guidance on requirements for operators, published in 2013 (Harvey, 2013) now expects that the acquisition, processing and interpretation of the 3D seismic survey, and the installation of a network of local seismometers, should all be carried out *before* a well is tested.

The Minister of State at the Department of Energy and Climate Change, the UK licensor, wrote to

Lord Browne, Chairman of Cuadrilla Resources Ltd., the operator, on 11 May 2012 (Hendry, 2012) to express his concern that the wellbore deformation, which might be linked to the fracking, had been concealed from his officials. This well casing deformation is crucial to understanding the tectonics, as will now be shown.

### 3.4 Tie of the 3D seismic reflection survey and well logs

The Preese Hall-1 well was planned and drilled on the basis of prior released 2D seismic data forming an irregular grid with a typical spacing of 1 to 2 km (Fig. 2, inset). Some of the data were reprocessed. The nearest line runs E-W and is offset to the north from the well by 400 m. The well was deviated to the east below about 2200 m. It appears that the deviation was designed to avoid two faults,

as shown on the reprocessed E-W line by de Paiter and Baisch (2011), on which it can be seen that the evidence for the faulting is very flimsy; this interpretation was revised following the acquisition of the 3D survey.

A composite log of Preese Hall-1 is reproduced by de Paiter and Baisch (2011), and the same log was presented by Cuadrilla (Turner, 2012). The log comprised stratigraphy, gamma ray, rate of





penetration, lithology, total gas, deep and shallow laterologs, compensated sonic, compensated density, and sandstone and base neutron porosity. Depths are measured in feet from the drilling platform, which are the units used in the following discussion.

The zone of interest is from the base of the Pendleside Limestone at 8450 ft, down to 8670 ft. Below

this depth the predominantly mudstone lithology is termed the Worston Shale Formation by Turner (2012) and by de Paiter and Baisch (2011), but elsewhere in the latter it is labelled the Hodder Mudstone. Andrews (2013) refers to the combined Bowland Shale - Pendleside Limestone - Hodder Mudstone sequence by the informal name Bowland-Hodder Unit. Within the depth of interest the lithology is interpreted as predominantly shale/limestone in the proportion 90/10 (Turner, 2012) except

at around 8600 ft, where the proportion of limestone rises to about 50/50.

### 3.5 Re-interpretation of the triggered fault

Clarke et al. (2014) present a sample of the 3D volume in the vicinity of the relocated 2 August 2011 seismic event. A vertical seismic reflection plane oriented E-W intersects a horizontal depth slice at

2930 m depth, that is, the hypocentral depth. A detail of their figure is reproduced in Fig. 4A. The depth slice is foreshortened to give a perspective view of the plane. The authors also show a map of the fault plane in question at 2930 m. At that depth it trends NE-SW and is 4.8 km long. I have reprojected the perspective seismic depth slice to match the fault trend with the map by stretching the slice by a factor of two in the N-S direction, as implied by the relative sizes of the vector scales shown in the bottom

left-hand corner of the projected depth slice. Comparison of the two images shows that the two versions of the fault - the seismic image and the map - mismatch by up to 300 m in the south-westerly direction from the hypocentre. The depth slice shows that the fault is more likely to comprise a zone about 200-300 m wide in the horizontal plane of section, rather than the simple nearly linear fault portrayed on the map.

The fault is shown by Clarke et al. on the vertical seismic section running upwards to the west from the hypocentre, but dying out about 100 m east of the wellbore (Fig. 4A). Such an interpretation mismatches the seismic data, because the dashed line representing the fault in Fig. 4A cuts across the westerly-dipping seismic layering. So I have re-interpreted the fault position to honour the seismic data as shown by the solid white line in Fig. 4B. This version is consistent with the earthquake focal plane

solution. It also intersects the wellbore.

To test this reinterpretation I have replotted the relevant data on a vertical E-W plane (Fig. 5). This requires interconversion between driller's depth down the hole, given in feet, from de Pater and Baisch (2011) and true vertical depth from a sea-level datum (TVD ss) in metres (Green et al., 2012). The earthquake hypocentre is depicted by the lilac ball, with size proportional to the ellipse uncertainty, and



error bar showing the depth uncertainty (Seismik, 2012). Figure 5 shows that the corrected fault location from Fig. 4B runs through the wellbore in the middle of the section of deformed drill pipe (lilac portion of wellbore). Frack stages 1 and 2 are shown in blue and orange, respectively. The seismic events were triggered only after stage 2 fracking, not during or after stage 1.

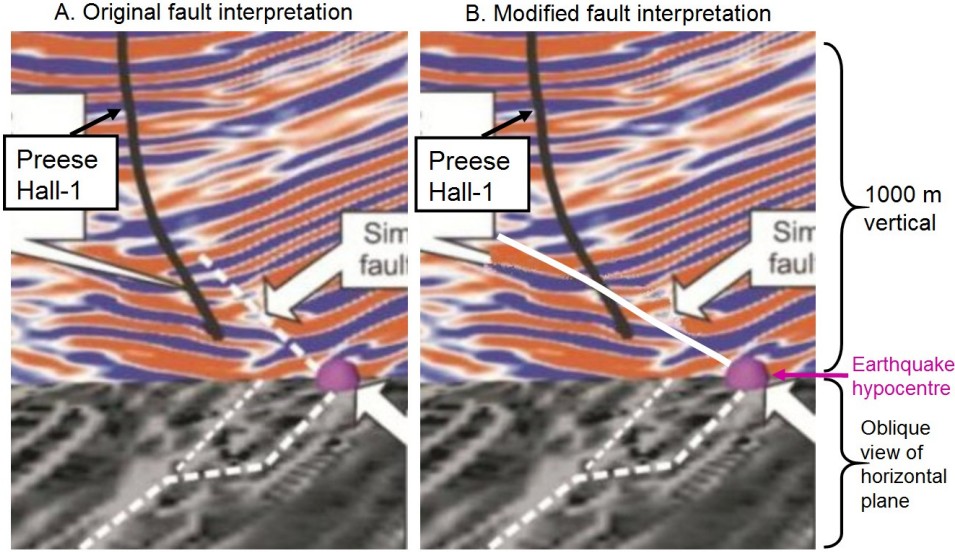

**Figure 4. A.** Extract from Clarke et al. (2014, fig. 4, © American Geophysical Union 2014, reproduced by permission) showing the original fault interpretation (white dashed lines) on a vertical seismic reflection plane (colour) and a perspective (foreshortened) horizontal depth slice (grey). The earthquake

hypocentre is shown by the lilac ball.

**B.** The original fault interpretation on the vertical plane has been digitally removed and replaced by the modified fault interpretation (solid white line), which avoids crossing continuous seismic layering and instead runs up and to the west between two distinct zones of different seismic layering dip.

De Paiter and Baisch (2011) noted that "*the 5½ in production casing was ovalized over a considerable distance of hundreds of ft. This ovalization is possibly related to the fault slip, but in view of the large interval of deformation it is most likely that the wellbore deformation is caused by shear slip on bedding planes, which is possibly associated with the fault slip.*" This conclusion was drawn before the relocation of the seismic events and before the 3D seismic reflection survey was obtained.

The shear displacements causing the significant ovalisation are in fact limited to just 160 ft (8480-8640 ft driller's depth), or under 50 m, not "*hundreds*" of feet, and the explanation of slip on many bedding



planes, which are more or less normal to the deviated wellbore, does not preclude the presence of a fault zone. The low angle of 23° at which the wellbore intersects the fault (in the E-W vertical plane of Fig. 5) leads to a solution to the problem of the extended wellbore deformation.

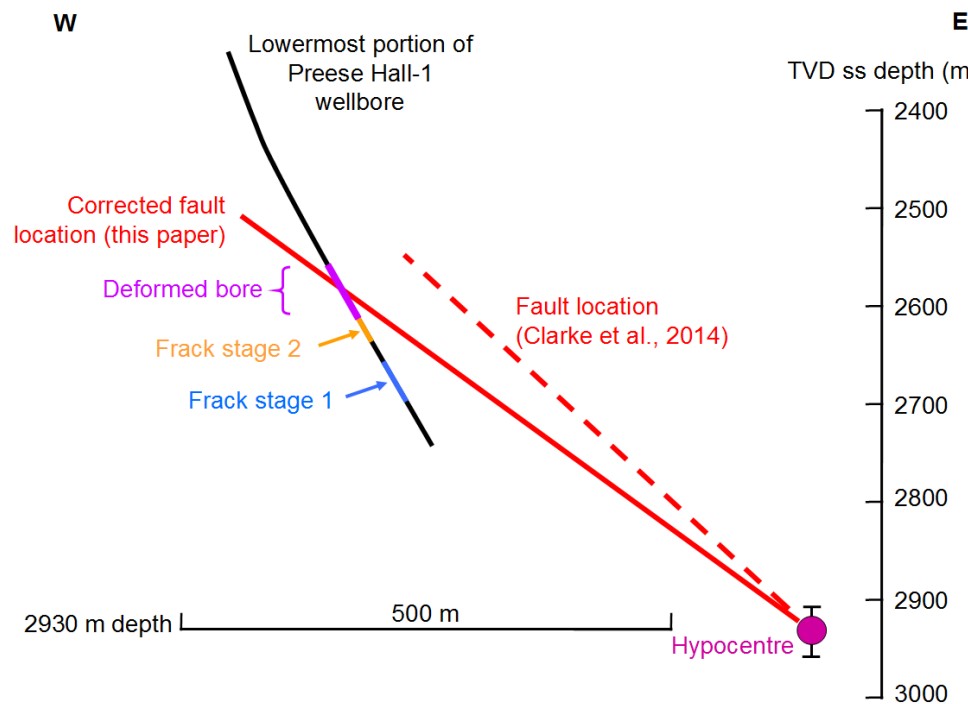

**Figure 5.** East-west oriented vertical projection of lowest part of Preese Hall-1 wellbore (black line) in relation to the hypocentre located at 2930 m depth (size corresponds to ellipsoid uncertainty, vertical bar behind shows depth uncertainty). The corrected fault location corresponds to the modified fault interpretation shown in Fig. 4B. Vertical scale is true vertical depth subsea (below sea level) in metres.

Figure 6 shows the ovalisation of the wellbore over the interval of interest, using data redrawn from de Pater and Baisch (2011). The vertical scale is driller's depth in feet. The left-hand diagram shows the azimuth, in degrees clockwise from north, of the minimum internal diameter of the pipe, i.e. the minor axis of the cross-sectional ellipse. The right-hand graph shows the magnitude of this minimum internal

15   diameter, with units in inches. The pipe is deformed from a nominal internal diameter of 4.8 in to a minimum of 3.5 in. The main zones of deformation are labelled for reference A - E in blue.




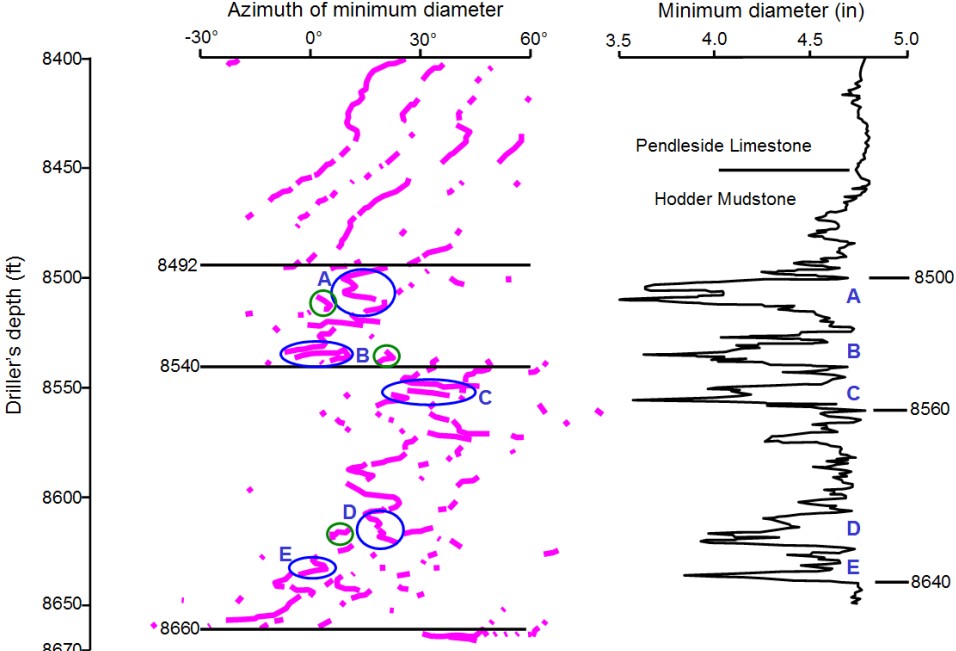

**Figure 6.** Ovalisation of the Preese Hall-1 wellbore, using data redrawn from de Pater and Baisch (2011). Vertical scale - driller's depth (feet). Left-hand diagram: azimuth in degrees from north of the minimum internal diameter of the pipe, i.e. the minor axis of the cross-sectional ellipse. Right-hand graph: magnitude of the minimum internal diameter (inches). The pipe is deformed from a nominal internal diameter of 4.8 in to a minimum of 3.5 in. The main zones of deformation are labelled for reference A - E in blue.

The upper part of the azimuth dataset shows three or four parallel trends. These are spaced at 15°, which is the angular interval, or discretisation, of the fingers of the tool used to sense the diameter. Where there is little or no ovalisation, above about 8500 ft, the fingers ambiguously indicate several azimuths. Based purely on the azimuth data, there are three breaks in the data, marked by horizontal lines at 8492, 8540 and 8660 ft, respectively. The uppermost of these corresponds closely to the upper bound of zone A at about 8500 ft. The five zones labelled A-E on the right correspond fairly unambiguously to the azimuths marked by blue ellipses on the left. Possible ambiguous azimuths, offset by 15°, are marked by green circles.

The significant deformation zone is confined to between 8500 and 8640 ft along the bore. Within that zone there is a dominant section comprising A and B, from 8500 to 8540 ft, but possibly including zone C as well, down to 8560 ft. Zones A-C correspond closely to an increase in log porosity between about 8500 and 8550 ft. Zones D and E are of lesser deformation amplitude. Assuming that the fault obliquely



traverses the wellbore as discussed above, we can estimate the width of this fault as either comprising 40 ft (zones A plus B) or 60 ft (zones A-C) along the bore. These figures yield a fault width of either 4.7 m or 7.1 m. Alternatively we may interpret the zones as three separate fault strands, each one to two metres wide, each separated by a similar distance. The deeper zones D and E may correspond either to a

separate parallel fault strand about 5 m in width and around 10 m away from the main fault, or else to two strands. Overall, the fault zone (A-E) is 17 m wide, made up of five or so separate strands.

### 3.6  Discussion of the Bowland Basin fault problem

The Preese Hall-1 well operator has not, in my view, correctly predicted the location of faults, even

after the location of hypocentres of events triggered by fracking has aided in fault location (Clarke et al., 2014). The original 2D seismic interpretation, on the basis of which the well was planned, mismatches the subsequent interpretation of the 3D survey. Andrew Quarles of Cuadrilla gave a presentation (Quarles, 2014) at the University of Bath in March 2014 in which yet another, very different, version of the faulting was illustrated, even though it was based upon the same E-W reprocessed seismic line as

illustrated by de Pater and Baisch. Quarles did not use a sample of the new 3D survey, the processing of which had been completed six months prior to his talk.

Clarke et al. (2014) explain the supposed lack of direct evidence for the fault in the well as follows: "*Approximately 45 min after the start of the stage 2 injection, the hydraulic fracture encountered the pre-existing fault located some 300m from the injection interval and consistent with a speed of fracture*

*propagation of 4–6 m/min (Fischer et al., 2008). This explains why no direct evidence for the fault was observed at or within the borehole itself.*" So their assumed speed is 6-7 m/min. Fischer et al.'s measured rates are 2 m/min in one horizontal direction and 5 m/min in the other. About 30 minutes elapsed between the drop in wellhead pressure, indicating the start of fracturing, and the beginning of a pressure rise (de Pater and Baisch, 2011). This event warned of a potential screen-out, so proppant

injection was stopped. The rise in pressure, implying fluid loss, was presumably due to the fault being encountered. Microseismic activity then started during stage 2, culminating in the $M_L$ 2.3 event 10 h later.

The relocated fault diagram (Fig. 5) places the stage 2 frack at only 20-30 m from the pre-existing fault, so frack fluid reaching it after 30 minutes implies, following Clark et al.'s reasoning, a low

fracture propagation speed of 0.7–1.0 m/min. But estimating distance in this way is too simplistic; the recent experiment of controlled fluid injection directly into a fault by Guglielmi et al. (2015) shows that there is firstly, a delay before displacement occurs, then a period of aseismic slip, and later there is seismic slip. So at Preese Hall we may safely conclude that the fault was very near frack stage 2. Clarke



et al. have omitted to take into account the *"direct evidence"* for the fault at the well, that is, the casing deformation.

Identification of faults within a thick shale sequence such as the Bowland-Hodder Unit is evidently very difficult, even with the aid of combined 3D reflection imaging and earthquake hypocentral location. The only faults in the Bowland Basin that can be mapped with some confidence are the regional, large-displacement, post-Hercynian normal faults that cut the Permian and Triassic as well as the older rocks (Figs. 2, 3). But even at depth the location of these faults is uncertain. The uncertainty in location of the Woodsfold Fault is discussed above. The Thistleton Fault, a major down-to-the-east normal fault cropping out 2 km east of the Preese Hall-1 well, has a heave (horizontal component of displacement) at the top of the Bowland-Hodder Unit of either 800 m (Cuadrilla Elswick Ltd, 2014) or else 600 m (Quarles, 2014) at the same location. Looking deeper on the same fault, the difference in horizontal location at around 3 km depth within the shales varies by over a kilometre, from the interpreted position of de Pater and Baisch (2011) to that of Quarles (2014). Pre-Hercynian faults cutting the Paleozoic rocks, including basement, are even harder to recognise.

## 4    Faulting in the Weald Basin, SE England

Drilling at Balcombe, Sussex, has been at the forefront of public concern about fracking for shale gas or oil in the Weald Basin. Cuadrilla Balcombe Limited got planning permission in 2013 to drill a new vertical well adjacent to Balcombe-1, which had been drilled by Conoco in 1986 on a gentle E-W trending structural high. Cuadrilla's original application (Cuadrilla Balcombe Limited, 2010) had included a contingent proposal to frack the Kimmeridge Clay, a Jurassic shale, but this proposal was later dropped. A vertical well, Balcombe-2, was drilled in 2013, and was then extended as a horizontal well, Balcombe-2z, in 2014, to target a 30-40 m thick micrite. This target, comprising a thin limestone embedded in thick shales, is closely analogous to the Bakken shale play of North Dakota.

Figure 7C shows the Balcombe wells (Bal) in the area of mature Kimmeridge Clay (hatched area, as defined by the BGS; Andrews, 2013) within the Wessex-Weald Basin (Fig. 7A). Existing wells are shown by red dots. Figure 7B shows the Balcombe locality. Here the mapped surface-outcropping normal faults are shown by red lines with teeth on the downthrown side. The green line is the CDP location of 2D seismic line TWLD-90-27, dating from 1990, and used in Cuadrilla's planning application without having been reprocessed. There is another older seismic line coincident with this one, but otherwise there are no other lines within the map area of Fig. 7B. The nearest seismic lines are sub-parallel to the line depicted, one lying 800 m to the east and another 2 km to the west. Balcombe-1 was mispositioned in Cuadrilla's 2010 application by about 250 m to the north, as shown by the green arrow of its projection onto the seismic profile.





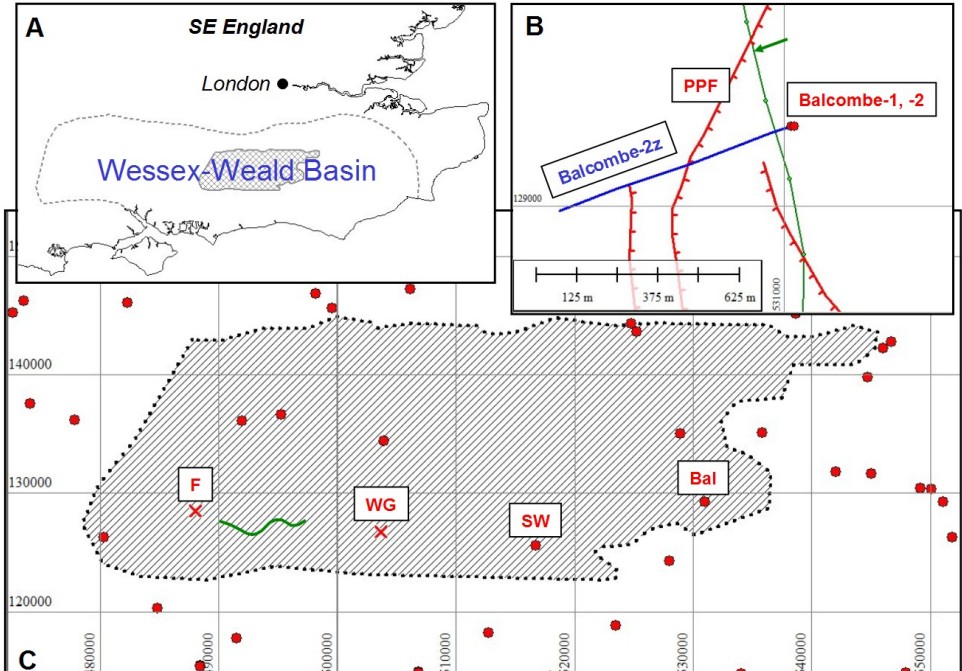

**Figure 7. A**. Location map showing area of thick mature Kimmeridge Clay (cross-hatching) in the Wessex-Weald Basin.

**B.** The Balcombe locality: mapped surface normal faults are shown by red lines with teeth on the downthrown side (PPF – Paddockhurst Park Fault). Blue line - subsurface location of horizontal well Balcombe-2z. Green line – 2D seismic line. Green arrow - location of Balcombe-1 mispositioned by Cuadrilla.

**C.** Area of mature Kimmeridge Clay. Existing wells are shown in red (Balcombe-1, -2 – Bal; SW – Southwater-1). Crosses show two proposed Celtique Energie wells for which planning permission was refused in 2014 (WG – Wisborough Green; F – Fernhurst).

The blue line in Fig. 7B shows the subsurface location of horizontal well Balcombe-2z. It was drilled 'blind' in a west-southwesterly direction, with no seismic control, only using measurement-while-drilling (MWD) technology at the drill bit. It landed in the upper Kimmeridgian micrite (the I-micrite) and followed it for 757 m horizontally from the wellhead.

From the hydrogeological perspective the important issue about this appraisal drilling campaign is whether faults were recognised. The initial planning application ignored the BGS surface fault mapping; however, these faults have throws at the limits of, or smaller than, the resolution of 2D seismic, which is about 30 m. The Paddockhurst Park Fault (PPF in Fig. 7B) has a throw of up to 30-40 m, but reducing



to just 6-9 m SW of the Balcombe drill pad (Gallois and Worssam, 1993). It is not recognisable on the seismic line; however, it is likely to have been transected by one or more of the Balcombe wells.

### 4.1  Faulting interpreted from well logs

I have compared the Balcombe-1 released well log, as a proxy for Balcombe-2, with other well logs in the basin, as reproduced by the BGS (Andrews, 2013). Balcombe-1 is only 10 m east of Balcombe-2. The aim is to see whether the Paddockhurst Park Fault can be recognised cutting the Kimmeridge Clay in the gamma ray and sonic logs of Balcombe-1. The BGS study does not include the two wells about 7 km north of Balcombe, Worth-1 and Turners Hill-1. It does include Bolney-1, km SSW of Balcombe;

the Bolney-1 and Balcombe-1 logs are very closely correlatable in the clay from the top of the I-micrite to about 80 m higher up, but then become poor. The Bolney-1 section is likely to be cut by faults intersecting the uppermost Kimmeridge Clay below the Portland and Purbeck Beds.

#### 4.1.1  Paddockhurst Park Fault cut by the vertical well

The log correlation between Southwater-1 (SW in Fig. 7C) and Balcombe-1 is remarkable. Southwater-1 lies 14.7 km WSW of Balcombe, approximately along strike of the basin axis and sub-parallel to the regional E-W faulting. Figure 8 (left) shows the pairs of logs, gamma ray and sonic, on the left and right, respectively, of the lithological log for Balcombe-1. The Southwater logs are in red, and Balcombe logs in black. The log scales and ranges are 0-150 API units for the gamma ray and 140-

40 μs/ft for the sonic. A near-match, down to the decimetre scale, has been obtained by (a) scaling the Southwater logs to 93% of the Balcombe scale, and (b) inserting a 10 m gap in the Balcombe logs at 620 m driller's depth. This gap, demonstrating missing Balcombe section relative to Southwater, is strong evidence that the normal Paddockhurst Park Fault cuts the vertical well at about 620 m driller's depth (615 m below ground level). The fault geometry corresponds closely to my prediction before

Balcombe-2 was drilled in summer 2013 (www.davidsmythe.org/fracking/cuadrilla%20sussex%20critique%20V2.0.pdf), but with a smaller throw. Given that the 10 m throw is somewhat larger than the estimated 6-9 m throw at the surface, it also implies that the fault penetrates at least as deep again – to 1200 m depth – before possibly dying out downwards. Cuadrilla has not to date acknowledged the existence of this fault.

#### 4.1.2  Fault intersected by the deviated well

Cuadrilla probably penetrated a second fault during its horizontal test of the I-micrite, again without acknowledging its existence. The transition from clay to micrite is gradational, as shown by the logs in Fig. 8A. The micrite has been interpreted from the extrema of the logs (i.e. 100% limestone) by Conoco,



whereas the BGS logs use the transition towards limestone from clay as marking the boundaries (green arrows in Fig. 8A). So according to Conoco and Cuadrilla the I-micrite is 33.5 m (110 ft) thick at Balcombe-1, whereas the BGS interpretation puts the thickness at 41 m.

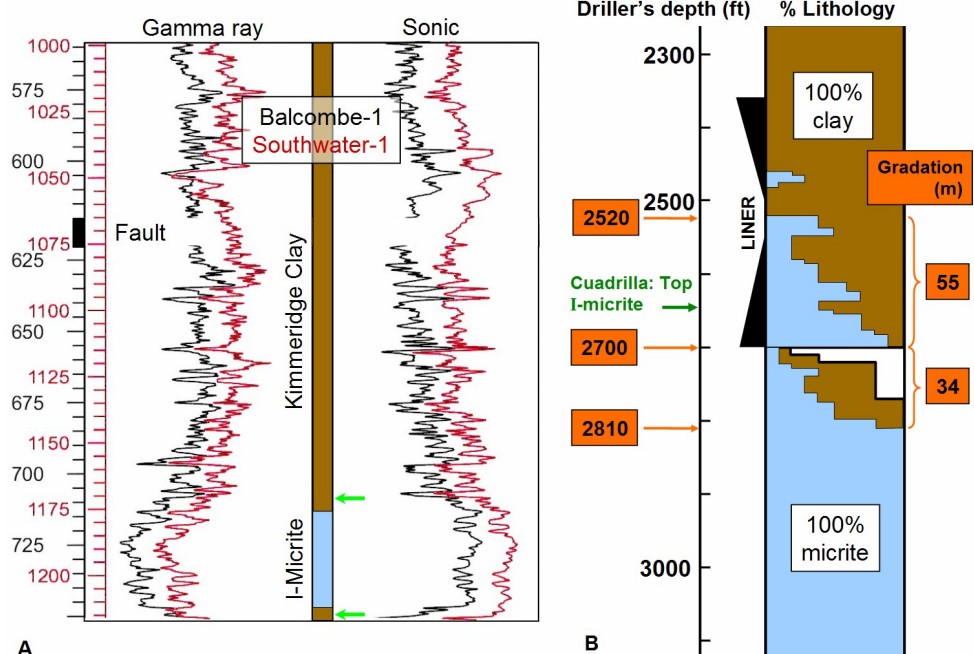

**Figure 8. A.** Gamma ray and sonic logs through Kimmeridge Clay in Balcombe-1 (black) and Southwater-1 (red), redrawn from Andrews (2013). Vertical scale is driller's depth in metres. The Southwater logs have been scaled by 93% and visually matched to the Balcombe logs, and the latter have had a 10 m gap introduced at 620 m. The lithology log corresponds to Balcombe-1, with the I-micrite boundaries from the original 1986 Conoco completion log. The green arrows mark these boundaries as interpreted by the BGS.

**B.** Detail of lithology log from deviated well Balcombe-2z in the zone where it is landing in the I-micrite. Driller's depth in feet is distance along the curved wellpath. At 2700 ft the wellbore is inclined at 15° to the horizontal. The dip of the geological layering is effectively zero. The log shows two gradational changes from clay to micrite. This indicates a normal fault with downthrow to the east.

Figure 8B shows a detail of the lithology log from the deviated well Balcombe-2z in the zone where it is landing in the I-micrite. The portion reproduced is driller's depth in feet, from 2300 ft to 3100 ft (701 m to 945 m). This is not vertical depth, but distance along the curved wellpath. Cuadrilla has simply marked the top of the I-micrite at 2640 ft (green arrow in Fig. 8B. At 2700 ft the wellbore is



inclined at 15° to the horizontal. The dip of the geological layering is around zero. The lithology log shows a gradational change from 100% clay to 100% micrite over 55 m, but the gradation is repeated below 2700 ft, this time over about 34 m. It is possible, but unlikely, to explain the repetition by assuming that two separate logging runs were made and then poorly spliced together; but an alternative

and more plausible explanation is that the wellbore went through a normal fault with a downthrow to the east (wellhead side). In contrast to the case of a vertical well crossing a normal fault at an acute angle, in which case section is missing, like the 10 m gap in Fig. 8A, section is repeated when a near-horizontal wellbore cuts a near-vertical normal fault. The apparent gradational distances of 55 m and 34 m are converted into true vertical gradations (assuming horizontal layering) of 14 m and 8.7 m,

respectively. These figures are 2.7 and 1.7 times larger, respectively, than the gradational distance of 5.2 m inferred from the gamma ray and sonic logs for the clay-to-micrite transition; however, the methods of estimating this transition are different. The throw is of the order of 10 m.

**4.2   Faulting on 2D seismic data smeared out by reprocessing**

Another instance of faulting in the Weald Basin, in the proposed development of shale, comes from Celtique Energie Limited in 2013. The crosses in Fig. 7C show Celtique's two proposed well locations, Wisborough Green (WG) and Fernhurst (F). The company proposed to drill horizontally along one of the two the Kimmeridge micrites, just as Cuadrilla had done at Balcombe.  The Wisborough Green planning application can be viewed on the West Sussex County Council Planning Department website

(http://buildings.westsussex.gov.uk/ePlanningOPS/searchPageLoad.do),     whereas    the     Fernhurst application can be found on the South Downs National Park Authority planning website at http://planningpublicaccess.southdowns.gov.uk/online-applications/applicationDetails.do?activeTab=makeComment&keyVal=MXND0MTU01X00. The applications did not include fracking of the horizontal wells, but the company stated that fracking might prove to be necessary, and that it would

in that eventuality submit additional applications.

Celtique submitted the same 8 km long sample of 2D seismic data in support of both applications. The location of this line was not disclosed by Celtique, but is shown by the green wavy line in Fig. 7C, identified after a search of the UKOGL database. Fernhurst is about 2 km west of the west end of this line, and Wisborough Green about 7 km east of the east end. The nearest seismic lines to Fernhurst lie

500 m to the north, and the nearest to Wisborough Green is about 100 m to the north. At both sites the proposed horizontal wells would be towards the SW, passing nowhere near any seismic lines.

The wavy line shown in Fig. 7C is TER-91-06, for which images and the sidelabel are available to view on the UKOGL website mentioned above. The version shown by Celtique has been reprocessed, and depicts nine colour-coded interpreted horizons. The structure appears to be flat, and no faults are



visible. But the original version, which has been time-migrated, shows clear evidence of faulting. Comparison of the old and the new versions of the line suggest that the reprocessing has smeared out fine details of the structure. This can be achieved, for example, by excessive use of residual statics. The reprocessed version therefore gives a misleading picture of unfaulted geology. Celtique also submitted a

misleading diagram in each application purporting to show that the target limestone was a conventional oil trap, despite being located in the axis of a regional syncline. Lastly, Celtique portrayed a 9 km long north-south shallow geology cross-section, with the Fernhurst proposed wellsite mispositioned by 400 m. Both applications were refused.

## 5    Faults as conduits for contamination of groundwater resources

### 5.1  Recognition of the importance of faults

This is a brief review of what has emerged as a large and complex field of study within the last five years. The first mentions of faults as pathways for contamination by fluid flow in the context of fracking

date from December 2009. Figure 9 is an organogram showing the time evolution, progressively downwards, of the principal reports and papers of which I am aware, and the main links between them. There are both peer-reviewed reports and non-peer reviewed reports, the latter shown in italics in Fig. 9. The numerical modelling studies are shown in green.

The submission to the New York State Department of Environmental Conservation by the Natural

Resources Defense Council (2009) included a memorandum by Dr Tom Myers, who was later the first to publish a quantitative fault modelling study, discussed below. It cited faults as potential contamination pathways. Northrup (2010) followed this up, also focussing on New York State, with the earliest published diagram of a fault linking fracked shale to an aquifer. The state banned shale fracking in June 2015.

In 2011 the University of Montpellier-2 published two explanatory documents on the risks of potential fracking in the south of France (Arnaud et al, 2011; Séranne et al., 2011), following the granting of shale exploration permits in the region a year earlier. They drew attention to the crucial role that faults play in the groundwater circulation system (Bicalho, 2010; Bicalho et al., 2012). France banned shale fracking of shale in July 2011, and in October of that year cancelled all the unconventional

exploration licences.

ALL Consulting LLC (2012) quantified the risk of upward migration of contaminants *via* a hypothetical fault, in a Canadian context, using a static comparison of heads (pressures). This calculation can be dismissed as simplistic and based upon unrealistic parameters. Only five modelling studies have been published to date, worldwide, on the influence of faults on fluid flow resulting from





fracking operations. The aim of these studies is to create a computer model of the geology of interest, including natural geological faults, then add in the hydraulic fracturing process and predict where and how rapidly the fluids (gas and water) migrate.

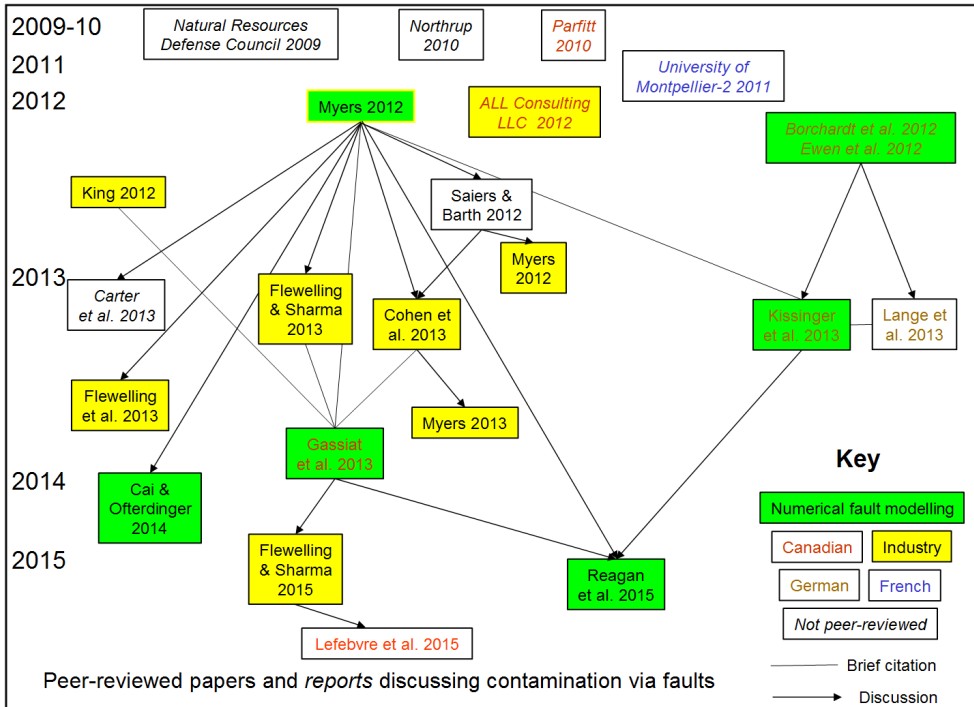

**Figure 9.** Organogram of published papers (upright text – peer-reviewed) and reports (italics – not peer reviewed) related to fluid flow up faults. The green boxes indicate numerical modelling papers. Yellow boxes are papers from industry authors. The links illustrate whether a paper or report has been cited briefly (line) or discussed in depth (arrow).

## 5.2  Fault modelling studies

### 5.2.1  Myers (2012) and critiques

The organogram of Fig. 9 shows that the paper by Myers (2012) aroused a lot of interest, much of it critical. Myers was the first to attempt quantitative hydrogeological modelling of a fault, and have it published in the peer reviewed literature. However, the German study, initially non-peer reviewed, is contemporary with Myers, and more comprehensive.

Table 2 summarises the principal results of the five model studies that predict a time for contamination to ascend. Myers used the Marcellus Shale as a basis for simulation. He assumed a 30 m




thick layer of shale and an overburden of 1500 m thickness, based on typical values for southern New York State. He assumed a homogeneous vertical fault 6 m wide traversing the overburden comprising a mixture of sandstone with subordinate components of shale, mudstone and limestone. The imposed head provided the driving force for flow.

**Table 2.** Five numerical modelling studies of flow up faults

| Year | Authors | Locality | Shale | Transit time (y) |
|------|---------|----------|-------|------------------|
| 2012 | Myers | New York State | Marcellus | <10 |
| 2012 | German study | NW Germany | Various | 30 |
| 2013 | Gassiat et al. | Canada | Utica | <1000 |
| 2014 | Cai & Ofterdinger | England | Bowland | 100 |
| 2015 | Reagan et al. | Generic | Generic | <0.7 (gas) |

He found that faults through the overburden could considerably speed the upward travel time in the steady-state model (before fracking). When fracking occurs the transport times of contaminated fluid

from the fracked shale to the near surface can be reduced to a few tens of years "*or less*". He argues for pre-fracking fault mapping, a 'setback' distance between the frack zone and the nearest faults, and for a system of deep and shallow monitoring wells before development begins.

Saiers and Barth (2012) criticised Myers for neglecting variations in salinity in the groundwater at the Marcellus level, and also for ignoring temperature influences. The boundary conditions for the

model are unrealistic, as is the assumed homogeneity of the overburden. Myers (in Saiers and Barth, 2012) provided a robust response, disputing, *inter alia*, the critics' view of Appalachian geology, the requirement for a 3D model, the need to incorporate density variations, and details of the staged fracking process. He also pointed out that injected fracking fluid is less dense than the briny groundwater at the shale level, and that the resulting convective instability, ignored in his model, would

enhance the upward flow.

Cohen et al. (2013) added additional criticism to the original Myers paper to that of Saiers and Barth. They were concerned about the constant-head assumption combined with fixed boundaries too close to the modelling zone, which forces the flow to be upwards, regardless of the particular scenario modelled. It also implies an unlimited source of water coming into the model from the formation below the shale.

Myers's reply (in Cohen et al., 2013) dismisses Cohen et al.'s claims of model errors as wrong, and goes on to restate his claim that his 2012 model is generally valid.

Carter et al. (2013) criticised Myers, making a valid point about the lack of faults in the Appalachian Plateau, and the nature of those that do exist. The relative lack of faults in the US basins connecting the fracked shale to the surface is one of many differences between US and UK geology (Sect. 1 above).



However, it does not prevent the Myers model being applicable to areas such as the UK shale basins where faults are prevalent; where they connect the shale to the potable groundwater zone; and where they are near-vertical.

### 5.2.2 German study, 2012

The comprehensive German study was published in German (Borchardt et al., 2012), and is therefore not widely known. The English summary of this report (Ewen et al., 2012) does not provide details of the modelling. The report was later published as two peer reviewed papers, of which one (Kissinger et al., 2013) deals with the modelling, but without all the detail of the original German-language report. The report deals in detail with passage of fluids up faults. Seven geological type-localities, or 'settings', were studied. One of them, Quakenbrück-Ortland in the Lower Saxony Basin, has a geological structural style which is remarkably similar to the shale basins of the north of England. The modelling found that contaminated fluid could reach the groundwater resource zone in around 30 years.

### 5.2.3 Canadian study and critique

Gassiat et al. (2013) modelled the fluid transport up a 10 m wide fault zone; a width which they say is consistent with a regional fault having hundreds of metres of displacement. The fault is situated in a regional basin, with the shale being simulated having the properties of the Utica Shale. This shale has a low permeability; the value adopted is 10 ndarcy for unfaulted shale. To contaminate the presumed shallow aquifer they find that the shale must be overpressured, and that it has to have been fracked. The timescale for the migration of contamination is of the order of 1000 years or less. The driving force for flow is overpressure in the shale.

They highlight some caveats in their modelling; single-phase flow only is simulated, and the salinity distribution assumed affects the migration time. An interesting and important result is that a tracer added to the shale fluid reaches the surface at 90% of its original concentration; in other words, 'slugs' of fluid travel upwards without getting significantly diluted. The 'dilute and disperse' model formerly used to justify ocean dumping of contaminants (for example, radioactive waste) evidently does not apply to migration of contaminants up a fault.

Gassiat et al. were criticised by Flewelling and Sharma (2015), who make the same criticism of the Canadian work as others did of Myers, that is, that the faults in the region are not vertical, and that the modelling assumptions force the fluid to go upwards. They also criticised the permeability values chosen for the overburden, and the assumption that the tracer representing contamination is conservative. In response, the Canadian group retorted (Lefebvre et al., 2015) that the modelling was not specifically intended to represent the Saint Lawrence Basin; they merely used conveniently



accessible data from that basin. They re-assert that continuous faults connecting the depth to the surface do exist, and that they are not necessarily self-healing. In summary, they conclude that their main original finding still stands, that "*fluid migration along permeable faults from hydrofracturing zones is plausible under some specific conditions and thus needs to be considered, and that assessment has to*

*consider a long-term time frame.*" Regarding the generic tracer, they note that it could represent a conservative species such as chloride, which could degrade groundwater quality or even limit its use.

#### 5.2.4   Modelling of the Bowland Basin (2014)

One modelling study (Cai and Ofterdinger, 2014) concerns the Bowland Shale discussed above. The

authors built a layer-cake computer geological model based on the geology at the Preese Hall-1 well, then added in hydraulic fractures (fracks) in the Bowland Shale near the bottom of the model. No faults were built into the model. So the 11 geological layers, comprising overburden and underburden, plus the shale, are a realistic simulation of Fylde geology, but without the faults shown in Figs. 2 and 3. In addition each layer is treated as anisotropic, with differing hydraulic conductivities. They did not have

data on the physical properties of the Bowland Shale, so those from the Marcellus Shale were used as a proxy. The effect of faults was then crudely simulated in some models by extending six of the fracks upwards into the Sherwood Sandstone Group (SSG) aquifer. They found that the SSG aquifer could become contaminated on the order of 100 years under certain conditions. Because they put in a sideways-directed head to simulate regional flow from the Bowland Fells west to the Irish Sea, most of

the flow was diverted sideways within the Collyhurst Sandstone (CS in Fig. 3), which is a high permeability layer between the SSG (above) and the Bowland Shale at depth.

The Cai and Ofterdinger study is flawed as a fault study, principally because the representation of major faults by vertical fractures up to 1 mm in width is unrealistic as a model for major pre-existing faults. Their critique of Myers (2012) reveals their misunderstanding. However, the study may be

applicable for the unlikely case of fracks propagating a long distance upwards.

#### 5.2.5   Generic study of gas migration up faults and wells (2015)

Reagan et al. (2015)  have published the first of what is intended to be a series of papers on modelling of the impact of fracking. This paper studies two generic failure scenarios: "*(1)*

*communication between the reservoir and aquifer via a connecting fracture or fault and (2) communication via a deteriorated, preexisting nearby well*." The results are similar in both cases, except that migration *via* the faulty well is faster. They emphasise that their study is parametric, using generalised representations of pathways that might lead to rapid gas transport. The depth to the fracked shale is shallow; overburden thicknesses (between the shale and the aquifer) were either 200 m or



800 m. The timescale for the modelling is limited to 2 years. Simulations (i.e. combinations of parameters) that show significant flow, but where gas 'breakthrough' (into the aquifer) does not approach a steady state, are reserved for a future paper. Their conclusions are principally that gas transport is aided by high permeability of the connecting pathway (fault or well) and by its overall

volume – unremarkable results. They also conclude that production of gas from the shale will mitigate upward migration – again, not a surprising result. The authors do not mention calculated breakthrough times either in the abstract or in their summary, conclusions, and comments section. For the fault simulations and an 800 m overburden distance, the breakthrough times, read from diagrams, are of the order of 0.02 to 200 days (the former figure is under one hour). The times are shorter or longer

depending on gas production strategy.

### 5.3 Case histories

There are many examples of potable water contamination alleged to be due to fracking, *sensu latu*, in the USA, but few details have been published in the scientific literature to date. The peer-reviewed

studies that do exist concern fugitive methane in the vicinity of wells. However, a recent study by Llewellyn et al. (2015)  proves beyond reasonable doubt that contamination of drinking water was caused by passage of frack fluid and/or produced water in part through the geology. Up till now only faulty well construction has been implicated in the contamination process in the many US water contamination case histories.

#### 5.3.1  Bradford County, Pennsylvania well contamination

Here is a brief summary of the history and results of the research. Chesapeake Energy, one of the major operators in the Marcellus Shale play of Pennsylvania, drilled five wells in Bradford County, NE Pennsylvania, in 2009 and 2010. Contamination of private water wells in the vicinity (1200 m away)

started almost immediately. In May 2011 the Pennsylvania Department of Environmental Protection fined the company $900,000. Chesapeake promised to pay for water treatment equipment on selected wells while maintaining that the problems arose from "*pre-existing detectable levels of methane*". The company had previously drilled three new water wells to replace three existing wells, but the contamination continued in these replacement water wells. In June 2012 the homeowners won a civil

case against the company, which had to buy the properties and compensate the owners. The five gas wells were identified as the probable source of the stray gas.

The consultant hydrogeologists acting for the former homeowners are co-authors of the new research. They used a sensitive analytical technique, novel in the field of environmental forensics, to identify the source of the contamination, which included white foam in the water wells, vapour intrusion




in the basement of a house, and bubbling of gas in the Susquehanna River. The new technique identified a specific compound called 2-BE, used in drilling additives, as well as organic unresolved compound mixtures (UCMs) in the impacted wells, whereas no detectable levels of these compounds were found in the background and comparison samples. The analysis rules out the possibility of surface spills of

drilling products or naturally occurring methane as sources of the contamination. The US Environmental Protection Agency (EPA) has recently suggested that 2-BE could be a useful indicator of contamination from fracking activities.

### 5.3.2   Hydrogeology at the Bradford County site

Figure 10 shows a schematic cross-section of the geology in the locality, redrawn and simplified from Llewellyn et al. Vertical exaggeration is about 2.5. The authors discuss how the contamination from the fracked layer, the Marcellus Shale, could have reached the water wells. The geological layers above the shale are gently folded, and a low-angle thrust fault (the solid red line in Fig. 10) is interpreted from seismic data to run from the surface south at an angle of about 16º to sole out into the

Marcellus Shale.

The water wells lie in a narrow linear valley, one of two such parallel features seen on very high-resolution digital elevation models (DEMs) and interpreted by the authors as fracture zones, but not previously mapped on published geology maps.  The fracture interpretation may appear on its own to be somewhat weak; however, a pump test showed that the water flow pattern in the district is aligned along

one of the valleys. This suggests a deep structural control such as the putative fracture zone. The authors cite the evidence of small-scale joints seen in the rock exposures at the surface, also trending in the same direction. In addition, but not mentioned by the authors, there are small normal faults elsewhere in Pennsylvania trending in the same direction, and occupying the same structural location, on the foreland just in front of the Appalachian thrust belt, as this part of Bradford County. The two fracture zones

identified by Llewellyn et al. are therefore probably minor normal faults.

The authors rule out the thrust fault as being a conduit for the contamination, even though it intersects three of the five offending gas wells below the level of the casing shown schematically in blue in Fig. 10. This is because the dip of the fault is low, so that vertical rock stress will tend to keep such a fault held tightly shut. In addition, the rate of progress of the contamination would be very slow along

such a feature. The thrust fault is an interpretation from seismic data which are not publicly available. However, this interpretation is, in any case, questionable, because elsewhere in NE Pennsylvania the few published interpretations of the subsurface faulting suggest that the thrust faulting is divided into two zones; (1) an upper set of shallow-angle thrusts which sole out downwards into the Tully Limestone (the light blue layer in Fig. 10), and (2) a deeper, steep set of thrusts or reverse faults which cut the





Marcellus Shale. In conclusion, the thrust fault, even if it has been accurately identified, is not suspected to be a pathway.

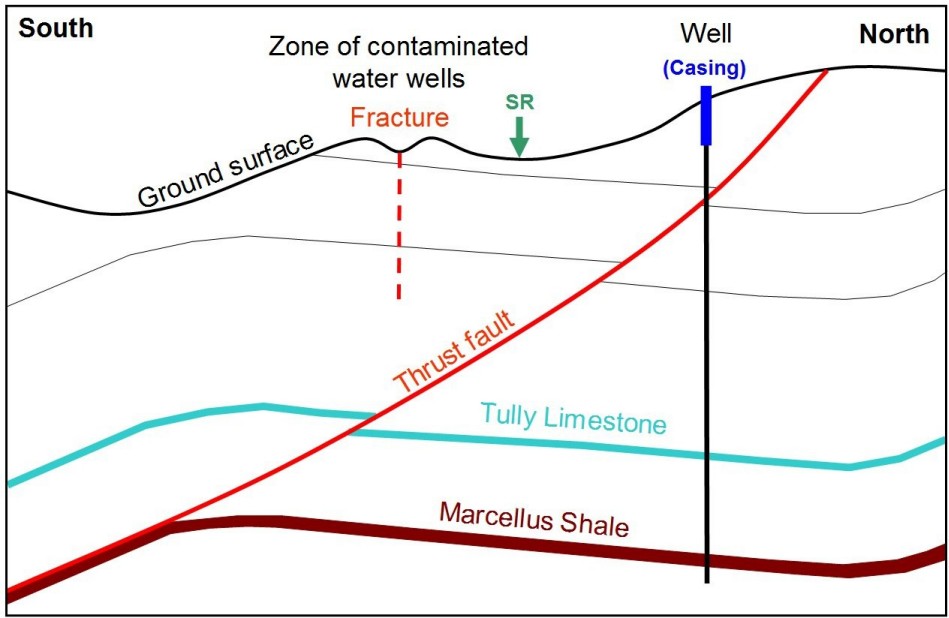

**Fig. 10.** Schematic cross-section to illustrate the salient features of the contamination pathways identified by Llewellyn et al. (2015) in Bradford County, Pennsylvania. The profile is about 10 km long. Vertical exaggeration is about 2.5 to 1. A NNW-SSE fracture zone (one of two identified) is shown by the vertical dashed red line. It is not known how deep it penetrates. The thrust fault is interpreted from unpublished seismic data. The schematic well (one of five) penetrates vertically to the Marcellus Shale, but is only cased (thick blue line) to about 300 m below the ground surface. Gas bubbling was observed in the Susquehanna River (SR).

### 5.3.3 Conclusions on the contamination pathway

Birdsell et al. (2015) have recently stated, in their review of frack fluid migration, that Llewellyn et al. conclude that "*if fracturing fluid did contaminate the shallow aquifer, it is much more likely that the fluid came from a surface spill or from a shallow subsurface leak rather than from the Marcellus*". This statement is completely wrong and misleading. In fact Llewellyn et al. conclude that the most likely pathway for the groundwater contamination is initial passage up the wells from the Marcellus, followed by lateral passage along bedding planes, inclined gently upwards to the south, and finally by travelling vertically upwards along bedrock joint planes and fractures. Overpressured gas well annuli are also implicated as a possible driving mechanism. The approximate pathway lengths from the Marcellus



Shale to the surface are: 1500-1700 m vertically up the uncased wells; 1200-2500 m sub-horizontally along bedding; 0-500 m vertically up faults.

## 6   Discussion

### 6.1  Faults as conduits

What are the lessons for the UK shale basins? The Bradford County, Pennsylvania case history shows that faults and fractures can and do act as conduits for contamination by fracking and subsequent production. But the relative lack of US case histories is due to the fact that the US shale basins do not have through-penetrating faults connecting the shale at depth to the biosphere. In contrast, there are many examples of groundwater and air pollution arising from faulty wells (e.g. Osborn et al. 2011). The lack of faulting in the US shale basins, compared to the English basins, is explicable simply by their differing tectonic style – they are foreland, as opposed to extensional, basins. This difference is equally valid for the many continental European basins in which shale exploration has either begun or has been considered. Some of these latter basins are even more complex, structurally, in that repeated tectonic episodes have been superimposed one upon the other. What they all have in common with the English shale basins is the presence of through-penetrating faults.

The modelling studies to date on flow up faults all confirm that fluid from the fracked shale may use faults as an upward migration route to aquifers; The assertion by Younger (2014) that groundwater flow would be downwards, and not "*upwards to the aquifers*" betrays a misunderstanding of the fault problem, as well as ignorance of the quantitative studies discussed above. The estimated modelling transit times to reach the near-surface vary by two orders of magnitude, from less than ten to a thousand years in the case of liquid. Gas transit is very rapid, of the order of one hour to hundreds of days. Given that such desk studies can be carried out in a few months, and at a tiny fraction of the cost of one well or of a 3D seismic survey, it is remarkable that so few have been undertaken to date.

The Bowland Basin and Weald Kimmeridge Clay exploration examples discussed above demonstrate that faults are only discovered during drilling, if at all. Normal faults are routinely missed in vertical wells.

Hydraulic fractures, or fracks, created by the fracking process can be identified and monitored in real time using a microseismic network. Sometimes the pattern of events indicates passage of the frack fluid along a pre-existing fault, but at other times the existence of a fault can only be inferred by a discrete space-time jump in the cluster of microseismic events from one horizon to another. These passageways are also called 'stealth zones' (e.g. Pederson and Eaton, 2013). This has been demonstrated in the USA (Pettitt et al., 2009, van der Baan et al., 2013). Therefore the use of a microseismic array cannot



guarantee that faults intersecting the fracking zone will be identified. In any case such an identification, if successful, is too late, because the aim is to avoid faults altogether, not to identify them during fracking. However, current UK regulations permit the drilling of faults (if indeed they are identified) either vertically or horizontally, on the way to the fracking zone. This is unacceptable, because cement

bonding of the casing, either in the deviation zone or in the horizontal section of the well, is difficult to achieve (Dusseault et al., 2014). The eccentricity of the drill casing with respect to the borehole means that it is hard to flush out drilling mud, and a subsequent cement job may then fail because the resulting cement-mud slurry does not make not a sound bond. As has been shown  at Preese Hall, a well that penetrates a fault can also be deformed by tectonic movements triggered by the hydraulic fracturing.

This deformation of the casing and the cement will increase the chance that the integrity of the well bore may become degraded.

For a fault to be a potential pathway it does not have to have a large displacement; it merely has to connect the fracked shale to the near-surface groundwater resources at risk; but a large fault will have a wider damage zone than a small one, so will be a greater conductor. The Bowland numerical study,

although geologically unrealistic, demonstrates that a 1 mm wide fracture can transmit fluid upwards in the order of 100 y (Cai and Ofterdinger, 2014). The faults shown in Fig. 3, for example, are not open fractures, but, on the other hand are likely to have permeable damage zones two to three orders of magnitude wider than the 1 mm modelled open fracture.

In the Bowland Basin the operator Cuadrilla has categorized faults as 'regional' or 'local', using its

own definitions. It proposes to drill and frack through local faults, while avoiding regional faults. The latter are defined as those mapped at 1:50,000 scale on BGS maps (Cuadrilla Elswick Ltd, 2014). But this definition is unacceptable, because firstly, it depends too much on the epoch at which the BGS mapped the terrain, and secondly, on the amount of solid rock exposure. There are examples in this basin where such faults stop abruptly at map sheet boundaries because mapping was completed at

different epochs. The surface locations of the regional faults as mapped by the BGS are in some localities inconsistent by hundreds of metres (Sect. 3.6 above). Lastly, no figure is supplied for the distance by which such faults will be avoided.

### 6.2   Regulation of unconventional drilling in the UK

#### 6.2.1   Multiple regulatory agencies

Regulation of unconventional energy is split across four main separate agencies or authorities. In order of action (although there will be some overlap) these are: the Department of Energy and Climate Change (DECC),  the Mineral Planning Authority (MPA), the Environment Agency (EA) and the





Health and Safety Executive (HSE). The following summary is restricted to the geological and hydrogeological aspects of regulation.

Environmental Impact Assessments (EIAs) may be required under European Commission legislation, adopted by Member States. But the exploration stages for unconventional resources, even involving

fracking, frequently avoid the requirement for an EIA by limiting the surface area of the development to under 1 Ha. Most of the sites to date cover an area of 0.99 Ha. The legal aspects of fracking in relation to EIAs have been discussed by Simons (2013). Guidance for councils on unconventional hydrocarbon planning applications was provided, not by any of the agencies mentioned above, but by the Department for Communities and Local Government (2013). This document was only available on the government

website from July 2013 until March 2014. It was then archived with a website redirection to a new website; but the latter (http://planningguidance.communities.gov.uk/) contains no equivalent information.

The British Geological Survey plays no formal role in the planning process. However, it may provide the MPA with guidance on, for example, induced seismicity, but only if so requested. DECC

periodically opens areas for exploration, and then issues petroleum exploration and development licences (PEDLs) to operators on a discretionary basis. It also publishes guidance. It may also, at its discretion, agree an alteration to the PEDL conditions, but is not involved in proposals for the location of wells or the acquisition of geophysical data.

**6.2.2   County Councils and the Environment Agency**

The burden falls upon the MPA – in practice, the planning departments of county councils - to decide whether or not  a particular proposed drilling and fracking application is based upon sound geological data and interpretation; but county councils do not have the in-house expertise to make such judgments, and therefore have to rely wholly on what the applicant chooses to present. The councils have neither

the time nor the money to seek independent advice. The time permitted for determining an application is sixteen weeks, but a requirement by the applicant to resubmit information or revised proposals has sometimes lengthened the decision time. Central government is currently calling in applications, i.e. to make the final decision itself, on the basis that shale development is part of the so-called 'National Infrastructure', and therefore too important to be left to county councils.

The EA has only a limited remit to issue (or withold) a permit, within the context, for example, of perceived risk to groundwater or air. But the EA's experience of hydrogeology only extends to the upper few hundreds of metres of bedrock. Even so, its misinterpretation of the hypersalinity readings in the Kirkham geothermal borehole, leading it to write off the whole of the SSG aquifer below the Fylde as being non-potable (Sect. 3.2 above), demonstrates that the agency has a poor understanding of some of





the resources for which it is responsible. In addition, its hydrogeologists are not necessarily familiar with the bigger picture of shale basins, possibly two or three kilometres deeper than its zone of expertise of groundwater and surface water. In a shale drilling planning application the EA has to submit its report to the council within the sixteen-week period, and, like the county council itself, has neither the

time nor the funding to seek outside advice. EA funding was cut by more than 20% between 2011 and 2015.

### 6.2.3   Relationship between the HSE and operators

The HSE has a role during drilling. In practice, once a drilling plan has been approved, this means

self-reporting by the operator to the HSE. Once again, Preese Hall-1 provides a case history. A series of emails between Mr Mike Hill, a process engineer, and the EA and HSE reveal that there was a well integrity failure, or leak of gas, after the well had been fracked. There is additional email correspondence on this issue between HSE and DECC which Greenpeace (2015) obtained under Freedom of Information legislation.

The well integrity failure was demonstrated by a 377 psi (2 MPa) annular pressure anomaly between the intermediate and the production casings (this anomaly is sometimes referred to as the bradenhead pressure). The HSE described this anomaly to DECC as 'small'. To give an idea of whether or not this problem is serious, the Colorado Oil and Gas Conservation Commission (COGCC), for example, investigated unconventional gas production and concomitant leaks in the long-contentious Mamm

Creek area (Andrews, 2011). Some 2% of the 2867 wells exceeded the level of concern of 150 psi, and remediation was called for in the 12 wells in which bradenhead pressure exceeded 250 psi. So the Preese Hall-1 anomalous pressure of 377 psi should be considered a serious problem. The email records show that a cement bond log (CBL) of the intermediate ( 9⅝ in) casing had never been run, even though Mr Hill had asked the HSE to require it of the operator in 2011. In addition, the CBL that was run on the

production casing proved that the cementing was inadequate, and that DECC knew this but tried to hide it.

Despite this documented history, the operator asserts that "*there have been no leaks to the environment, nor is it believed that there is any prospect of such leaks.*"
(http://www.cuadrillaresources.com/our-sites/locations/weeton/). Its website (as of December 2015)

refers only to a groundwater monitoring report dated February 2014, before the leak started. HSE never visited the wellsite during the drilling and testing phases, but did so only on 30 April 2014, after the integrity failure came to light. The leak has since allegedly been remediated by the operator, by agreement with DECC, and the well has been plugged and abandoned. Groundwater monitoring is to continue for one year. The HSE is no longer involved, nor will be in the future. Monitoring of the





effectiveness or otherwise of the repair and final plugging rests entirely on information that the operator chooses to disclose.

### 6.2.4 Regulation: discussion

The UK academic societies report discussed in Sect. 1.1 above (Royal Society and Royal Academy of Engineering, 2012) also reviewed UK regulation in some detail, but with some confusion and with a perceptible pro-industry bias; it wrongly stated that MPA permission precedes the issue of a permit by the EA; it introduced the irrelevant example of Wytch Farm in asserting that the "*UK has the experience of best practice to draw on*"; it believed that the more wells are drilled, the greater will be

the "*probability of an instance of a failed well*" - an elementary misunderstanding of statistics - unless it was seeking to imply that safety standards drop as more wells are drilled. The committee recommended that the disparate regulatory bodies be brought under one central overseeing body, but this has not come to pass.

On the specific issue of faults in the shale basins, the evidence presented above shows that

prospective operators are currently free to:

- Define what they mean by local and regional faults.
- Drill through faults on the way to the target shale.
- Supply either misleading examples of seismic data, or none at all.
- Drill vertical wells without even 2D seismic data control.
- Drill horizontal wells blind, with no seismic image as a guide for the drill-bit.
- Drill and frack adjacent to major faults.
- Ignore published fault map information.

More generally, they are also able to:

- Start drilling before undertaking adequate baseline monitoring studies.
- Persist in fracking even when seismic activity has been triggered.
- Self-regulate; deciding when (or even if) to inform the authorities of problems like deformed well casing or anomalous wellhead pressure.
- Disguise unconventional appraisal as conventional exploration.
- Salami-slice exploration/appraisal planning applications so that fracking is postponed to a later
planning application.
- Deny that serious problems such as anomalous wellhead pressure may exist.
- Remediate any well problems, then plug and abandon without subsequent independent control by regulatory authorities.



Only the issue of earthquake triggering has been addressed since the Preese Hall-1 experience, by the introduction of a 'traffic light' system of seismic monitoring during fracking. Baseline groundwater monitoring is now being introduced, but only in an inadequate manner; for example, a few 30 m deep boreholes will never detect pollution problems at 1500 m depth.

In conclusion, it is evident from the case histories summarised above that UK regulation of onshore unconventional exploration is inadequate. Hawkins (2015) describes the current regulatory regime, from a legal perspective, as "*far from satisfactory*". Four of the eight members of the working group set up by the joint academic societies (Royal Society and Royal Academy of Engineering, 2012) were earth scientists; nevertheless the group failed to note the fundamental differences between US and UK basin

structure. This suggests that the UK academic community – or, at any rate, that part represented by the academic societies - cannot be relied upon to comment independently and authoritatively on UK shale development.

### 6.3   How to identify faults

Do geophysical methods exist for fault identification in thick shales? Evidently seismic reflection, the best method we have so far, is hardly up to the task. Potential-field methods, including gradiometry, will be of little or no use because of the lack of contrast in physical properties across a fault or within the fault zone, compounded by the depth to the contrast zone. Special processing of high-quality 3D seismic reflection volumes has been shown to reveal hydrocarbon seeps up faults and through cover

rocks (e.g. Aminzadeh et al., 2013), but this requires marine 3D data, which currently have much higher quality than onshore seismic. So the quality of onshore 3D will need to be brought up to a similar standard to that of marine 3D seismic. However, onshore 3D does have an extra attribute not available in the marine sailing environment, and that is the capability of 3-component  (3C) acquisition.

### 25   7   Conclusions

The USA experience of fracking in shale basins cannot be applied to the UK shale basins, or, for that matter, shale basins anywhere in western Europe, because the geometry of the basins is completely different. The major normal faults which cut through the shale to the surface, a universal feature of the UK extensional basins, but absent in the US shale basins, are likely to be transmissive to groundwater,

as the modelling research to date demonstrates. The faulted and mainly permeable cover rocks are an inadequate seal for prevention of upward migration of wastewaters and gas from any future unconventional production.



UK shale exploitation is still at a very early stage, with only one shale well having been fracked to date; that is why this study has focussed on the only two basins where preliminary unconventional exploration has been carried out; the Bowland Basin in Lancashire and the Weald Basin of SE England.

One well, Preese Hall-1 in Lancashire, was fracked in 2011 by Cuadrilla Bowland Limited to test the
shale. The fracking triggered earthquakes. Analysis of two independent datasets – a 3D seismic survey and wellbore deformation – demonstrates that the fault on which the earthquakes were triggered by fracking was transected by the wellbore. This contradicts the conclusion of the operator (Clarke et al., 2014), who determined that the triggered fault lies some hundreds of metres from the wellbore. In short, the operator failed to identify the fault even though it had drilled through it.

Cuadrilla Balcombe Limited, the operator at Balcombe, Sussex (Weald Basin), obtained planning permission to drill and frack in 2010, but subsequently dropped the plans to frack. In 2014 it drilled a vertical well, Balcombe-2, beside an existing conventional dry well, Balcombe-1, and then sidetracked the hole into a horizontal well (Balcombe-2z) along a 40 m thick limestone sandwiched between two oil-prone shale layers, the Kimmeridge Clay. The drilling was blind in that it did not have a pre-existing
image to follow. I have shown that the new wells intersected two normal faults, neither of which were foreseen by the operator, even though the shallower fault is known from BGS mapping.

Another operator in Sussex, Celtique Energie, used the same 2D seismic example to illustrate geological structure in two separate planning applications, for wells some 16 km apart. The seismic section, overlain by the operator's interpretation of unfaulted horizons, had been reprocessed with the
result of smearing out the faulting clearly present on the original version of the data.

The Department of Energy and Climate Change (DECC) regulatory guidance that has been published (Harvey, 2013) requires that any future unconventional shale gas or oil development should identify, in advance of drilling, all faults which have the possibility to be (a) triggered by fracking, or (b) conduits for upward flow of fluid. But DECC does not suggest how this can be achieved, nor to what scale and
precision the faults need to be identified. It is evident from the case histories described above that current legislation and oversight is not competent to prevent shale operators from committing serious errors.

It is going to be very difficult, using current exploration and seismicity-monitoring technology, to identify faults in any thick shale basin. However, a full-azimuth wide-angle, high resolution 3D seismic
survey with 3-component acquisition ('3D-3C') to help resolve imaging problems due to velocity anisotropy, as has been used over the Marcellus Shale in Pennsylvania (e.g. Rebec et al., 2011) and Eagle Ford Shale in Texas (e.g. Treadgold et al. 2011), might go some way to resolving the fault imaging problem.





In the UK there is as yet neither legislation nor guidance on the what should be the minimum ('respect' or stand-off) distances from faults, vertically and horizontally, of both the wellbores and the fracked shale volumes. Other countries such as Germany have concluded that fracking should not be permitted in areas of faulted basins where the faults penetrate the full thickness of the overburden (Kissinger et al., 2013). It would therefore be prudent not to undertake further unconventional exploration in the UK shale basins until the following problems are addressed:

1. New techniques need to be developed for imaging faults within thick shale sequences.

2. Objective, evidence-based criteria for faults which may be fluid conduits (e.g. Lunn et al., 2008) need to be agreed.

3. Respect, or stand-off, distances to avoid triggering of earthquakes and to minimise the possibility of faults acting as conduits must be defined.

The operators themselves must:

1. Carry out the necessary geophysical surveys (which will probably have to involve some advanced method of 3D-3C imaging) in advance of shale drilling.

2. Drill deep monitoring boreholes and to monitor them for a minimum of one year, to set a baseline before shale development starts.

3. Incorporate a passive marker into the frack fluid so that any fugitive fluid may be identified.

4. Monitor microseismic activity during fracking (this requirement is already obligatory).

5. Not be permitted to reinject produced water, due to the risk of triggering or inducing high-magnitude earthquakes.

6. Put up a bond to cover the costs of decommissioning, faulty well remediation, and compensation for possible pollution of water resources, depreciation of land and house prices, and earthquake damage.

The current UK regulatory system is over-complex and not fit for purpose. Its government has adopted a *laissez-faire* approach, in which the exploration companies are trusted to operate and report back to the regulators both honestly and competently; the examples discussed above show that neither is always the case; in short, the operators are acting with impunity. Since the unconventional hydrocarbon industry remains adamant that it causes negligible environmental or third-party economic damage, then insurance for the proposed bond system (item 6 above) will cost very little. What will be unacceptable would be a repetition of the derisory bond system in place in the UK coal industry (see, for example, Monbiot, 2015). The recent comprehensive study by Chernov and Sornette (2016) on the concealment of risk information in contributing to man-made catastrophes suggests that the US shale industry experience to date is such a catastrophe in the making, both from an environmental and a financial perspective. There is no sound reason to risk such a repetition in the UK, or anywhere else in Europe.



In conclusion, the complex faulted geology of the UK shale basins does not favour exploitation by unconventional means. A moratorium of, say, five years would permit the necessary advances in fault understanding and imaging to take place. If fracking of shale is ever to proceed in the UK on a safe environmental basis, far more rigorous regulation of the operators is also required than is current

practice.

**Acknowledgments**

This work is not funded by any external agency. I have no link to any current research group at the University of Glasgow, and declare no competing interests. I thank Andy Skuce and Stuart Haszeldine

for commenting on an early version of part of this paper, Séverin Pistre for discussions of French unconventional exploration, and Mike Hill for discussions on UK shale regulation.

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
