# Peer review of "Hydraulic fracturing in thick shale basins: problems in identifying faults in the Bowland and Weald Basins, UK"

_Solid Earth, 2015_

## Short Comment (SC2) · 5 Feb 2016

Introduction

In this comment on the Smythe (2016) discussion paper I shall try to avoid distraction by this author's record in the field of shale gas and fracking (e.g., Smythe, 2014, 2015), and will concentrate on technical issues related to three topics: the geometry of the 2011 occurrence of induced seismicity at Preese Hall in northwest England; the risk of drawing mistaken conclusions by selective citation of the literature; and the implications of this 2011 case study for the regulation of a future UK shale industry.

.
[Figure]

Geometry of the Preese Hall induced seismicity

The 2011 Preese Hall, Lancashire, case study of induced seismicity that was strong enough to be felt, caused by fracking for shale gas, has been investigated by many workers; my own outputs on this topic (none of which are cited by Smythe, 2016) include the Westaway and Younger (2014) and Westaway (2015, 2016) publications and the Younger and Westaway (2014) report, which was placed in the public domain as a result of a freedom of information request from an environmental activist. Smythe (2016) has proposed an interpretation in which the seismogenic fault transects the Preese Hall-1 borehole within the Hodder Mudstone Formation, one of the shale formations that was fracked. Deformation of the Preese Hall 1 wellbore, which was documented several days after the largest of the induced events in 2011 (with magnitude 2.3), can thus be interpreted as a direct consequence of the coseismic slip, rather than being some sort of independent phenomenon as was inferred in the consultancy reports that were commissioned by the well operator, Cuadrilla, in the aftermath of this induced seismicity.

Notwithstanding differences in detail relating to the geometry, to be discussed below, essentially the same interpretation has been proposed by Westaway (2016); moreover, the latter version has also been presented at conferences (Fig. 1) during which it has been discussed at some length with representatives of Cuadrilla and other subject specialists. Smythe's (2016) interpretation is based on projecting an inferred fault plane from the hypocentral location deduced by Clarke et al. (2014), which is ∼500 m east of and ∼300 m deeper than the observed wellbore deformation, updip at a 30° angle, as is illustrated in his Figures 4 and 5. Smythe (2016) was highly critical of the reliability of much of the Preese Hall dataset, but based his argument on uncritical acceptance of the accuracy of this Clarke et al. (2014) hypocentre. On the other hand, his 30° assumed dip of the fault plane differs from the 70° value determined by Clarke et al. (2014) in their fault plane solution, and also differs from the 45° dip at which Clarke et al. (2014) drew this fault plane on their seismic section (repeated here in Fig. 1(b)).
Smythe (2016) also noticed a number of other vagaries in the Clarke et al. (2014) paper, including apparent mismatches of hundreds of metres between positions of features in different diagrams and problems with the display of the 3-D seismic data, which were also noted by Westaway (2016). The great public interest in this type of dataset and the need for confidence in interpretations to give a future UK shale industry a 'social licence to operate' mean that disclosure of data (such as the Preese Hall 3-D seismic dataset) is a necessity, as Westaway (2016) has suggested. Smythe (2016) has made the same point; somewhat to my astonishment, I therefore find myself for once agreeing with something he has stated.

The unexplained mismatch between these 70° and 45° dip values is one of many vagaries and mistakes in the Clarke et al. (2014) paper that were recognized shortly after its publication and prompted the Westaway (2016) reassessment of the Preese Hall dataset. This reassessment led to the realization that the Clarke et al. (2014) hypocentre is unreliable for two main reasons, making it an inappropriate starting point for analysis of the geometry of the induced seismicity and related rock mechanics. First, Clarke et al. (2014) used a seismic velocity structure around the depth of the Hodder Mudstone Formation that was representative of crustal basement, rather than Carboniferous mudstone, i.e., much too fast, which had the effect of 'pulling' the hypocentre too deep. Second, the WNW deepening of each sedimentary formation in the vicinity (evident in Fig. 1) makes the true seismic velocity structure faster to the ESE than to the WNW. The use by Clarke et al. (2014) of a seismic velocity structure with no such lateral variation thus had the effect of 'pulling' the reported epicentre ESE of its true location. A third issue considered by Westaway (2016) concerned the vagaries in picking of some of the arrival times of seismic phases by Clarke et al. (2014). Taking all these factors into account, Westaway (2016) inferred that the most likely location of the Preese Hall induced seismicity was south of the borehole, rather than east of it as Clarke et al. (2014) had suggested. Since the local stress field will have caused the induced fracture network to develop in a vertical plane oriented north-south, it can thus be presumed that the southward component of fracture propagation resulted in
it intersecting the fault, allowing fracking fluid to leak into the fault and causing the in-
duced seismicity (cf. Davies et al., 2013). However, Westaway (2016) was unable to
resolve the depth of the seismicity relative to the fracking. Another vagary of Clarke
et al. (2014) concerned the focal mechanism of the induced earthquakes: it is evident
even from a cursory inspection of their paper that their fault plane solution is not drawn
correctly and the angles used to represent it are not reported using the standard defi-
nitions. Westaway (2016) determined a new fault plane solution, for which the inferred
fault plane has strike 030°, dip 75°, and rake -20°, indicating almost pure left-lateral
slip on a steep normal fault dipping ESE.

The fault plane inferred by Westaway (2016) projects closer to the Preese Hall-1 bore-
hole than the Clarke et al. (2014) fault plane, and - as Fig. 1(b) shows - can be pro-
jected through part of the seismic section where seismic reflectors are offset. This fault
is evidently steep in the Clitheroe Limestone Formation (CLL), then its dip evidently
flattens upward into the Hodder Mudstone Formation (HOM), this being explained by
Westaway (2016) as an instance of 'stress refraction' caused by the different mechan-
ical properties of these lithologies. The induced seismicity might thus have occurred in
the Clitheroe Limestone Formation, thereby accounting for the steepness of the seis-
mogenic fault plane.

When the illustration in Fig. 1 was first presented in November 2015, its inference, that
this fault plane might continue upward, possibly reverting to a steep dip in the Pendle-
side Limestone Formation (PDL) and maybe also in the overlying Bowland Shale For-
mation (BSG), carries the implication of possible fluid migration to and contamination
of rocks at shallower depths (see also below), as was immediately noted by members
of the audience. Representatives of Cuadrilla were equally quick to assure me that no
evidence of upward propagation of this fault exists in the part of this 3-D seismic sec-
tion excerpt (Fig. 1(b)) that they obliterated with their unnecessarily large label, even
though a fault in roughly the same place is depicted in the older 2-D seismic section in
Fig. 1(a). Publication of the 3 D seismic dataset will resolve this matter.

.

Selective referencing

Westaway (2016) emphasized the need for objectivity in dealing with uncertain datasets such as that discussed above. Although the preceding text indicates that I agree with some of Smythe's (2016) points, I do not regard his approach as good practice: he has uncritically adopted some existing data (notably, the Clarke et al., 2014, hypocentre), rejected others without good reason (or omitted any mention of them), and has 'tweaked' others (notably, the dip of the Clarke et al., 2014, fault plane) to fit what he wants the solution to be. Both Westaway (2016) and Smythe (2016) have nonetheless recognized an essential point, that the fracking that resulted in the largest induced earthquake occurred within the zone of wellbore deformation, where the seismogenic fault intersects the wellbore, making the cause and effect connection between the two indisputable, this point having not hitherto been apparent. However, the detailed geometry of flow of fracking fluid within this fault and the induced fracture network, and the physical cause of the observed time delays between fracking and seismicity, remain to be resolved; Smythe's (2016) analysis has not added anything here beyond details already published, for example by Davies et al. (2013) and Westaway (2015, 2016), none of which references he has cited.

A similar selective approach is evident in the citation of other references by Smythe (2016): for example, he praises the paper by Myers (2012) which deduces rapid upward migration of fracking fluid and the resulting possibility of contamination of shallow aquifers, when many workers (e.g., Saiers and Barth, 2012; Cohen et al., 2013; Cai and Ofterdinger, 2014) have pointed out that Myers (2012) did not construct his numerical model appropriately, casting doubt on his conclusions. Conversely, Smythe (2016) criticises the study by Cai and Ofterdinger (2014), which shows that the creation of induced fracture networks of plausible dimensions will result in no significant contamination of groundwater at shallow depths.

Cai and Ofterdinger (2014) also considered a 'worst case scenario' of a 1 mm wide open fracture reaching to shallow depths, to represent the possibility of an induced fracture network intersecting a large, pre-existing fault, for which contamination was shown to take ∼100 years to reach shallow depths. Smythe (2016) describes this analysis as 'flawed' and 'unrealistic', presumably because he is aware that major faults consist of 'damage zones' that are many metres wide (rather than being open apertures of width 1 mm) through which he thinks groundwater will flow much faster than Cai and Ofterdinger (2014) calculated. However, Cai and Ofterdinger (2014) assigned this aperture a hydraulic conductivity of 0.73 m sˆ-1 which means, for the specified width, a transmissivity of 7.3 x 10ˆ-4 mˆ2 sˆ-1. A subsequent worldwide inventory analysis of drillcore transecting faults, by Ishii (2015) (another reference that Smythe, 2016, does not cite), including data from Sellafield in northwest England, reports an upper bound to fault transmissivity of ∼10ˆ-4 mˆ2 sˆ-1. Rather than being a ridiculously low value, the higher upper bound to transmissivity considered by Cai and Ofterdinger (2014) thus appears to be an exaggeration; these authors were indeed aware of this possibility, since they noted that much of the cross-section of fault 'damage zones' is occupied by rock fragments and carbonate precipitates, so does not provide a conduit for groundwater flow. The true timescale for contamination from fracking fluid to reach shallow aquifers used for water supply is thus likely to be much longer than the 'worst case scenario' estimate by Cai and Ofterdinger (2014), especially if steps are taken to avoid fracking near faults, which might be based on estimation of dimensions of induced fracture networks and/or geophysical logging to reveal the faults (see below).

Smythe (2016) is also highly critical of the Fisher and Warpinski (2012) paper, which demonstrates that the vertical extents of the induced fracture networks produced by fracking have a limit of ∼600 m. Smythe (2016) queries these authors' choice of case study localities, when Fisher and Warpinski (2012) clearly state that they chose the four localities with the most data, and questions the fact that the underlying dataset is proprietary. However, partial disclosure of 'anonymized' proprietary datasets is common in Earth Science, when the only alternative would be non-disclosure, which would

benefit no-one. For example, the BGS / DECC assessments of the shale resource in Britain (Andrews, 2013, 2014; Monaghan, 2014) make extensive use of confidential borehole datasets, which have been reported to BGS (a legal requirement in the UK) on condition that essential details are not made public. Fisher and Warpinski (2012) also showed from first principles, using fracture-mechanical theory by England and Green (1963), how the observed limit to induced fracture growth follows from the practical limit to the volume of water available per frack job. Westaway and Younger (2014) and Westaway (2015) have subsequently showed that this theory accounts for the upper bound to the size (expressed as magnitude or seismic moment) of induced earthquakes caused by fracking (cf. McGarr, 2014). As Westaway (2015) has pointed out, the current regulatory limit for water volume used per frack job (introduced by the Environment Agency for England and supported by the Scottish Environmental Protection Agency for Scotland) of 750 m3 imposes an effective limit of induced fracture growth of $\sim$250 m; the $\sim$600 m limit observed in North America (Fisher and Warpinski, 2012) arises because in U.S. and Canadian jurisdictions developers routinely use much larger volumes of water per frack job, this being one aspect where regulation is relatively tight in the UK (rather than being hopelessly lax as Smythe, 2016, has claimed; see, also, below). In summary, rather than being 'severely flawed', as Smythe (2016) has claimed, the Fisher and Warpinski (2012) study is of fundamental importance; it indeed represents a milestone in understanding the physics of fracking.

.

Implications for regulation

As regards the implications for public policy and regulation of fracking and the shale industry in general, I am afraid that I disagree with almost everything Smythe (2016) has written. In most respects the shale industry is being regulated in the UK by applying directly, or building upon, existing regulations covering drilling and other industrial activity, which have in most cases operated uncontroversially for many years and are accepted as conducive to environmental protection. His repeated claims that regulation of shale

in the UK is extremely lax can therefore be discounted. For example, Smythe (2016) states as an example of supposed laxity that developers are allowed to drill through faults. However, if there were such a prohibition, datasets on the physical properties of faults, such as that recently analysed by Iichi (2015), would be unobtainable, which would mean that misinformation on this topic would not be open to challenge. Smythe (2016) later suggests that it is essential to be able to detect faults using geophysical techniques, so they can be avoided when fracking, but in the absence of drilling there would be no way to validate such investigations. Smythe (2016) also claims that no method currently exists for detecting faults within shale, but Dohmen et al. (2014) have described one approach, known as 'sacrificial well completion', in which shale well laterals are drilled then logged; if a fault is detected (say, from a gamma ray log), a number of planned frack stages on either side of it are skipped, to reduce the risk of induced fractures intersecting this fault. A similar approach may well prove feasible in Britain, but would of course require calibration for the physical properties of the local shales. Smythe (2016) implies that determination of such 'stand off' distances should be a matter of government regulation, but it is arguably more reasonable to leave this to the judgement of shale operators, who will have to balance losses of revenue from the reduced gas production from the 'skipped' frack stages with the possibility of compensating local residents for the nuisance caused by any induced seismicity (see below) and the costs any other environmental issues that might result.

Most of Smythe's (2016) account conflates reporting on what happened at Preese Hall in 2011 with what will be permitted in the UK in future, when in the meantime a regulatory framework has been put in place. As Westaway (2015, 2016) has already discussed, it is clear that the sequence of actions at Preese Hall was far from ideal; the need to do things differently in future is accepted by all stakeholders in this field. For example, as soon as the induced seismicity started, it should have been realised by applying the series of standard tests established by Davis and Frohlich (1993) that the fracking was the cause, so it should have ceased, pending further investigations, rather than continuing for almost two months until being 'voluntarily' terminated just before
the UK government imposed a moratorium. As another example, the in situ stress dataset collected during drilling of Preese Hall-1 might have been analysed before the fracking began, rather than afterwards, and might thus have alerted the operator to the high differential stress in the vicinity and that faults in this vicinity were therefore already near critically stressed, meaning that a clear possibility was apparent that fluid pressure increases associated with fracking might cause induced seismicity. As an alternative, a literature search might have been carried out on the state of stress in Britain, which might have located publications that document measurements indicative of high differential stress (e.g., Cartwright, 1997; Mark and Gadde, 2008). Alternatively, the work of Pine and Batchelor (1984) might have been consulted; this explains why the 1980s' 'hot dry rock' geothermal energy project in Cornwall 'went wrong', as a result of the effect of high differential stress on its geometry of hydraulic fracturing. As a final alternative, they might have read the Westaway et al. (2006) and Westaway (2010) publications that had reported the first discoveries of active faults in Britain and thus provide prima facie evidence of high differential stress at shallow depths. In the light of such information, Cuadrilla might reasonably have decided that the Preese Hall-1 site, which was known to adjoin faults recognized on existing seismic reflection profiles (such as that in Fig. 1(a)), was too 'risky', and might thus have switched their first attempt at fracking to one of their other boreholes in the area.

The one instance of a completely new form of regulation, introduced into the UK since 2011 and praised by Smythe (2016), is the current 'red traffic light' system for 'regulating' induced seismicity, which entails shutting down fracking operations if any earthquake of magnitude >=0.5 occurs. However, as previously discussed (e.g., by Westaway and Younger, 2014), such small earthquakes pose no risk of damage and will probably not even be felt; they might well also be difficult to detect given ambient levels of ground vibration from a wide range of sources in this densely-populated country. Smythe (2016) has argued that shale developers should be made to compensate local residents for 'earthquake damage', but the aforementioned limitation on the magnitude of induced earthquakes, associated with the regulatory limit on the volume of fracking

fluid used, means that 'damage' is most unlikely and one is dealing, at worst, with the possibility of 'nuisance' from ground vibrations (Westaway and Younger, 2014). It is reasonable for developers to compensate for such nuisance, if it exceeds a specified threshold, but this threshold should be set based on strength of ground vibration, not earthquake magnitudes, as has been done in the UK for many years for other forms of 'nuisance' ground vibrations such as those arising from quarry blasting (Westaway and Younger, 2014).

.

Conclusions

Smythe (2016) is arguably correct to infer that the seismogenic fault plane for the 2011 Preese Hall induced seismicity transected the Preese Hall-1 borehole within the Hodder Mudstone Formation and was responsible for the deformation experienced by this part of the wellbore. However, this is not a new deduction, as I have already published it; moreover, the geometry proposed by Smythe (2016) is incorrect in detail, being based on projection from hypocentral co-ordinates reported by Clarke et al. (2014) which were themselves mislocated. Much of the rest of his paper is based on selective use of the literature; he attacks or omits to cite much of this literature and bases his conclusions on work with known flaws as long as it supports his anti shale gas agenda. His inference that regulation of the shale industry in the UK is lax is absurd; much of this regulatory framework has developed by applying directly, or building upon, existing regulations covering drilling and other industrial activity, which have operated uncontroversially for many years.

.

.

References

Andrews, I.J., 2013. The Carboniferous Bowland Shale gas study: geology and re-
source estimation. British Geological Survey for Department of Energy and Climate Change, London, 64 pp.

Andrews, I.J., 2014. The Jurassic shales of the Weald Basin: geology and shale oil and shale gas resource estimation. British Geological Survey for Department of Energy and Climate Change, London, 89 pp.

Cai ZuanSi, Ofterdinger, U., 2014. Numerical assessment of potential impacts of hydraulically fractured Bowland Shale on overlying aquifers. Water Resources Research, 50, 6236–6259.

Cartwright, P.B., 1997. A review of recent in-situ stress measurements in United Kingdom Coal Measures strata. In: Sugawara, K., Obara, Y., eds, Rock Stress: Proceedings of the International Symposium on Rock Stress, Kumamoto, Japan, 7-10 October 1997. Balkema, Rotterdam, pp. 469-474.

Clarke, H., Eisner, L., Styles, P., Turner, P., 2014. Felt seismicity associated with shale gas hydraulic fracturing: The first documented example in Europe. Geophysical Research Letters, 41, 8308–8314.

Cohen, H.A., Parratt, T., Andrews, C.B., 2013. Comment on "Potential contaminant pathways from hydraulically fractured shale to aquifers" by T. Myers. Ground Water, 51, 317–319.

Davies, R., Foulger, G., Bindley, A., Styles, P., 2013. Induced seismicity and hydraulic fracturing for the recovery of hydrocarbons. Marine and Petroleum Geology, 45, 171-185. Davis, S.D., Frohlich, C., 1993. Did (or will) fluid injection cause earthquakes? Criteria for a rational assessment. Seismological Research Letters, 64, 207–224.

de Pater, C.J., Baisch, S., 2011. Geomechanical study of Bowland Shale seismicity: synthesis report. Cuadrilla Resources Ltd., Lichfield, 71 pp. Available online: http://www.rijksoverheid.nl/bestanden/documenten-en-publicaties/rapporten/2011/11/04/rapport-geomechanical-study-of-bowland-shale-
seismicity/rapport-geomechanical-study-of-bowland-shale-seismicity.pdf (accessed 3 February 2016)

Dohmen, T., Blangy, J.-P., Zhang, J., 2014. Microseismic depletion delineation. Interpretation, 2 (3), SG1-SG13.

England, A.H., Green, A.E., 1963. Some two-dimensional punch and crack problems in classical elasticity. Mathematical Proceedings of the Cambridge Philosophical Society, 59, 489-500.

Fisher, K., Warpinski, N., 2012. Hydraulic-fracture-height growth: Real data. Society of Petroleum Engineers, Productions and Operations Journal, 27, 8–19.

Ishii, E., 2015. Predictions of the highest potential transmissivity of fractures in fault zones from rock rheology: Preliminary results. Journal of Geophysical Research, Solid Earth, 120, 2220–2241.

McGarr, A., 2014. Maximum magnitude earthquakes induced by fluid injection. Journal of Geophysical Research, Solid Earth, 119, 1008–1019.

Mark, C., Gadde, M., 2008. Global trends in coal mine horizontal stress measurements. In: Peng, S.S., Tadolini, S.C., Mark, C., Finfinger, G.L., Heasley, K.A., Khair, A.W., Luo, Y. (eds), Proceedings of the 27th International Conference on Ground Control in Mining. West Virginia University Press, Morgantown, West Virginia, pp. 319-331.

Monaghan, A.A., 2014. The Carboniferous shales of the Midland Valley of Scotland: geology and resource estimation. British Geological Survey for Department of Energy and Climate Change, London, 105 pp.

Myers, T., 2012. Potential contaminant pathways from hydraulically fractured shale to aquifers. Ground Water, 50, 872–882.

Pine, R.J., Batchelor, A.S., 1984. Downward migration of shearing in jointed rock during hydraulic injections. International Journal of Rock Mechanics and Mining Sciences
& Geomechanics Abstracts, 21 (5), 249-263.

Saiers, J.E., Barth, E., 2012. Comment on "Potential contaminant pathways from hydraulically fractured shale aquifers" by T. Myers. Ground Water, 50, 826–828.

Smythe, D.K., 2014. Reputational smears in the UK gutter press. Available online: http://www.davidsmythe.org/professional/smears.html (accessed 3 February 2016)

Smythe, D.K., 2015. The insolence of office. Available online: http://www.davidsmythe.org/professional/insolence.html (accessed 3 February 2016)

Smythe, D.K., 2016. Hydraulic fracturing in thick shale basins: problems in identifying faults in the Bowland and Weald basins, UK. Solid Earth Discussion; doi: 10.5194/se-2015-134, 45 pp.

Westaway R., 2010. Cenozoic uplift of southwest England. Journal of Quaternary Science, 25, 419-432.

Westaway, R., 2015. Induced Seismicity. In: Kaden, D., Rose, T.L. (eds.), Environmental and Health Issues in Unconventional Oil and Gas Development. Elsevier, Amsterdam, pp. 175-210.

Westaway R., 2016. The importance of characterizing uncertainty in controversial geoscience applications: induced seismicity associated with hydraulic fracturing for shale gas in northwest England. Proceedings of the Geologists' Association. doi: 10.1016/j.pgeola.2015.11.011, 17 pp.

Westaway, R., Bridgland, D.R, White, M.J., 2006. The Quaternary uplift history of central southern England: evidence from the terraces of the Solent River system and nearby raised beaches, Quaternary Science Reviews, 25, 2212-2250.

Westaway, R., Younger, P.L., 2014. Quantification of potential macroseismic effects of the induced seismicity that might result from hydraulic fracturing for shale gas exploitation in the UK. Quarterly Journal of Engineering Geology and Hydrogeology, 47,
333–350.

Younger, P.L., Westaway, R., 2014. Review of the Inputs of Professor David Smythe in Relation to Planning Applications for Shale Gas Development in Lancashire (Planning Applications LCC/2014/0096 /0097 /0101 and /0102) and Associated Recommendations. Report to Lancashire County Council, 12 pp. + 1 p. preface. University of Glasgow; available online: http://eprints.gla.ac.uk/108343/

.

Rob Westaway, School of Engineering, University of Glasgow, James Watt (South) Building, Glasgow G12 8QQ, U.K.

.

Figure 1.

Adaptation of a Powerpoint slide that I presented at the 'Geomechanical and Petrophysical Properties of Mudrocks' meeting at the Geological Society, London, on 17 November 2015. (a) Excerpt from a west-east-trending seismic reflection profile passing ∼400 m north of the Preese Hall-1 wellhead, shot in 1983 and interpreted by de Pater and Baisch (2011), re-published with adaptations as Fig. 3 of Westaway (2016). (b) Excerpt from the 3-D seismic reflection survey that was commissioned by Cuadrilla in 2012, following the 2011 induced seismicity, rendered as a vertical section. This excerpt is part of Fig. 4 of Clarke et al. (2014), but I have shaded it (using information from de Pater and Baisch, 2011), to depict the stratigraphy; it is published as Fig. 4 of Westaway (2016). The red bar in the base shows the interpreted position of the 2011 seismogenic fault, from Westaway (2016), the red circle and brown bar on the figure being the hypocentre and fault plane interpreted by Clarke et al. (2014), which I regard as incorrect. Further details are provided in the Westaway (2016) figure captions. Both parts are ornamented to summarize (in yellow) the Westaway (2016) interpretation that the seismogenic fault plane is steep below the Hodder Mudstone Formation, then flat-
tens upward into this formation, where it runs through the zone of wellbore deformation that is depicted as a white spot on the black Preese Hall-1 well track. It is also tentatively suggested that the same fault might steepen upward and continue upward for some distance, possibly coinciding with one of the faults depicted in part (a), although this cannot be verified in part (b) as Clarke et al. (2014) obliterated this part of the diagram with their excessively large label.
* * *
[Figure]

**Fig. 1.** Please refer to the main document for this Figure caption.

---

## Editor Comment (EC1) · F. Rossetti (Editor) · 9 Feb 2016

The previous short comment (SC1) uploaded by Dr. Westaway was substituted by an updated version after a complaint from Dr. Smythe.

federico rossetti

---

## Short Comment (SC6) · 18 Feb 2016

In this article, Smythe repeatedly makes some sweeping and erroneous hydrogeological assumptions which he never justifies; indeed the text gives the impression that Smythe is not even aware that these ARE assumptions that require some justification.

The first of these erroneous assumptions is typified by the claim in the abstract that, because faults occur with and on the boundaries of shale basins in Europe "there is thus an inherent risk of groundwater resource contamination via these faults during or after unconventional resource appraisal and development". As a hydrogeologist of 30 years' standing this is the first time I have seen such a claim. Consultation of the principal groundwater textbooks (e.g. Todd 1980; Domenico and Schwartz 1997; Fetter

2001; Younger 2007) will reveal no support for the assumption that faults "inherently" represent potential pathways for groundwater contamination. Some do; many don't. The reality, as explained in those text books, is that faults are hydrogeologically ambiguous: some present permeable zones (most notably where they cut relatively hard rocks such as sandstone, limestone or igneous / metamorphic lithologies); many serve as profound barriers to groundwater flow. Fault zones acting as hydraulic barriers are not only commonplace in aquifers; they are well-known in oil basins (e.g. Spencer and Larsen 1990). The reasons why faults may actually serve as impermeable barriers to fluid movement range from the tendency (especially common where such faults cut shales) for the fault-planes to become tightly lined with 'fault gouge' (i.e. comminuted debris of the wallrocks; Younger 2007) to the compressional closure of otherwise unlined fault planes where these lie oblique to then present-day azimuth of compressive stress in the Earth's crust (e.g. Ellis et al. 2014). Even where optimum conditions exist for faults to display permeability, it is rare for this to be continuous over large vertical intervals: vertical hydraulic connectivity over 10s of metres is common (e.g. Clarke 1962, referring to the Carboniferous of northern England); over 100s of metres less so, though exceptions have been documented where the wall-rocks are particularly competent (e.g. Younger 2005); over 1000s of metres through sequences dominated by shales, which is the case claimed here by Smythe, I am not aware of any cases. Certainly Smythe cites no evidence for this. Indeed the two papers he cites in support of his claims over fault permeability (Bilcalho 2010; Bilcalho et al. 2012) both relate to karstified limestones – the most extremely permeable of all natural hydrogeological systems, in which fault apertures are widened by dissolution of the soluble wall-rocks! No such process occurs in shales. Nor does Smythe offer any evidence that hydraulic fracturing will produce a comparable effect. Rather, the hydraulic fractures produced in shale gas operations have to be propped open; it is physically infeasible for the sand grains used for this purpose to migrate thousands of metres up adjoining faults and prop them open over large vertical intervals.

But even if one were to believe that faults cutting thick shale sequences will defy common hydrogeological experience and become permeable throughout the vertical extent, this still does not mean that there is an "inherent risk of groundwater contamination", as Smythe claims. This is because his analysis makes a second erroneous hydrogeological assumption, namely that there will always exist a hydraulic gradient that favours upflow of water (and any pollutants) to shallow aquifers. Many unconventional hydrocarbon reservoirs are, in the natural state, under-pressured, which means that the pre-development hydraulic gradient is not in itself sufficient to promote upward flow (such as may occur where conventional, over-pressured reservoirs are penetrated). Certainly in the UK, present-day tectonic conditions make under-pressured conditions likely in most shale gas plays. While a net upward gradient will locally be induced in close proximity to the well during the process of hydraulic fracturing, this is likely only to persist over a timescale of hours. Thereafter, shale gas wells are dewatered, to allow the (usually weak) partial pressure of gas in the fractures to overcome the hydraulic head in the production well and thus flow to up through the well to surface. Thus a net downward gradient is actively maintained for as long as the well remains in productive use. Indeed it is precisely because the gradient is oriented this way that shale gas developers avoid creating fractures that provide hydraulic connectivity to prolific aquifers: this would lead to flooding of their wells with unmanageable quantities of water. After a shale gas well ceases to produce economic quantities of gas, this downward gradient will lead to a gradual re-flooding of the dewatered fractures and, eventually, the borehole (though under UK regulations, at least, this will have been long-since blocked by pressure injection of cement). Thus after the end of shale gas production, a long period of slow re-equilibration of hydraulic heads in and around the fractured shale zones will occur. Given the extremely low permeabilities of shale (Yang and Aplin 2007), which will remain at pre-development values throughout the rock mass beyond the relatively modest zones affected by hydraulic fracturing, such a period of re-equilibration can be expected to take many decades, and more likely centuries. Thereafter, it is difficult to conceive of any mechanism by which an excess of head could build up sufficient to establish an upward hydraulic gradient towards shallow aquifers being developed. In

former coalfields, where permeable zones at created by mining at depth are connected up-dip via old workings to distant, elevated recharge zones, such head clearly has developed after the cessation of mining, giving rise to substantial outflows of polluted mine waters at surface (Younger et al. 2002). As deep shale gas zones are not connected up-dip to recharge areas by belts of equally permeable strata, no such through-flow can possibly be established after shale gas exploitation. By erroneously assuming that upward hydraulic gradients will be established, Smythe jumps to physically untenable conclusions.

But let us suspend disbelief, and accept Smythe's tacit (and apparently unwitting) assumptions that faults are permeable enough throughout their vertical extent AND will be subjected to sustained upward hydraulic gradients: will groundwater contamination not then ensue, just as he proposes? No. Even if upward flow is occurring, the loading of pollutants must be sufficient to make a detectable difference in the chemistry of the overlying aquifer. At the most extreme imaginable rates of upflow along Smythe's "through-penetrating faults", the loadings of, say, salinity arriving in freshwater aquifers would need to be far greater than those of the few naturally occurring saline springs in northern England (Younger et al. 2015) to result in detectable changes in water quality in the prolific and fast-flowing freshwater aquifers of the region (which are, inter alia, documented by Younger 1995). These weak saline springs DO NOT contaminate their receiving freshwater bodies, as there flows are far so small that they are immediately diluted beyond detectability. (Incidentally, these natural saline springs DO NOT arise from thick shale sequences, but from sequences dominated by limestones, sandstones and igneous rocks).

Hence Smythe makes three erroneous hydrogeological assumptions that lead to badly mistaken conclusions. I leave the reader to consult my published works and decide whether Smythe is justified in his 'ad personam' references to my work, suggesting I display a "misunderstanding of the fault problem" and "ignorance" of the relevant literature. Similarly, I leave to others to assess whether (p. 19, line 15) Smythe is in any

position to offer readers "the hydrogeological perspective" to which he lays claim.

However, having served on the joint Royal Society / Royal Academy of Engineering panel to which Smythe refers in Section 6.2 of his manuscript, I feel some clarifications are in order. For the benefit of international readers, it is important to clarify that the Royal Society and the Royal Academy of Engineering are not simply "academic societies", as Smythe characterises them in section 6.2.4. Rather, they are the UK's national academies of science and engineering respectively. One cannot apply to join them; Fellowship in both is the highest professional honour in those disciplines in the UK. Nor is the Royal Academy of Engineering even predominantly 'academic' in membership: it is very much an academy of highly experienced, practicing engineers from all disciplines. The report to which Smythe refers was not born of some self-generated initiative; rather the two national academies were asked by the UK Government to independently review the evidence over the risks posed by 'fracking'. The membership of the panel specifically excluded anyone with interests in any shale gas company – which is at odds with Smythe's unsubstantiated opinion that the report (Mair et al. 2012) had "a perceptible pro-industry bias". What it DID have (unlike Smythe's work) is a perceptible grounding in engineering practice and genuine hydrogeological experience.

Smythe's paper includes many similar unsubstantiated opinions, and I frankly do not have time to deal with them all in detail. To mention but two: Smythe's claims about the possibility that fresh groundwater occurs in the Sherwood Sandstones beneath saline water in the Mercia Mudstones is at odds with all known sites in the UK where this setting has been monitored (e.g. in many English coalfields). Furthermore, Smythe pretends to knowledge of the CVs of all hydrogeologists working for the Environment Agency, and suggests they are ignorant of hydrogeological conditions deeper than "the upper few hundred metres of bedrock" (section 6.2.2). This is simply untrue; many EA hydrogeologists of my long-standing acquaintance have direct experience of ground water in former deep mines, for instance, as well as direct experience of deep drilling projects in many parts of the country.

Finally, his summary dismissal of the entire UK regulatory system as "not up to the task" is at stark odds with the findings of the two Royal academies (Mair et al. 2012), and indeed of the subsequent independent expert panel on unconventional gas convened by the Scottish Government (Masters et al. 2014) (on which I also served). I again leave it to readers to decide whether the anecdotal hearsay and unsubstantiated charges levelled by Smythe in section 6.2 should carry more weight than those of the panel of internationally-renowned professionals convened by the Royal academies.

In summary, Smythe's foray into the world of hydrogeology has led him to make several fundamentally erroneous assumptions that do not stand up to rational scrutiny. The result is a paper that pretends to be a serious hydrogeological analysis, but ends up as a pastiche of innuendo, wayward misinterpretation, 'ad personam' and 'ad societatem' slurs, and utterly misleading inferences. It is an exercise in para-hydrogeology, which should not be mistaken for anything more than a dismal parody of true hydrogeological science.

References

Bilcalho, C. C. (2010) Hydrochemical characterization of transfers in karst aquifers by natural and anthropogenic tracers. Example of a Mediterranean karst system, the Lez karst aquifer (Southern France), Thesis accomplished in the Laboratory: Hydro-Sciences Montpellier (UMR 5569, CNRSIRD- UM1-UM2),.

Bicalho, C. C., Batiot-Guilhe, C. Seidel, J. L., Van Exter, S., and Jourde, H. (2012) Geochemical evidence of water source characterization and hydrodynamic responses in a karst aquifer http://www.sciencedirect.com/science/article/pii/S0022169412003733

Clarke, A.M. (1962) Some structural, hydrological and safety aspects of recent developments in south-east Durham. The Mining Engineer, 122: 209 - 231.

Domenico, P.A. and Schwartz, F.W. (1997) Physical and Chemical Hydrogeology second edition, John Wiley & Sons, New York.

Ellis, J., Mannino, I., Johnston, J., Felix, M.E.J., Younger, P.L. and Vaughan, A.P.M. (2014) Shiremoor Geothermal Heat Project: reducing uncertainty around fault geometry and permeability using Move™ for structural model building and stress analysis. European Geosciences Union General Assembly 2014, Vienna, 27th April–2nd May 2014. EGU2014-15069.

Fetter, C.W. (2001) Applied Hydrogeology (4th edn), Prentice-Hall Inc., New Jersey.

Mair, R., Bickle, M., Goodman, D., Koppelman, B., Roberts, J., Selley, R., Shipton, Z., Thomas, H., Walker, A., Woods, E., and Younger, P.L. (2012) Shale gas extraction in the UK: a review of hydraulic fracturing. Royal Society and Royal Academy of Engineering, London. 76pp.

Masters, C., Shipton, Z., Gatliff, R., Haszeldine, R.S., Sorbie, K., Stuart, F., Waldron, S., Younger, P.L., and Curran, J. (2014) Independent Expert Scientific Panel – Report on Unconventional Oil and Gas. Scottish Government, Edinburgh. 102pp.

Spencer, A.M., and Larsen, V.B. (1990) Fault traps in the northern North Sea. Geological Society, London, Special Publications, 55: 281-298, doi:10.1144/GSL.SP.1990.055.01.13

Todd, D. K. (1980) Ground Water Hydrology (2nd edn), John Wiley & Sons, New York.

Yang, Y., and Aplin, A.C. (2007) Permeability and petrophysical properties of 30 natural mudstones. Journal of Geophysical Research, 112, B03206 (doi:10.1029/2005JB004243).

Younger, P.L. (1995) Hydrogeology. Chapter 11. In Johnson, G.A.L., (Editor). Robson's Geology of North East England. (The Geology of North East England, Second Edition). Transactions of the Natural History Society of Northumbria, Vol 56, (5). pp 353 - 359.

Younger, P.L. (2005) Westfield pit lake, Fife (Scotland): the evolution and current hydrogeological dynamics of Europe's largest bituminous coal pit lake, Proceedings of the 9th International Mine Water Association Congress 2005, Oviedo. pp 281-288.

Younger, P.L. (2007) Groundwater in the Environment: An Introduction. Blackwell Publishing, Oxford.

Younger, P.L., Banwart, S.A., and Hedin, R.S. (2002) Mine Water: Hydrology, Pollution, Remediation. Kluwer Academic Publishers, Dordrecht. 464pp.

Younger, P.L., Boyce, A.J., and Waring, A.J. (2015) Chloride waters of Great Britain revisited: from subsea formation waters to onshore geothermal fluids. Proceedings of the Geologists' Association, 126: 453–465. (DOI: 10.1016/j.pgeola.2015.04.001)

---

## Short Comment (SC7) · 19 Feb 2016

I will begin my review with some general comments, before providing section-by-section comments in greater detail.

The most obvious aspect of this paper is that it's not really a scientific paper at all, in any normal sense of the term: i.e., the presentation of a hypothesis, and the marshalling of evidence for and against said hypothesis, in an impartial a manner as possible, and contributing something novel that advances our science understanding, be it by collecting new datasets, by developing a novel theory, by inventing a new method for data analysis, or by contributing to the sum total of human knowledge in some other way.

[Figure]

This paper does none of these things – the only information that we gain by reading this paper is that we learn about Smythe's opinions about various other papers in the literature, as well as his opinions about particular operators currently active in the UK shale gas industry, governmental regulators and the wider UK geoscience academy. As pointed out in the previous comments (SC2, SC4, SC6), this paper even falls short in this limited remit, missing out important papers that contradict the arguments made, and misrepresenting other papers, twisting them to support the authors position when in fact they provide evidence against it.

In places this manuscript is little more than a diatribe, insulting the professional capability of the many engineers and scientists employed by government bodies such as the Environment Agency (EA) and the Health and Safety Executive (HSE). The author is entitled to express his opinions of course, but I do not believe that a scientific journal such as Solid Earth is the appropriate place to do so.

Comments on a section-by-section basis follow:

Section 1: Smythe makes claims about the different degrees of faulting in the USA vs the UK. However, no evidence is provided to justify these claims – a reference to an as-yet unpublished and un-peer-reviewed desk study, the contents of which are not available for readers to consider for themselves, cannot be considered evidence. These claims must be removed unless robust evidence can be provided. Even if a reference were to be provided, I do not believe that such conclusions can be reached from a purely desk-based study because, as noted by Younger and Westaway (2014), "Different geological surveys apply different standards as regards the scale of structures worth recording, depending on the purpose of the mapping. Typically, with much larger territory to map, US survey teams undertaking general mapping record less detail than their UK counterparts".

Section 1.1: While it would indeed be better if all of the datasets used in the Fisher and Warpinski (2012) study were made publicly available, I am in agreement with the com-

ments by Westaway (SC2) that "partial disclosure of 'anonymized' proprietary datasets is common in Earth Science, when the only alternative would be non-disclosure, which would benefit no-one." The underlying insinuation that Smythe is clearly trying to make in this section is that Fisher and Warpinski have somehow distorted or fabricated their results. He has no basis for making this insinuation. Instead, one is left with the view that the hard evidence from hundreds of shale gas wells presented by Fisher and Warpinski directly contradict his own views, and therefore he is trying to smear their professional integrity, rather than engaging with their results.

Fisher and Warpinski is not the only paper to make such observations about fracture height growth during hydraulic stimulation. Maxwell (2011) has independently produced similar plots for a range of shale plays, showing very limited hydraulic fracture growth. Maxwell (2011) has not been referenced in this paper.

For my own part, I have worked with a large quantity of microseismic datasets collected during hydraulic fracturing operations, and the conclusions reached by Fisher and Warpinski match with my own experience: hydraulic fractures rarely extend more than about 50m above the injection zones, and in the most extreme cases have only propagated a few hundred metres above the injection zone, even where they have intersected pre-existing faults. There are also sound physical reasons to expect the height growth of hydraulic fractures to be limited, as described by Flewelling et al. (2013) (another important paper that this manuscript fails to reference) and Maxwell (2011) for example.

Smythe then claims that microseismic data cannot be used to detect fluid movement along a fault. As evidence, he references a paper that uses microseismic data to identify where fluids have moved along a fault. This argument is self-contradictory. Considering the issue more broadly, the interactions between hydraulic fractures and faults are commonly observed and tracked using microseismic data (e.g. Wessels et al., 2011; Maxwell et al., 2008; Maxwell et al., 2009; Hammack et al., 2014). That microseismic events can be used to track pre-existing discontinuities has also been

demonstrated in controlled, laboratory experiments (e.g., Stanchits et al., 2012). Nor is evidence for hydraulic fracture-fault interaction uncommon: microseismic monitoring service providers have estimated that they see evidence for interactions between hydraulic fractures and faults in about 30% of the datasets they acquire (Verdon and Kendall, 2015).

The next assertion that is made is that faults intersected by hydraulic fractures would allow previously-trapped gases to migrate once released by fracking. The erroneous hydrogeological assumptions made have been described at length in the comment by Younger (SC6). Smythe misses the fact that the primary permeability pathways created by fracking must be linked to the well bore. It is possible (indeed it is common) that a hydraulic fracture will intersect a fault. However, if and when it does so, the easiest flow pathway in terms of permeability out of the reservoir rock mass will be along the propped fractures and thereby to the production well. Unless the fault was already well-connected to significant portions of the rock mass (in which case gas will already have escaped), it will not become well connected to the rock mass except through the propped fracture system created by hydraulic fracturing.

Once the stimulation is complete, an operator will begin to produce fluids from the well. As such, the pressure gradient through the permeable system created by hydraulic fracturing will be from the reservoir rock mass, along the propped hydraulic fractures, and into the production well. As such, the direction of fluid flow will follow the most permeable pathway (the propped hydraulic fractures) and the pressure gradient (along said fractures to the wellbore), meaning that the fault is unlikely to provide a conduit for flow away from the reservoir. Engelder (SC4) and Younger (SC6) both raise this same point in their comments.

I also note that Smythe has neglected to reference the most significant study that has been performed into this issue to date (Hammack et al., 2014). The USA National Energy Technology Laboratory conducted an experiment where a faulted section of Marcellus Shale was fracked, using fluids containing chemical tracers to track subsurface

fluid movement. Hammack et al. (2014) monitored for said frack fluid in overlying layers, as well as for increases in methane flux, or even for pressure changes, that would indicate a hydraulic connection between the reservoir and overlying layers through the faults that cut the shale layer. The interactions between hydraulic fractures and faults were observed using microseismic monitoring (as described above). However, no evidence for upward fluid migration or hydraulic connection from the Marcellus into overlying layers, despite the interaction between hydraulic fractures and faults. This paper provides a direct counter-example, with hard evidence, showing that the claims made in this paper do not correspond to reality.

Section 2: This section reads more like Smythe is trying to settle scores with other academics than trying to enlighten the reader. For my own part, I have never tried to imply that the hydraulic stimulations into conventional reservoirs performed in the 1980s are identical to those which are proposed now. Smythe would be better off referencing written work, rather than trying to assume what might have been said at a public meeting based solely on powerpoint slides which are, obviously, an incomplete record of the event.

Regardless, it is equally misleading to suggest that the differences between hydraulic fracturing in shale and in conventional reservoirs is a binary one. Hydraulic fracturing technology has seen continuous evolution over the last 65 years – I recommend for example the history provided by Montgomery and Smith (2010). Many of the stimulations now carried out in conventional reservoirs would meet the "super-fracking" definition defined by Turcotte et al. (2014).

The DECC statement provides an accurate statement of the present situation, and could be referenced without the need for 2 pages of additional discussion. However, if Smythe does wish to persevere with an extended discussion about the state of hydraulic fracturing in the UK, then the situation offshore in the North Sea must also be discussed, where multi-stage fracking in horizontal wells using large fluid quantities – what is referred to in this paper as "super-fracking" – is a common activity (e.g. Cham-

bers et al., 2010; Schrama et al., 2012).

Section 3: In this section, Smythe presents a re-interpretation of the fault position presented by Clarke et al. (2014). That different geoscientists sometimes come to different geological interpretations, is hardly surprising. While the exact hypocentre location as found by ourselves (O'Toole et al., 2013) and Westaway (2016) are slightly different, I find myself in broad agreement with the overall theme of Westaway's comments here (SC2). This section is not providing anything new that has not already been published.

Other authors have already published different interpretations of the earthquake and fault discussed by Clarke et al., (e.g., O'Toole et al., 2013; Westaway, 2016). The simple fact of the matter is that these events were not well monitored – they were recorded by 4 stations at ∼100km distance, one fully functional local (<5km distance) station, and one partially functional local station (where the vertical component was functional, but the horizontal components were not). So I'm sure that if we were to give the data to another 10 different geophysicists, they'd come up with 10 slightly different interpretations. But publishing 10 different scientific papers every time would not serve to advance our scientific understanding in any way, and would not be an appropriate use of the academic literature, unless such a re-interpretation substantially changed our understanding of how hydraulic fractures and faults interact (which this paper does not).

It should also be pointed out that the interpretation of Clarke et al. (2014) is based on the digital 3D seismic volume itself, while the re-interpretations made by both Westaway (2016) and this paper are reliant on a small picture of a 2D slice through the data. Therefore, all other things being equal, one would expect the interpretation presented by Clarke et al. to be more robust. Again, as stated in Westaway's comment (SC2), it would of course be better if the full 3D volume was made available to academics, but it remains a commercially sensitive dataset.

Section 4: This section discusses conventional oilfield activities in the Weald Basin

at Balcombe. The UK's 2015 Infrastructure Bill precludes hydraulic stimulation taking place at depths of less than 1,000m. The wells in question are at depths of approximately 800m. It is therefore difficult to see the relevance of this section to the development of hydraulic fracturing in the UK. Smythe's apparent unfamiliarity with significant legislation relating to hydraulic fracturing in the UK is discussed in more detail in my comments on the following sections.

Section 5: I find myself in agreement with the comments made by Westaway (SC2) and Engelder (SC4) that the review of hydraulic fracture-fault interactions is flawed. It misses the most significant study performed to date of hydraulic fracture-fault interaction (Hammack et al., 2014). In as much as this study conducted a real experiment, rather than a modelling study, it's conclusions should carry far more weight than modelling studies. The fact that it is not included in this review is surprising. I share the same concerns about the realism of some of the referenced models as raised by by Engelder (SC4), particularly that they do not simulate flowback and subsequent hydrocarbon production, which produces an inward pressure gradient. Myers (2012) does not appear to include gas production in their simulation, while Cai and Ofterdinger (2014) do not even appear to include flowback of fracking fluid in their simulations.

Section 5.3: Smythe presents the Bradford County case (Llewellyn et al., 2015) as an example of fracking-related contamination caused by faulting, and therefore to provide a justification of his concerns regarding faults intersecting shale gas reservoirs. I do not have Engelder's (SC4) in-depth and first-hand knowledge of this particular case study, but as he points out, it is obvious even to the layman that Smythe (2016) is severely misrepresenting this case study, because it does not in fact support his case (i.e. that faults can provide a pathway for contamination from fracking depths to the surface). I note in passing that Smythe has made the same misrepresentations about Llewellyn et al. when submitting objections on behalf of an anti-fracking activist organisation to planning applications made by UK operators (e.g., Smythe, 2015).

This paper attempts to use Llewellyn et al. (2015) as a case study where a fault has

provided a pathway for fluids to migrate from depth to contaminate groundwater. In fact, if fluids have propagated upwards from depth (and I note the caveats mentioned by Engelder, SC4), the migration pathway would be the poorly-cemented wellbore. Had the well been drilled with the appropriate procedures, no groundwater contamination would have occurred, regardless of the presence of the fault.

In this case, faults and/or fractures seem to have only provided a pathway for fluid migration in the upper 300m or so of the subsurface. To use an example of a fault providing a flow conduit within a few of hundred metres of the ground surface to imply that faults at thousands of metres depth will provide flow conduits to the surface, demonstrates a poor understanding of hydrogeology for a number of reasons (also discussed by Younger, SC6): - In these near-surface layers, compressive stresses will be very low, and so faults and fractures are much more likely to provide conduits to flow. At depth, as compressive stresses increase, faults and fractures will become less conducive to fluid flow. - Faults at depth are more likely have experienced more diagentic effects (mineral precipitation for example) that will occlude flow pathways, again making them less conducive to flow. - A fault will cut through different lithologies at different depths, with clay-rich layers likely to smear along the faults, affecting the permeability. A fault in near-surface layers will be likely to have different properties to the same fault at greater depths. So the fact that fluids might have flowed along a fault in the near surface does not mean that the same fault could provide a flow conduit from 2km depth to the surface.

It is clear that Smythe is twisting the conclusions reached by Llewellyn et al. (2015) because he is otherwise unable to find a single case example of fracking-related contamination via fault leakage. I am happy to add the best of my knowledge to the (far more extensive) best of Engelder's knowledge (SC4) in stating that the literature does not contain such a case.

Section 6: I will refrain from detailed comment on what is essentially an attempt to discredit the various regulatory bodies that are involved in the UK. I am in agreement

with Westaway (SC2) and Younger (SC6) that the claims made about regulations in the UK are not appropriate. I believe that these agencies are capable of stating for themselves whether or not they are competent to perform the job in hand. Indeed, should Solid Earth decide to proceed with publication of this paper, I would suggest that the editors contact representatives of the EA and HSE for comment prior to doing so, since this paper makes a direct challenge as to the competence of their employees.

However, I will point out some of the more egregious examples of the invective used in this section.

Smythe claims that Younger (2014) misunderstands the "fault problem", and accuses him of "ignorance" (without any attempt to produce evidence or explanation as to why Younger's position is incorrect). This is simply a personal slur without any basis. In the General Terms section of the Solid Earth website, it is stated that "The SE editorial board reserves the right to remove referee reports and any other comments if they contain personal insults." I would assume that the same conditions apply to manuscripts submitted to the journal.

Smythe claims that the EA "has a poor understanding of some the resources for which it is responsible". Unless he has carried out some sort of performance review of the EA's capabilities, he is in no position to make such a statement, which insults the competency of the many hundreds of engineers and scientists employed by the EA, and is completely unacceptable for the scientific literature.

The statement that UK academic community "cannot be relied upon to comment independently or authoritatively on UK shale development" is an obvious attempt to smear both the competency and professional integrity of the entirety of the UK earth science academia. I ask the editors of Solid Earth to consider whether the EGU is serving its community effectively by hosting such invective.

Section 6.4: The author discusses UK shale gas regulations with respect to the the Preese Hall well, drilled in 2011. However, the regulatory landscape has changed

substantially since then. Most notably, the Infrastructure Act (2015) has imposed numerous additional regulations on the industry. By failing to mention these additional rules, this paper paints an inaccurate picture of present regulations (available online at: http://www.legislation.gov.uk/ukpga/2015/7/section/50/enacted).

Acknowledgements: Smythe declares no funding for this research from competing interests or external agencies. However, parts of this paper bear a very close resemblance to a Planning Enquiry submission made to Notts County Council by Smythe on behalf of Bassetlaw Against Fracking (Smythe, 2015), an anti-fracking activist organisation. Based on crowdfunding websites set up to fund this work, it would appear that he was paid nearly £,000 by this organisation for at least a portion of the work presented here. This funding should be declared in the publication.

References: Chambers K., Kendall J-M., Barkved O., 2010. Investigation of induced microseismicity at Valhall using the Life of Field seismic array: The Leading Edge 29, 290-295.

Flewelling S.A., Tymchak M.P., Warpinski N., 2013. Hydraulic fracturing height limits and fault interactions in tight oil and gas formations: Geophysical Research Letters 40, 1-5.

Hammack R., Harbert W., Sharma S., Stewart B., Capo R., Wall A., Wells A., Diehl R., Blaushild D., Sams J., Veloski G., 2014. An Evaluation of Fracture Growth and Gas/Fluid Migration as Horizontal Marcellus Shale Gas Wells are Hydraulically Fractured in Greene County, Pennsylvania: EPAct Technical Report Series, U.S. Department of Energy, National Energy Technology Laboratory, Pittsburgh, PA, NETL-TRS-3-2014.

Maxwell S.C., 2011. Hydraulic fracture height growth: Canadian Society of Exploration Geophysicists, Recorder 36(9), 18-22.

Maxwell S.C., Shemeta J., Campbell E., Quirk D., 2008. Microseismic deformation rate

monitoring: SPE Annual Technical Conference, SPE116596.

Maxwell S.C., Jones M., Parker R., Miong S., Leaney S., Dorval D., D'Amico D., Logel J., Anderson E., Hammermaster K., 2009. Fault activation during hydraulic fracturing: SEG Annual Meeting, Expanded Abstracts.

Montgomery C.T., and Smith M.B., 2010. Hydraulic fracturing: history of an enduring technology: Journal of Petroleum Technology, December 2010 Issue, 26-32.

O'Toole T., Verdon J.P., Woodhouse J.H., Kendall J-M., 2013. Induced seismicity at Preese Hall – A review: EAGE Annual Meeting, Expanded Abstracts.

Schrama E., Naughton-Rumbo R., van der Bas F., Shaoul J., Norris M., 2012. Tight gas horizontal well fracturing in the North Sea: Offshore Engineer, October 2012, 45-48.

Smythe D.K., 2015. Planning application no. ES/3379 by Island Gas Limited to drill at Springs Road, Misson, Nottinghamshire: Objection on grounds of geology and hydrogeology. Downloaded from http://bassetlawagainstfracking.co.uk/wp-content/uploads/2015/12/DAVID-SMYTHE-REPORT-Dec1st-2015.pdf on 12.2.2016.

Stanchits S., Surdi A., Edelman E., Suarez-Rivera R., 2012. Acoustic emission and ultrasonic transmission monitoring of hydraulic fracture initiation and growth in rock samples: 30th European Conference on Acoustic Emission Testing, University of Granada.

Verdon J.P. and Kendall J-M., 2015. Response to Call For Evidence on the Environmental Risks of Fracking from the Commons Select Environmental Audit Committee. Downloaded from http://data.parliament.uk/writtenevidence/committeeevidence.svc/evidencedocument/environmental-audit-committee/environmental-risks-of-fracking/written/17012.pdf on 12.2.2016.

Wessels S.A., De La Pena A., Kratz M., Williams-Stroud S., Jbeili T., 2011. Identifying faults and fractures in unconventional reservoirs through microseismic monitoring: First Break 29, 99-104.

Westaway R., 2016. The importance of characterizing uncertainty in controversial geoscience applications: induced seismicity associated with hydraulic fracturing for shale gas in northwest England: Proceedings of the Geologists' Association, in press.

Younger P.L. and Westaway R., 2014. Review of the inputs of Professor David Smythe in relation to planning applications for shale gas development in Lancashire (planning applications LCC/2014/0096 /0097 /0101 and /0102) and associated recommendations. Downloaded from http://eprints.gla.ac.uk/108343/1/108343.pdf on 12.2.2016.

---

## Editor Comment (EC2) · F. Rossetti (Editor) · 19 Feb 2016

Removal of comments by Dr. Engelder (SC3) and Dr. Younger (SC5) was due to minor typos which were corrected. Updated versions are re-posted as SC4 and SC6, respectively.

Federico Rossetti

---

## Short Comment (SC9) · 2 Mar 2016

Reply to "Hydraulic fracturing in thick shale basins: problems in identifying faults in the Bowland and Weald Basins, UK"

I would like to address three specific issues in response to points made by Smythe (2016)

A. The position of the fault with respect to the Preese Hall-1 well bore

B. The origin of the casing deformation at Preese hall-1

C. Balcombe-2 faulting

[Figure]

A. Smythe (2016) argues that the faulthich appears to be responsible for the tremors at Preese Hall-1, actually crosses the well bore. This so-called 're-interpretation' does not use the 3D seismic data or well bore image data, but merely re-positions the trace of the fault on Figure 4 from the Clarke et al. (2014). One could have the impression that this is merely a device in order to provide a platform to expound his views regarding the hydrogeological risks of hydraulic fracturing in the Bowland Basin. Our contention is that, in fact, all of the evidence collected to date supports the observation that the wellbore was within 300m of a fault but does not intersect it (Clarke et al. op cit). The evidence is as follows:

1) The stratigraphy encountered within the Preese Hall wellbore correlates near to identically with both the Thistleton-1 and Grange Hill-1 wellbores. This applies both to the detailed wireline log correlation between these wells but also the specific ammonoid biozone have been identified over the crucial interval. These observations indicate that there is no missing or repeated sections. All three wellbores correlate strongly with each other in the shale section and have no signs of significant faulting, see figure 1. 2) The image log collected within the Preese Hall wellbore has no indications of faulting at the depth Smythe suggests, (de Pater, C. J., and S. Baisch, 2011). This image log is of high quality and resolution. It provides the most direct evidence of what faulting actually occurs within the wellbore. This poses the question of why such data, which is publically available, and the evidence it provides has been omitted from this submitted article. 3) In considering the fault interpreted by Smythe that has been redrafted on the original 3D seismic image Smythe states that this is a consistent fault interpretation with the hypocentre focal mechanism provided in the Clarke et al paper. While the uncertainty of the inverted source mechanism is significant, low dipping planes are inconsistent with observed amplitudes as illustrated in Appendix S2 of Clarke et al. When considering the Smythe fault plane one might assume the azimuth of his redrafted fault to be consistent with the focal mechanism, the dip of the fault interpreted by Smythe is approximately 30o from horizontal, but Appendix S2 of Clarke et al shows that a fault interpretation with given dip would result in larger misfit

than the 60-70o dipping planes. While it is certainly possible to reconsider picks of P- or S-waves (as suggested by in SC2 of Westway) one must bear in mind that picking is subjective and unless picked very unusually the new picks are unlikely to result in low dipping plane fitting the observed data. So they are in fact not compatible observations and disprove the redrafting interpretation. 4) It is also the case that in the current strike slip stress environment the fault Smythe has redrafted would have a much reduced slip tendency and considered less critically stressed than the Clarke et al interpretation and therefore not as likely, if at all possible to have failed given the Preese Hall 1 stimulation operations, in the current stress regime. 5) Westway (SC2, 2016) suggest alternative location of the hypocentre resulting from a velocity model. As pointed out by Verdon (2016, SC7) there is a large uncertainty in the location. Considering that Clarke et al estimated uncertainty relative to the velocity model 150m and 250m in horizontal and vertical directions, respectively, the differences in locations seem to be within these uncertainties. To conclude, the location of the weak aftershock may fit spatially within the uncertainty of a shallow dipping plane, but the observed amplitude do not.

The obvious question that should be posed when assuming a fault intersects the well bore would be why it is not the case that the hypocenteral location for the seismic event does not occur also where Smythe proposes a fault intersects the wellbore. This is to say if the fault were to intersect the Preese Hall wellbore, it would be the point that would experience the highest fluid volume entry, and experience the largest reduction in effective normal stress. Therefore being the most obvious point to have failed and provide a hypocentral location at the wellbore, not c.300m away as observed. Instead the hypocentre is distant from the wellbore and consistent with the fault interpretation provided in the Clarke et al 2014 paper. For these reasons the redrafted fault provided by Smythe should be ignored and the original Clarke et al fault consider the most appropriate interpretation at this point in time.

B. With regard to the casing deformation which Smythe attempts to use as an argument for a fault crossing the wellbore, a more robust mechanism for this deformation

is bedding parallel slip as outlined in the geomechanical study of Preese Hall 1, (de Pater, C. J., and S. Baisch, 2011). It should be noted that the deformation occurred over a section of the wellbore and not in a unique plane as would be expected if it was caused by a unique plane such as a fault (T. Keiser, 2014). Wellbore deformation over a broad section is known and documented to have been caused by bedding parallel slip globally (M. Dusseaul et al, 2001) and does not require seismic activity to be associated. The author should refer to why his argument is counter to that of currently held and accepted understanding of such observations and mechanical causes.

C. Considering figure 8b, for which I cannot find a reference, Smythe has misinterpreted the formation evaluation log from Balcombe 2. Central to his argument is the interpretation of this log showing a normal fault with downthrown to the east and using this interpretation to strongly suggest a hydrogeological risk. However the Smythe interpretation has no supportive evidence and indicates poor understanding of drilling processes. The apparent repeat section referred to in figure 8b is due to drilling out the cement shoe following the installation of the liner in the top of I micrite (reference Cuadrilla planning application). As this cement shoe is drilled out into formation, there is increasingly less cement returned and increasingly more formation cuttings returned until drilling of the shoe is complete and 100% of formation is returned via the drill bit. This is a fundamental drilling observation and provides no evidence towards any fault. As such the Smythe interpretation should be rejected. It should also be noted for completeness that the bedding dip is not zero but 3 degrees from horizontal which is available from information in the Balcombe 1 wellbore.

Smythe refers to his own blog written about the Cuadrilla Balcombe operation here (www.davidsmythe.org/fracking/cuadrilla%20sussex%20critique%20V2.0.pdf) which he uses as evidence to support that his conclusions are true to his predictions with regard to fault interpretations. However along with the previous unequivocal argument regarding drilling returns and there not being a fault, other predictions made by Smythe within this document are also contradicted by this submitted publication. D. The central

theme of the Smythe manuscript proposes that faults intersected by hydraulic fractures may form conduits through which fluids can potentially pollute ground water. He also argues that monitoring of fracture-fault interactions in not possible through seismic monitoring. Frieberg et al. (2014) and BGOC (2012) show detailed studies in Ohio, USA and Bristish Colombia, Canada, where hydraulic fractures intersected pre-existing faults and induced seismicity below the injection intervals. Such observations are consistent with the location of seismicity below injection at Preese Hall as published by Clarke et al (2014) and suggest that when hydraulic fractures intersects pre-existing fault sthe fluids penetrate to greater depths. Futrthermore, Zoback (2007) shows that fluid conductive faults are usually faults favourably stressed for shear failure. Such faults when lubricated often create shear events which are detectable by seismic monitoring. Hence, the proposed seismic monitoring prevents not only the induced seismicity but also fracturing into large pre-existing faults and is considered adequate and best practice. Comments regarding operator competence would be more appropriately directed to the regulator of these activities. The opinion Smythe states on the regulatory system in the UK is also counter to that which is widely held; that UK oil and gas regulation is viewed as a global exempla (Royal Society, 2012).

References: Clarke, H., Eisner, L., Styles, P., and Turner, P.: Felt seismicity associated with shale gas hydraulic fracturing: the first documented example in Europe, Geophys. Res. Lett. 41(23), 8308-8314, doi:10.1002/2014GL062047, 2014.

de Pater, C. J., and S. Baisch (2011), Geomechanical study of Bowland Shale seismicity, Synthesis Report, Cuadrilla Resources Ltd., Lancashire, U. K., (2 November 2011). http://www.cuadrillaresources.com/wp-content/uploads/2012/02/Geomechanical-Study-of- Bowland-Shale-Seismicity_02-11-11.pdf.]

British Colombia Oil and Gas Commission (BCOGC) (2012). Investigation of observed Seismicity in the Horn River Basin, British Colombia Oil and Gas Commission Report, 29 pp.

Cuadrilla Balcombe Limited: Lower Stumble Hydrocarbon Exploration Site. Cuadrilla Resources Limited Planning Application to West Sussex County Council , Appendix A: Geological summary, log and cross-section. http://buildings.westsussex.gov.uk/ePlanningOPS/tabPage3.jsp?aplId=1634 , 2013

Friberg P.A., G.M. Besana-Ostman, and I. Dricker, 2014, Characterization of an Earthquake Sequence Triggered by Hydraulic Fracturing in Harrison County, Ohio, Seismological Research Letters, 85 (6), 1295-1307, DOI 10.1785/0220140127

Trent Kaiser 2014, Wellbore Deformation in Unconventional Resources, SPE (http://www.spe.org/dl/docs/2014/Kaiser.pdf)

Maurice B. Dusseaul et al, 2001, Casing Shear: Causes, Cases, Cures. SPE (http://www.advancedgeomechanics.com/article/spe72060.pdf)

Royal society, 2012. Shale gas extraction in the UK: a review of hydraulic fracturing. (https://royalsociety.org/~/media/policy/projects/shale-gas-extraction/2012-06-28-shale-gas.pdf)

Zoback, M. D. [2007] Reservoir geomechanics: Cambridge University Press

[Figure]

[Figure]

**Fig. 1.**

---

## Short Comment (SC10) · 4 Mar 2016

The latest comment on the Preese Hall saga, by Clarke (2016), raises a number of issues that have not previously been covered in this thread.

The first of these concerns uncertainties in the geometry of the induced seismicity and the related seismogenic fault. Clarke (2016) is correct to state that one expects location of any microearthquake using only a small number of seismograph stations to be subject to considerable uncertainty. However, a key issue, which I attempted to convey both in my recent publication (Westaway, 2016a) and in my previous commentary (Westaway, 2016b), is that in addition to the forms of uncertainty that one expects in any microseismic study, the Clarke et al. (2014) analysis included some pretty funda-

mental errors. Other studies that have managed to avoid such mistakes, such as my own, are therefore inherently more likely to yield correct results.

Second, the geological structure at Preese Hall-1 and neighbouring boreholes, presented in Fig. 1 of Clarke (2016), differs dramatically from that which has featured in all previous literature on this topic, including the illustrations in my own recent outputs (Westaway, 2015, 2016a, 2016b). He now reports that the Emstites leion (Cravenoceras leion, or E1a1) marine band, which defines the boundary between the Visean and Namurian stages of the Carboniferous, at ∼2500 m depth (MD) in the Preese Hall-1 borehole. This means, essentially, that the part of the sedimentary succession that was previously reported (e.g., in my publications) as the Bowland Shale Formation has been reinterpreted as merely the 'Upper Bowland Shale' (i.e., the upper, or Namurian, part of the Bowland Shale Formation). The rocks penetrated at greater depths (previously interpreted as the Pendleside Limestone, Hodder Mudstone, and Clitheroe Limestone formations) have thus been reinterpreted as the upper part of the 'Lower Bowland Shale' (i.e., the upper part of the lower, or Visean, part of the Bowland Shale Formation). In this revised scheme, the complexity in the gamma ray and sonic log records, reported by Clarke (2016), indicates alternations between mudstone-dominated and limestone-dominated bedding, as was previously documented, but the limestone thus revealed is internal to the Bowland Shale Formation and not indicative of other formations. If this substantial reinterpretation (the basis of which has not been explained, as far as I am aware) is correct, it means that the deepest frack stages of the Preese Hall-1 well and the associated wellbore deformation were in the upper part of the 'Lower Bowland Shale', rather than in the Hodder Mudstone Formation as was previously thought. This change does not affect the conclusions reached in earlier publications (including mine), although it means that the labelling of many figures (including mine) is incorrect. It is nonetheless extraordinary for such an important revision to stratigraphy to be published (apparently in the first instance, as no reference is cited) as part of a commentary on another paper rather than as a publication in its own right.

Third, the excerpt from the 3-D seismic section that was originally published by Clarke et al. (2014) and was re-published by Westaway (2016a, 2016b) with stratigraphic labelling, shows the component of section-parallel bedding dip changing downward from westward, across a zone of deformed bedding, to eastward in the deepest ~200 m of the Preese Hall-1 well. In contrast, measurements made from the borehole image log, reported by Harper (2011), indicate that the bedding in this depth interval dips WNW at ~30-40°, steepening downward to ~70-80°. These two forms of evidence pertaining to the bedding are inconsistent. One possible explanation, tentatively raised by Westaway (2016a) on other grounds, is that Clarke et al. (2014) did not draw the well track on the seismic section in the correct place. However, at no point on this seismic section does the bedding appear steeper than ~30° in any direction, raising the alternative possibility that the image log has yielded incorrect information. This aspect requires resolution. As Westaway (2016b) noted, this 3-D seismic reflection dataset remains unpublished, except for the excerpt reported by Clarke et al. (2014). In the circumstances it is not helpful for Clarke (2016) to criticise Smythe (2016) for not using this 3-D seismic dataset, to which (like me) he has no access. Given the necessity for the British public and the UK scientific community to develop confidence that potential environmental issues relating to shale gas (such as induced seismicity and wellbore deformation) are understood, timely publication of this and the various other essential datasets, relating to the Preese Hall-1 well, which are not yet in the public domain, is strongly recommended.

Fourth, Clarke (2016) criticizes the Smythe (2016) interpretation that the 2011 seismogenic fault cut across the Preese Hall-1 borehole, accounting for the observed wellbore deformation. However, as previously discussed (Westaway, 2016b), this struck me as pretty much the most useful aspect of the Smythe (2016) contribution, since it confirms – in general terms – my own stated view; the principal problem with it being that it was based on an incorrect geometry of the fault, having been confounded by some of the mistakes in the original Clarke et al. (2014) publication. From de Pater and Baisch (2011) and Harper (2011), the wellbore deformation was concentrated

between depths of ∼8500 and ∼8650 feet (MD) (∼2591-2637 m MD), equivalent to ∼2540-2590 m (TVD). This is close to what was previously interpreted as the top of the Hodder Mudstone Formation, just below the base of the Pendleside Limestone Formation, and is now regarded as near the top of the Lower Bowland Shale, not far below the aforementioned Emstites leion marine band. De Pater and Baisch (2011) reported that the bedding in this vicinity dips WNW at ∼30° and the wellbore deformation involved strike-parallel shearing, at an azimuth of ∼N30°E, but due to the ambiguity inherent in such measurements (using a multi-fingered caliper tool) had no means of resolving whether the sense of shear was top-to-the-NNE or top-to-the-SSW. This part of the stratigraphic succession consists of interbedded mudstones and limestones; the correlation between the gamma ray log and the wellbore deformation indicates that slip occurred on bedding planes at changes in lithology (cf. de Pater and Baisch, 2011; Harper, 2011). Clarke (2016) is correct to note this interpretation of bedding plane slip, but it does not mean that the slip was not caused by the induced seismicity; the Dusseault et al. (2001) reference cited by Clarke (2016) indeed includes examples of bedding plane slip caused by seismicity. Figure 1 indicates schematically in cross-section how this bedding plane slip might have linked to the coseismic faulting; the inferred geometry resembles a conventional 'horsetail splay' (e.g., Sylvester, 1988) or 'contractional imbricate fan' (e.g., Woodcock and Fischer, 1986) fault termination. From consideration of the magnitude (2.3) and, thus, seismic moment, of the largest Preese Hall induced earthquake, a coseismic displacement of ∼10 mm can be estimated (cf. Westaway and Younger, 2014); it is envisaged that, beyond the up-dip limit of this fault, this shear displacement would have been partitioned across the various planes of weakness within the deformed zone. One reason why this wellbore deformation has not hitherto been associated with the induced seismicity was that Clarke et al. (2014) located the seismicity so far from the wellbore. This argument was superseded by the realisation of the mistakes in their paper and the resulting adjustment of the position of the seismogenic fault much closer to the wellbore (Westaway, 2016a). Another reason has been because de Pater and Baisch (2011) were unable to identify a clear conceptual link between the seismogenic fault and the wellbore deformation. Since such a link is now evident (Fig. 1), this issue warrants further attention. In the meantime, I note in passing that Clarke's (2016) comment that 'all of the evidence collected to date supports the observation that the wellbore was within 300 m of a fault but does not intersect it' is not in fact correct. On the contrary, it would appear that the wellbore intersected the 'horsetail splay' or 'contractional imbricate fan' at the up-dip termination of this fault (Fig. 1); by most definitions this is regarded as part of the fault.

.

References

Clarke, H., 2016. Reply to "Hydraulic fracturing in thick shale basins: problems in identifying faults in the Bowland and Weald Basins, UK" by D.K. Smythe. Interactive Discussion item SC9, 7 pp. Available online: http://www.solid-earth-discuss.net/se-2015-134/#discussion (accessed 3 March 2016)

Clarke, H., Eisner, L., Styles, P., Turner, P., 2014. Felt seismicity associated with shale gas hydraulic fracturing: The first documented example in Europe. Geophysical Research Letters, 41, 8308–8314.

de Pater, C.J., Baisch, S., 2011. Geomechanical study of Bowland Shale seismicity: synthesis report. Cuadrilla Resources Ltd., Lichfield, 71 pp. Available online: http://www.rijksoverheid.nl/bestanden/documenten-en-publicaties/rapporten/2011/11/04/rapport-geomechanical-study-of-bowland-shale-seismicity/rapport-geomechanical-study-of-bowland-shale-seismicity.pdf (accessed 3 March 2016)

Dusseault, M.B., Bruno, M.S., Barrera, J., 2001. Casing shear: causes, cases, cures. SPE paper 72060. SPE Drilling & Completion Journal, 16 (2), 98-107.

Fisher, K., Warpinski, N., 2012. Hydraulic-fracture-height growth: Real data. SPE paper 145949. SPE Productions & Operations Journal, 27 (1), 8–19.

Harper, T.R., 2011. Well Preese Hall‐1: The mechanism of induced seismicity. Geosphere Ltd., Beaworthy, Devon, 67 pp. Available online: http://www.cuadrillaresources.com/wp-content/uploads/2012/06/Geosphere-Final-Report.pdf (accessed 3 March 2016)

Smythe, D.K., 2016. Hydraulic fracturing in thick shale basins: problems in identifying faults in the Bowland and Weald basins, UK. Solid Earth Discussion; doi: 10.5194/se-2015-134, 45 pp.

Sylvester, A.G., 1988. Strike-slip faults. GSA Bulletin, 100, 1666-1703.

Westaway, R., 2015. Induced Seismicity. In: Kaden, D., Rose, T.L. (eds.), Environmental and Health Issues in Unconventional Oil and Gas Development. Elsevier, Amsterdam, pp. 175-210.

Westaway R., 2016a. The importance of characterizing uncertainty in controversial geoscience applications: induced seismicity associated with hydraulic fracturing for shale gas in northwest England. Proceedings of the Geologists' Association. doi: 10.1016/j.pgeola.2015.11.011, 17 pp.

Westaway, R., 2016b. Comment on "Hydraulic fracturing in thick shale basins: problems in identifying faults in the Bowland and Weald basins, UK" by D.K. Smythe. Interactive Discussion item SC2, 8 pp. Available online: http://www.solid-earth-discuss.net/se-2015-134/#discussion (accessed 3 March 2016)

Westaway, R., Younger, P.L., 2014. Quantification of potential macroseismic effects of the induced seismicity that might result from hydraulic fracturing for shale gas exploitation in the UK. Quarterly Journal of Engineering Geology and Hydrogeology, 47, 333–350.

Woodcock, N.H., Fischer, M., 1986. Strike-slip duplexes. Journal of Structural Geology, 8, 725-735.

.

Figure 1. Schematic representation, not to scale, but representing vertical and horizontal distances of up to several hundred metres, of the geometry of faulting associated with the induced seismicity and related wellbore deformation at Preese Hall in 2011. This is a vertical cross-section oriented WNW-ESE depicting the Preese Hall-1 well track (brown), on which a tick marks the part of the casing that was perforated for frack stage 2 that led to the induced seismicity. Vertical blue line marks the geometry of the resulting induced fracture network, which developed in the plane perpendicular to the minimum principal stress and was thus vertical, with an azimuth circa N7°E S7°W (Westaway, 2016a). The induced fracture network is assumed to have developed mainly upwards, rather than downwards, from its point of initiation, as is expected if the pressure of the fracking fluid was only slightly above the minimum necessary for fracture initiation (e.g., Fisher and Warpinski, 2012; Westaway and Younger, 2014). Thick red line indicates the orientation of the fault that slipped in the induced seismicity, which had a focal mechanism with strike 030°, dip 75° and rake 20° according to Westaway (2016a). The predominant sense of slip on this fault plane was thus left-lateral, indicated with dot and cross symbols to denote motions in and out of the section plane, together with a minor component of normal slip, indicated by paired arrows. According to Westaway (2016a), the patch of fault that slipped was south of the section plane, where the fault (oriented perpendicular to this section) intersected the induced fracture network (oriented oblique to the section). From Westaway and Younger (2014), the magnitude and seismic moment of the largest induced earthquake indicate slip on a patch of fault with dimensions of ∼100 m, with up to ∼10 mm of slip. Curved black lines indicate schematically the 'horsetail splay' or 'contractional imbricate fan' at the up-dip termination of the fault, which is inferred on the basis of the 'bedding plane slip' that caused the deformation to the Preese Hall-1 wellbore. Given the ∼30° WNW dip of the bedding in this vicinity, according to de Pater and Baisch (2011) and Harper (2011), based on the borehole image log, this bedding is subperpendicular to the neighbouring steeply ESE-dipping part of the fault. The strike-slip component of motion on the steep part of the fault is thus accommodated by top-to-the-NNE bedding plane slip,

again represented by dot and cross symbols, whereas its normal component of slip is accommodated by contraction, perpendicular to the bedding planes, represented by chevron symbols. This schematic model provides a potential resolution, for the first time, to a significant conundrum relating to this instance of induced seismicity: once the induced fracture network had propagated upwards into the zone of weakness where the 'bedding plane slip' occurred, why did the pressure of the fracking fluid not simply force open these bedding planes and the fluid then leak along them into the adjoining steep part of the fault? The answer is that this effect of fluid pressure would have facilitated a component of dip slip on this steep part of the fault in the opposite sense to that observed – reverse slip on the steep, ESE-dipping part of the fault being compatible with tensile opening of the weak bedding planes – and so would have been opposed by the local stress field. In order to induce seismicity, the high-pressure fracking fluid had to enter the steep part of the fault directly, not via bedding planes, which was only possible at the intersection between the fault and the induced fracture network to the south of the borehole. Hence, the seismicity occurred in this more southerly location. This proposed configuration is also consistent, to first order, with the geometry of the fault as imaged on the 3-D seismic section (Fig. 4 of Westaway, 2016a; Fig. 1(b) of Westaway, 2016b), which indicates that the overall ∼100 m of normal slip on its steep ESE-dipping part is accommodated by ∼100 m of contraction in what was formerly regarded as the Hodder Mudstone Formation but is now considered to be the upper part of the 'Lower Bowland Shale' (although this seismic section does not show the 30° dip of the bedding apparent on the borehole image log). In detail, the geometry of the deformation is more complicated than is depicted here, because the bedding is not precisely perpendicular to the steep part of the fault, the two make an angle of ∼105° (∼75°+30°), so normal slip on the steep part of the fault will be accommodated by contraction and distributed simple shear across the bedding, rather than just contraction.
* * *
[Figure]

**WNW**    **ESE**

**Fig. 1.** Please refer to the main document for this Figure caption.

---

## Short Comment (SC11) · 4 Mar 2016

We want to voice our concerns about a number of issues with "Hydraulic fracturing in thick shale basins: problems in identifying faults in the Bowland and Weald Basins, UK" by David Smythe. Smythe seems to misunderstand a number of arguments from the literature (e.g. Fisher and Warpinski (2012); Llewellyn et al. (2015)). In their summary of numerical models of hydraulic fracturing fluid migration they focus much of their discussion on one of the first models (Myers, 2012) while offering little discussion of, or entirely neglecting subsequent models that account for shortcomings that were identified in the early model. The following paragraphs expand upon these misunderstandings, which should be addressed before the review of Smythe (2016) is complete.

1. Misunderstandings in the Literature:

There are two instances where Smythe seems to misunderstand arguments in the literature. The first, relatively minor, misunderstanding is about an argument in Fisher and Warpinski (2012). This misunderstanding reduces the evidence for Smythe's premise that faults can commonly communicate fluids from the depth of typical shale gas units to shallow drinking water aquifers. The second, more important, misunderstanding is about Llewellyn et al. (2015), which Smythe uses as a cornerstone of his paper to "prove" that hydraulic fracturing fluids have traveled via permeable faults from great depths to contaminate shallow drinking water aquifers.

1.1 Misunderstanding of Fisher and Warpinski (2012):

In the Introduction, Smythe (2016) writes of Fisher and Warpinski (2012), "The authors argue that if faults were conduits they would have leaked all the gas away by now. This is clearly false; the whole point of fracking is to release gas which is trapped and therefore unable to migrate." In this excerpt, Smythe has confused the order of the argument. Various publications (Fisher and Warpinski, 2012; Flewelling and Sharma, 2015) suggest that because there is still gas in the shale reservoirs there must not be highly permeable faults near or through the shale reservoirs. If highly permeable faults did exist, the hydrocarbons would have leaked out of the reservoir during the millions of years since the hydrocarbons were generated. Smythe has misunderstood the qualitative argument that the presence of hydrocarbons indicates that either there are no faults near the shale reservoir, or that if there are faults, they do not conduct fluids at a high rate.

Admittedly, recent modeling papers have shown that overpressure can exist near a permeable fault for longer than 300,000 years (Gassiat et al., 2013; Lefebvre, 2015), which counters the arguments of Fisher and Warpinski (2012) that the presence of gas within a reservoir indicates that there are no permeable/conductive faults nearby. But 300,000 years is a relatively short amount of time compared to the age of many source

rocks, and there was some decrease in overpressure during the course of the simulation (Gassiat et al., 2013). So perhaps it is better to think of the Fisher and Warpinski (2012) argument in terms of rates. For example, if there is a highly permeable fault near a source rock, then the gas in the source rock would leave at a fast enough rate that the source rock would not have a high gas concentration remaining today (assuming it has been longer than 300,000 years since hydrocarbon generation). On the other hand if there is a fault near a source rock that still has a high concentration of gas present, then we can conclude that the rate of gas leakage via the fault is extremely slow and the permeability of the fault is small. There is an extremely low chance that this low-permeability fault could transmit a large amount of moderately buoyant fracturing fluid over the course of tens to thousands of years since it transmitted a very small amount of highly buoyant gas over the course of hundreds of thousands to tens of millions of years. The general concept of the Fisher and Warpinski (2012) argument stands: if there is a high concentration of gas in a source rock, there probably is not a highly permeable fault nearby.

1.2 Misunderstanding of Llewellyn et al. (2015):

In Section 5.3, Smythe summarizes and discusses a paper in which a chemical 2-BE and organic unresolved complex mixtures (UCMs) are found in a drinking water aquifer overlying the Marcellus shale (Llewellyn et al., 2015). Smythe misrepresents the findings as an unambiguous example of hydraulic fracturing fluids migrating from the depth of the Marcellus shale and misunderstands that gas found in the aquifer could have come from a different source and location than the 2-BE and the UCMs.

In an attempt to interpret Llewellyn et al. (2015) as proof that fracturing fluids migrated from the depth of the Marcellus to the aquifer Smythe writes, "Birdsell et al. (2015) have recently stated, in their review of frack fluid migration, that Llewellyn et al. conclude that 'if fracturing fluid did contaminate the shallow aquifer, it is much more likely that the fluid came from a surface spill or from a shallow subsurface leak rather than from the Marcellus'. This statement is completely wrong and misleading. In fact Llewellyn

et al. conclude that the most likely pathway for the groundwater contamination is initial passage up the wells from the Marcellus, followed by lateral passage along bedding planes, inclined gently upwards to the south, and finally by travelling vertically upwards along bedrock joint planes and fractures. Overpressured gas well annuli are also implicated as a possible driving mechanism." The statement of Birdsell et al. (2015a) is not incorrect as evidenced by: (1) a direct quote from a co-author of Llewellyn et al. (2015) (SC4, Engelder comment on Smythe (2016)), and (2) the following excerpts from Llewellyn et al. (2015) (bold added for effect):

1) "If HVHF fluids did contaminate the water wells, **it would be surprising if such contamination were due to fluids returning upward from deep strata**, given that (i) this has never been reported (6), (ii) the time required to travel 2 km up from the Marcellus along natural fractures is likely to be thousands to millions of years (31), and (iii) Fig. 6 shows that the Cl:Br ratios in the drinking waters indicate the absence of salts that would be diagnostic of fluids from the Marcellus Shale (e.g., flowback/production waters). **The most likely way for HVHF fluids to contaminate the shallow aquifers would therefore be through surface spillage of HVHF fluids before injection or by shallow subsurface leakage during injection.**"

2) "The data released here **do not implicate upward flowing fluids along fractures from the target shale as the source of contaminants but rather implicate fluids flowing vertically along gas well boreholes and through intersecting shallow to intermediate flow paths** via bedrock fractures. Flow along such pathways is likely when fluids are driven by high annular gas pressure or possibly by high pressures during HVHF injection. Such shallow to intermediate depth contaminant flow paths are not limited to HVHF but rather have been previously observed with conventional oil and gas wells. As shale gas development expands worldwide, problems such as those that occurred in northeastern PA will only be avoided by using conservative well construction practices, such as intermediate casing strings, proper cementation, and mitigating overpressured gas well annuli."

In addition to misrepresenting the certainty that 2-BE and UCMs came directly from the Marcellus, we feel that David Smythe misunderstands the argument that stray gas could come from one source (e.g. the target formation or any intermediate gas-bearing formation that is intersected by a well with a poorly cemented annulus) while the UCM and 2-BE come from another source. Even if the source of the 2-BE and UCM is related to drilling, it does not necessarily implicate that hydraulic fracturing fluids came from the Marcellus shale. For example, one possible source of the UCM and 2-BE is suggested to be drilling fluids associated with well remediation at much shallower depths. Excerpts from Llewellyn et al. (2015) follow:

3) "It is possible that the **provenance of the UCM and 2-BE was different from that of the stray gas.** Indeed, the most reasonable explanation for the natural gas impacts to water wells is that gas migrated from Welles 3-2H or possibly from multiple gas wells drilled on the Welles 3–5 pads due to excessive annular pressures and lack of competent annular cement that allowed gas to move vertically upward along the wellbore and into shallow uncased portions of bedrock fractures, including an identified fault zone (Table S1, Fig. 1, and Figs. S9 and S10)."

4) "Notably, the Welles 1 gas well pad was the location of a drilling fluid pit leak in August 2009 (Table S1). Further, well construction issues required remedial efforts in the Welles 3–5 series gas wells. Therefore, drilling fluids used in their installation could reasonably account for the observed foam impacts to household Wells 1–6 (Fig. 1C). **Since 2-BE and the UCM were identified together, drilling fluids might be the source of both.**"

2. Numerical Modeling of Fracturing Fluid Migration:

Smythe discusses previous numerical modeling studies of fracturing fluid migration towards aquifers at length, but he neglects at least one important critique and one modeling study. One critique that applies to the modeling studies of fracturing fluid migration (Myers, 2012; Gassiat et al. 2013; Kissinger et al., 2013; Cai and Ofterdinger, 2014),

is the neglect of capillary imbibition (comment on Myers, 2012 by Saiers and Barth, 2012), which can sequester large volumes of fracturing fluids in the target formation so that they cannot migrate towards an overlying aquifer (Engelder, 2012; Engelder et al., 2014; Birdsell et al., 2015b). Reagan et al. (2015) accounts for imbibition, but their analysis focuses on gas migration rather than fracturing fluid migration.

Birdsell et al. (2015a) identify five stages of a well lifetime with respect to hydraulic fracturing: prior to drilling, injection of fracturing fluids, shut-in during which capillary imbibition can occur, production of hydrocarbons and other fluids, and after the well is plugged and abandoned. In addition to reviewing fracturing fluid migration, Birdsell et al. (2015a) set up numerical models that faithfully account for all five stages of the well lifetime while addressing the major concerns raised about the other numerical modeling studies (e.g. buoyancy due to salinity, capillary imbibition, injection and production, and overpressure). Their results are significantly different from previous numerical modeling studies that did not represent the combined influence of injection, imbibition, well suction, and buoyancy. For instance, well suction and capillary imbibition significantly reduce the amount of hydraulic fracturing fluid that can reach an overlying aquifer. Birdsell et al. (2015a) in fact considers a worst-case scenario of a high permeability overburden with a continuous fault or poorly cemented wellbore directly connecting the top of the shale to the bottom of the aquifer, but shows that adding well production and capillary imbibition can drastically reduce the amount of HF fluids reaching overlying aquifers. Birdsell et al. (2015a) also investigate the sensitivity of amount of fracturing fluids reaching an aquifer to a large number of parameters including fault/wellbore parameters, well suction, amount of overpressure, imbibition rate, relative permeability, overburden heterogeneity, and buoyancy. The Birdsell et al. (2015a) paper should be included in the discussion of modeling studies in Smythe (2016) Section 5 and Figure 9 since it represents the most current peer-reviewed modeling study of fracturing fluid migration and faithfully accounts for the five stages of a well lifetime and the dominant processes within each stage.

References:

Birdsell, D. T., H. Rajaram, D. Dempsey, and H. S. Viswanathan (2015a), Hydraulic fracturing fluid migration in the subsurface: A review and expanded modeling results, Water Resour. Res., 51,doi:10.1002/ 2015WR017810.

Birdsell, D. T., H. Rajaram, and G. Lackey (2015b), Imbibition of hydraulic fracturing fluids into partially saturated shale, Water Resour. Res., 51, 6787-6796, doi:10.1002/ 2015WR017621.

Cai, Z., and U. Ofterdinger (2014), Numerical assessment of potential impacts of hydraulically fractured Bowland shale on overlying aquifers, Water Resour. Res., 50, 6236–6259, doi:10.1002/2013WR014943.

Engelder, T. (2012), Capillary tension and imbition sequester frack fluid in Marcellus gas shale, Proc. Natl. Acad. Sci. U. S. A., 109(52), E3625–E3625, doi:10.1073/pnas.1216133110.

Engelder, T., L. M. Cathles, and L. T. Bryndzia (2014), The fate of residual treatment water in gas shale, J. Unconv. Oil Gas Resour., 7, 33–48, doi:10.1016/j.juogr.2014.03.002.

Fisher, M. K., and N. R. Warpinski (2012), Hydraulic-fracture-height growth: Real data, SPE Prod. Oper., 27(01), 8–19.

Flewelling, S. A., and M. Sharma (2015), Comment on "Hydraulic fracturing in faulted sedimentary basins: Numerical simulation of potential contamination of shallow aquifers over long time scales" by Claire Gassiat et al., Water Resour. Res., 51, 1872–1876, doi:10.1002/2014WR015904.

Gassiat, C., T. Gleeson, R. Lefebvre, and J. McKenzie (2013), Hydraulic fracturing in faulted sedimentary basins: Numerical simulation of potential contamination of shallow aquifers over long time scales, Water Resour. Res., 49, 8310–8327, doi: 10.1002/2013WR014287.

Kissinger, A., R. Helmig, A. Ebigbo, H. Class, T. Lange, M. Sauter, M. Heitfeld, J. Klunker, and W. Jahnke (2013), Hydraulic fracturing in unconventional gas reservoirs: Risks in the geological system, part 2, Environ. Earth Sci., 70(8), 3855–3873.

Lefebvre, R., T. Gleeson, J. M. McKenzie, and C. Gassiat (2015), Reply to comment by Flewelling and Sharma on "Hydraulic fracturing in faulted sedimentary basins: Numerical simulation of potential contamination of shallow aquifers over long time scales", Water Resour. Res., 51, 1877–1882, doi:10.1002/2014WR016698.

Llewellyn, G. T., F. Dorman, J. Westland, D. Yoxtheimer, P. Grieve, T. Sowers, E. Humston-Fulmer, and S. L. Brantley (2015), Evaluating a groundwater supply contamination incident attributed to Marcellus shale gas development, Proc. Natl. Acad. Sci. U. S. A., 112(20), 6325–6330, doi: 10.1073/pnas.1420279112.

Myers, T. (2012), Potential contaminant pathways from hydraulically fractured shale to aquifers, Ground Water, 50(6), 872–882, doi: 10.1111/j.1745-6584.2012.00933.x.

Reagan, M. T., G. J. Moridis, N. D. Keen, and J. N. Johnson (2015), Numerical simulation of the environmental impact of hydraulic fracturing of tight/shale gas reservoirs on near-surface groundwater: Background, base cases, shallow reservoirs, short-term gas, and water transport, Water Resour. Res., 51, 2543–2573, doi:10.1002/2014WR016086.

Saiers, J. E., and E. Barth (2012), Potential contaminant pathways from hydraulically fractured shale aquifers, Ground Water, 50(6), 826–828, doi:10.1111/j.17

---

## Author Comment (AC1) · 5 Mar 2016

This is a preliminary response to Dr Westaway. Substantial responses to him and to the several other commenters are in course of preparation. I take all comments seriously.

Dr Westaway complains that Mr Clarke, of the Lancashire exploration licensee Cuadrilla, appears to have presented a substantially revised stratigraphy of the Bowland Shale in the Preese Hall-1 well.

I took the trouble to pre-order the released well data from DECC's agents in early 2015, and obtained it on the day of release on 17 April 2015. Perhaps if Dr Westaway had taken the same care in sourcing his data, about which he has written and published

so extensively, he would not now be in the litho- and bio-stratigraphic mess he finds himself in.

As part of the well package, there is a 795-page final well report dating from August 2012. It includes a biostratigraphy report prepared for the operator by Dr Nick Riley (MBE), explaining the reasons behind the revised stratigraphy.

In my view there is no excuse for Dr Westaway to have submitted for publication any paper going into detail on Preese Hall-1 after mid April 2015, as he has done, without taking into account the substantial new body of information in the public domain represented by the released well data package.

---

## Short Comment (SC12) · 10 Mar 2016

I thank Smythe (2016a) for drawing my attention to the release in April 2015 of the Preese Hall-1 well log and associated stratigraphic report, in relation to the Westaway (2016a) online posting. I was already well aware of this release, however, although I was unaware (until I checked in the last few days) that the Cuadrilla Grange Hill-1 well log (from an adjacent site in Lancashire; likewise illustrated by Clarke, 2016) has also now been released (as of 9 February 2016, according to the well agents, TGS). The data sources for the illustrations presented by Clarke (2016) are, therefore, now clear; if this author had stated that these released documents (as opposed to other potential documents internal to Cuadrilla, possibly relating to their 3 D seismic survey, which

has not yet been released) were the sources of his illustrations, there would have been no need for further dialogue on this matter. The essential issue is that Clarke (2016) depicts the Bowland Shale Formation as extending to the base of the Preese Hall-1 well, at 2773 m MD, and depicts, at ∼2500 m MD, the Emstites leion Marine Band, which marks the Visean-Namurian boundary and thus separates the 'Lower Bowland Shale' from the 'Upper Bowland Shale'. In contrast, in the de Pater and Baisch (2011) stratigraphic scheme, adopted in the Westaway (2015, 2016b, 2016c) publications, the base of the Bowland Shale is at 2507 m MD, this formation being reportedly underlain by the Pendleside Limestone Formation and Hodder Mudstone Formation, with the well bottom supposedly in the Clitheroe Limestone Formation.

As regards the implications of these releases of data for my publications, I note that the Westaway (2015) paper was commissioned in December 2014 and the Westaway (2016b) paper was drafted in early March 2015, as is evident from the citation dates of the various online references. The point of my annotating the excerpt from the Cuadrilla 3 D seismic survey (after its publication by Clarke et al., 2014) was anyway to establish how it correlates with the various extant interpretations of the older 2-D seismic lines, notably the interpretation published by de Pater and Baisch (2011), which has informed most of the subsequent discussion regarding the Preese Hall-1 well. The aim of doing this was to show how several datasets (wellbore deformation, in situ stress measurements, seismic sections, induced seismicity, etc.) can be integrated, which no-one had done before. In order to be able to compare the old 2-D and new 3-D seismic sections they needed to be annotated consistently with each other, which was what I did. I suppose I could have also provided a second version of the 3-D seismic section, annotated consistent with the newly released stratigraphy, but (as Westaway, 2016a, already noted) that would merely have shown the whole of the lower part of the section labelled as 'Bowland Shale Formation', which would not have added anything. Since journal space is limited, the extra length that would have been added to my paper did not seem justified, not to mention the demands that production of an additional diagram would have made on my time. The essential point, as Westaway (2016a)

also noted, is that the Westaway (2016b, 2016c) interpretations have established a conceptual model for how the induced seismicity, wellbore deformation, and deformed stratigraphy are interrelated. I am sure that most people can see that it is immaterial to such an analysis what the rocks in which the wellbore deformation occurred happen to be called. Thus, rather than being a 'litho- and bio-stratigraphic mess', as Smythe (2016a) has claimed, the Westaway (2016b) analysis would appear to have something useful to say.

Smythe's (2016a) criticism of my work thus follows his now-familiar pattern, apparent from his (2016b) draft manuscript and many of his web pages (e.g., Smythe, 2014a, 2014b, 2014c, 2015), of claiming that research output that he does not 'like' is fundamentally flawed, when the issue at hand is insubstantive, often combining this process with personal attacks on authors; this is the flip side of his praise of outputs that contain fundamental errors, provided they support his agenda of opposing shale gas and fracking. Engelder (2016) referred to this tendency as 'advocacy-based science', whereas Westaway (2016c) called it 'selective citation of the literature'. A related issue is this author's lack of objectivity regarding the merits of his own contributions; for example, even though virtually every aspect of the Smythe (2016b) draft manuscript has been challenged by one or more subject specialists (not to mention earlier critiques, for example that by Younger and Westaway, 2014, of the Smythe, 2014c, document), one of his web pages (posted as Smythe, 2014a) continues to claim that 'no-one has ever challenged my findings in any detail. Instead, some pro-industry, pro-government geologists resort to ad hominem attacks, without even bothering to read what I have written.' I also note in passing that it is somewhat ironic for Smythe (2016a) to insist on citation of the new interpretation of the stratigraphy in the Preese Hall-1 well, when he has previously subjected its author, Dr Nick Riley, to particularly severe attacks; for example, Smythe (2015) describes his work in general as 'flawed' and 'misleading' and dismisses his 2012 report on Preese Hall-1, now released, as '...typical of the commercial links between the BGS and the oil industry', whereas Smythe (2014b) states 'Dr Riley seems to be blind to the obvious geological problems inherent in drilling through and adjacent to fault zones. He refuses to reveal his client list; my suspicion is that it includes companies currently prospecting in the UK for unconventional oil and gas.'

Furthermore, once the issue of inconsistent versions of the stratigraphy at Preese Hall-1 and in neighbouring localities had been raised by Smythe (2016a), I looked back through earlier outputs and discovered other inconsistencies in usage. The first publication incorporating any element of the 'new' stratigraphy was Andrews (2013): its page 30 states 'The Thistleton 1 well drilled 2911 ft (887 m) of the Bowland-Hodder unit, but terminated in Brigantian-aged shales and sandstones (N.J. Riley pers. comm.) and the lower part of the unit was not reached'; and its Fig. 28 shows the Bowland Shale Formation persisting to the well bottom, with the Visean-Namurian boundary (marked, as noted above, by the Emstites leion Marine Band) at ∼1540 m depth. Accompanying text states that an estimated 2800 feet or ∼850 m of the Bowland-Hodder unit remained undrilled, so its total thickness is locally ∼1740 m. In contrast, the seismic sections in Figs 7 and 8 of de Pater and Baisch (2011) show the well bottom in the Clitheroe Limestone Formation, it being so depicted in my recent publications (Westaway, 2015, 2016b, 2016c). However, the stratigraphic column in Fig. 4 of de Pater and Baisch (2011) depicts the well bottom in the Bowland Shale Formation, consistent with Andrews (2013) but inconsistent with the other illustrations by de Pater and Baisch (2011). I am not aware that this mismatch had previously been noted; I only just noticed it myself.

As regards the Preese Hall-1 well, I now also see that the depth of the base of the Bowland Shale Formation of 8225 feet MD or 2507 m MD in Fig. 3 of de Pater and Baisch (2011), which I have previously taken as definitive for use in my publications, is contradicted by the figure of 7460 feet MD or 2274 m MD in Fig. 4 of de Pater and Baisch (2011); both these values differ from the depiction in the 'new' stratigraphy of the Bowland Shale reaching the well bottom (Clarke, 2016), as already noted. The stratigraphic column in Fig. 3 of de Pater and Baisch (2011) indeed depicts the Bowland Shale between 6540 and 8225 feet or 1993 and 2507 m (MD), indicating a

thickness of ~514 m, with the 'Worston Shale Formation' (i.e., Hodder Mudstone Formation) between 8450 and 9004 feet or 2576 and 2744 m (MD), indicating a thickness of ~168 m. The combined thickness of these shale formations in this scheme is thus ~680 m, whereas in the 'new' interpretation, illustrated by Clarke (2016), the ~780 m drilled (between 1993 m MD and the well bottom at 2773 m MD) spans only part of the Bowland Shale Formation and did not reach the Hodder Mudstone Formation.

Similar discrepancies exist in other documentation, for example in documents submitted by Cuadrilla in support of their planning applications for shale gas developments in Lancashire (these development plans were rejected in 2015 but this decision is currently subject to appeal). Thus, for instance, at the proposed Preston New Road well site, Cuadrilla (2014a) reported the top and base of the 'Upper Bowland' at estimated depths (TVD) of 1350 and 1540 m, and the base of the 'Lower Bowland' at ~1930 m, indicating an overall thickness of ~580 m. On the other hand, Fig. 6 of Cuadrilla (2014b) depicts the top and base of the Upper Bowland Shale at estimated depths (TVD) of ~1550 and ~1960 m, and the base of the Lower Bowland Shale at ~2830 m (Fig. 1(a)), indicating an overall thickness of ~1280 m, more than double the other estimate. This seismic section also depicts almost 1000 m of the Hodder Mudstone Formation, the base of which is not shown, making the combined thickness of these two shale formations well in excess of 2000 m in this locality. The caption to this Figure states that 'the interpretation of the 3D geophysical (seismic) survey was made by Cuadrilla . . .', implying that it is based on the 3-D seismic reflection dataset that remains unpublished but is consistent with the 'new' stratigraphic interpretation that has now been released (cf. Clarke, 2016). I note in passing that Smythe (2014b) described the structural interpretation in Fig. 1(a), by Cuadrilla, as 'geologically improbable' and proposed the revised interpretation with more extensive faulting in Fig. 1(b), notwithstanding the fact that he has not seen the 3-D seismic reflection dataset on which the interpretation is based and the members of staff of Cuadrilla who produced the interpretation in Fig. 1(a) obviously had access to this dataset.

Finally, it goes without saying that the tone of Smythe's (2016a) comment was inappropriate for scholarly discourse, acceptable though this knockabout style might be for postings on his own website. Nonetheless, it would seem that this exchange has shed some light on the stratigraphy of the Preese Hall-1 well and its vicinity, so something useful has been accomplished.

.

References

Andrews, I.J., 2013. The Carboniferous Bowland Shale gas study: geology and resource estimation. British Geological Survey for Department of Energy and Climate Change, London, 56 pp.

Clarke, H., 2016. Reply to "Hydraulic fracturing in thick shale basins: problems in identifying faults in the Bowland and Weald Basins, UK". Interactive Discussion item SC9, 7 pp. Available online: http://www.solid-earth-discuss.net/se-2015-134/discussion (accessed 9 March 2016)

Clarke, H., Eisner, L., Styles, P., Turner, P., 2014. Felt seismicity associated with shale gas hydraulic fracturing: The first documented example in Europe. Geophysical Research Letters, 41, 8308–8314.

Cuadrilla, 2014a. Temporary Shale Gas Exploration, Preston New Road; Environmental Statement. Appendix B - Scheme Parameters. Cuadrilla Bowland Ltd, 14 pp. Available online: http://planningregister.lancashire.gov.uk/PlanAppDisp.aspx?recno=6586 (accessed 9 March 2016)

Cuadrilla, 2014b. Temporary Shale Gas Exploration, Preston New Road; Environmental Statement. Appendix L – Induced Seismicity. Cuadrilla Bowland Ltd, 133 pp. Available online: http://planningregister.lancashire.gov.uk/PlanAppDisp.aspx?recno=6586 (accessed 9 March 2016)

de Pater, C.J., Baisch, S., 2011. Geomechanical study of Bowland

Shale seismicity: synthesis report. Cuadrilla Resources Ltd., Lichfield, 71 pp. Available online: http://www.rijksoverheid.nl/bestanden/documenten-en-publicaties/rapporten/2011/11/04/rapport-geomechanical-study-of-bowland-shale-seismicity/rapport-geomechanical-study-of-bowland-shale-seismicity.pdf (accessed 9 March 2016)

Engelder, T., 2016. Advocacy-Based Science. Interactive Discussion item SC4, 11 pp. Available online: http://www.solid-earth-discuss.net/se-2015-134/discussion (accessed 9 March 2016)

Smythe, D.K., 2014a. Reputational smears in the UK gutter press. Available online: http://www.davidsmythe.org/professional/smears.html (accessed 9 March 2016)

Smythe, D.K., 2014b. Nick Riley starry-eyed about UK shale geology. Available online: http://www.davidsmythe.org/frackland/?p=61 (accessed 9 March 2016).

Smythe, D.K., 2014c. Planning application no. LCC/2014/0096 by Cuadrilla Bowland Limited to drill at Preston New Road, Lancashire: Objection on grounds of geology and hydrogeology. Available online: http://www.davidsmythe.org/fracking/Smythe Preston New Road objection v1.3.pdf (accessed 9 March 2016)

Smythe, D.K., 2015. The insolence of office. Available online: http://www.davidsmythe.org/professional/insolence.html (accessed 9 March 2016)

Smythe, D.K., 2016a. Failure by Dr Westaway to incorporate well data released in April 2015. Interactive Discussion item AC1, 2 pp. Available online: http://www.solid-earth-discuss.net/se-2015-134/discussion (accessed 9 March 2016)

Smythe, D.K., 2016b. Hydraulic fracturing in thick shale basins: problems in identifying faults in the Bowland and Weald basins, UK. Solid Earth Discussion; doi: 10.5194/se-2015-134, 45 pp.

Westaway, R., 2015. Induced Seismicity. In: Kaden, D., Rose, T.L. (eds.), Environmental and Health Issues in Unconventional Oil and Gas Development. Elsevier, Amsterdam, pp. 175-210.

Westaway, R., 2016a. Some additional thoughts on Preese Hall. Interactive Discussion item SC10, 9 pp. Available online: http://www.solid-earth-discuss.net/se-2015-134/discussion (accessed 9 March 2016)

Westaway R., 2016b. The importance of characterizing uncertainty in controversial geoscience applications: induced seismicity associated with hydraulic fracturing for shale gas in northwest England. Proceedings of the Geologists' Association. doi: 10.1016/j.pgeola.2015.11.011, 17 pp.

Westaway, R., 2016c. Comment on "Hydraulic fracturing in thick shale basins: problems in identifying faults in the Bowland and Weald basins, UK" by D.K. Smythe. Interactive Discussion item SC2, 8 pp. Available online: http://www.solid-earth-discuss.net/se-2015-134/discussion (accessed 9 March 2016)

Younger, P.L., Westaway, R., 2014. Review of the Inputs of Professor David Smythe in Relation to Planning Applications for Shale Gas Development in Lancashire (Planning Applications LCC/2014/0096 /0097 /0101 and /0102) and Associated Recommendations. Report to Lancashire County Council, 12 pp. + 1 p. preface. University of Glasgow; available online: http://eprints.gla.ac.uk/108343/ (accessed 9 March 2016)

.

Figure 1. (a) Structural cross-section through the proposed Preston New Road shale gas well site, interpreted by Cuadrilla based on their 3-D seismic reflection survey, showing the 'new' interpretation of the thickness of the Bowland Shale Formation and the extent of its disruption by localized faulting. (b) Alternative interpretation by Smythe (2014c), prepared without access to the underlying 3-D seismic reflection survey, interpreting more extensive faulting. On the basis of this new interpretation, this author states that 'the direct paths 1 and 4 to the surface in the vicinity of the drillsite (say within 5 km horizontally of the fracked zone below) will lead to potential contamination of the minor groundwater sources within the Quaternary, as well as of rivers and streams', thus presenting inferences from his reinterpretation as fact. Modified from Fig. 5.2 of Smythe (2014c); part (a) is based on Fig. 6 of Cuadrilla (2014b) with the precise location of the cross-section depicted in Fig. 7 of Cuadrilla (2014b).

[Figure]

[Figure]

**Fig. 1**. Please refer to the main document for this Figure caption.

---

## Author Comment (AC5) · 24 Mar 2016

<h1 style="text-align:center">Reply to Huw Clarke of Cuadrilla Resources Ltd [se-2015-134-SC9]</h1>

Mr Huw Clarke, Exploration Geologist of Cuadrilla Resources Ltd, has commented on my discussion paper (Smythe 2016) under the following headings, which all concern Cuadrilla's exploration in the UK:

A. The position of the fault with respect to the Preese Hall-1 wellbore.

B. The origin of the casing deformation at Preese Hall-1.

C. Balcombe-2 faulting.

D. Hydraulic fractures intersecting faults, regulation.

**A. The position of the fault with respect to the Preese Hall-1 (PH-1) wellbore**

Mr Clarke disputes my reinterpretation of the earthquake-triggered fault, stating that I merely repositioned the fault on Clarke et al.'s figure 4, so as to expound my views on the hydrogeological risks of fracking in the Bowland Basin. This is not the case; I repositioned the fault in relation to the Clarke et al. hypocentral location, so as to honour the small sample of 3D seismic data that has been published. Figure 1 illustrates this as objectively as possible.

I have digitally erased both the wellbore and the Clarke et al. version of the fault in Figure 1A, then illustrated in Figure 1B my line drawing of the seismic reflectors (which can be discerned in Fig. 1A). XX and YY denote the interpreted positions of the fault according to Clarke et al. and to myself, respectively. I lettered the Figure 1B in a very small font so as not to draw the eye unnecessarily. Both fault versions originate at the hypocentral location. The Clarke et al. version cuts across seismic layering, which is unacceptable as an interpretation, whereas my version takes the most feasible path between the major groups of reflectors.

Mr Clarke notes firstly that the stratigraphic columns of PH-1 and two other neighbouring wells match very well in the Lower Bowland Shale, in the area of interest of PH-1 around 8500-8700 feet. But these logs are only precise enough to rule out a very large normal or reverse fault with a throw of (say) a hundred metres or more. Furthermore, the identification of a repeat or missing section as a means of fault identification in a vertical well does not apply to strike-slip faults. The fault in question is of this type.

Secondly, Mr Clarke states that the image log for PH-1 is of high quality, shows no sign of faulting in the area of interest, and asks why I have not referred to these data. I have not referred to these image logs because, contrary to Mr Clarke's statement, they are not in the public domain. They do not form part of the well release package for PH-1 which I obtained in April 2015. The only

wellbore image data I am aware of is the example to illustrate borehole break-outs shown in figure 19 of de Pater and Baisch (2011), over the interval 4923-4930 feet within the Millstone Grit Group.

[Figure]

*Figure 1. A. Clarke et al. (2014, fig. 4) sample of east-west 3D seismic through the Preese Hall-1 well. B. Line drawing of reflectors with alternative fault positions noted. The semicircle on the lowermost blocked-out area is the upper half of the hypocentral location.*

The Lower Bowland Shale comprises *"Mudstone, dark grey to black, blocky or shaly, calcareous, pyritic, petroliferous, with subordinate interbedded limestones and sandstones. Limestones in the lower part especially include conglomerates and turbiditic debris beds."* (Hird and Clarke 2012); therefore I question whether image data of dark grey to black mudstones can reliably image fault zones. But if Mr Clarke can make the image data available for public inspection I will gladly reconsider my view.

I discuss the problem of the earthquake hypocentral location and the fault dip in my reply to Dr Westaway.

Mr Clarke raises an objection to my fault placement, which is that if the well does indeed cut the fault, then why was the hypocentre not at the wellbore rather than 300 m away? I have already answered this in my discussion section 3.6, referring to the recent experiment by Guglielmi et al.

(2015). In fact, the hypocentre is *expected* to be at some distance from the fluid injection zone, because of the interplay of fluid injection and aseismic slip, before seismic slip is finally triggered.

**B. The origin of the casing deformation at Preese Hall-1**

Mr Clarke prefers the mechanism of bedding parallel slip over an extended portion of the wellbore, to account for the casing deformation. Bedding parallel slip is indeed a documented mechanism for wellbore deformation, but so are faults, as described in both papers he cites (Dusseault et al. 2011, Kaiser 2014). He does not seem to have appreciated that my analysis postulating the fault zone cut by the wellbore implicates four or five narrow fault strands between 8500 and 8640 feet depth. The total fault zone width is either 4.7 m or 7.1 m, depending on which strands are identified. The fault zone occupies a long segment of wellbore because it intersects the latter at an acute angle of just 23°.

I conclude that Mr Clarke's final statement on this subject that my *"argument is counter to that of currently held and accepted understanding of such observations and mechanical causes"* is wrong; my explanation fits in very well with current understanding of wellbore deformation.

**C. Balcombe-2 faulting**

Mr Clarke discusses my Figure 8B, which was re-drawn from Cuadrilla Balcombe Ltd (2013), fig. A01. I stated in section 4.1:

> *"The lithology log shows a gradational change from 100% clay to 100% micrite over 55 m, but the gradation is repeated below 2700 ft, this time over about 34 m. It is possible, but unlikely, to explain the repetition by assuming that two separate logging runs were made and then poorly spliced together; but an alternative and more plausible explanation is that the wellbore went through a normal fault with a downthrow to the east (wellhead side)."*

I reproduce a detail from my Fig. 8B, with a scale of 10% increments in lithology added.

I preferred my second explanation (a normal fault) because I could hardly imagine that two separate logging runs could have been spliced together so poorly. I was, of course, aware that the repeat section coincided with the casing shoe set at exactly 2700 ft, and as can be seen I took care to include the shoe and liner in my diagram.

I agree with Mr Clarke's explanation that the repeat section is due to a drilling artefact, and is not evidence of a normal fault. But several questions remain, so I return to my separate logging run explanation.

Mr Clarke accuses me of a poor understanding of drilling processes, and explains that the drilling out of the cement shoe resulted in the apparent repeat of the Kimmeridge Clay to I-micrite transition. But his explanation fails to fit the facts. If the same wellbore was being drilled for the second time through the shoe there would be a mixture of cement and micrite in the cuttings, with the former decreasing from 100% as the latter increased. The cement is implied by the white area in Fig. 1 (it is a light grey in the original diagram, as also shown by the litholog higher up in the vertical portion of the well, where the 9-5/8 inch shoe was drilled through). But where has the clay below 2700 ft come from? At 2750 ft the litholog shows (as marked in percent in Figure 1) 30% micrite, 50% clay and 20% cement.

[Figure]

*Figure 1. Detail of Balcombe-2z lithology log (modified from Smythe 2016 fig. 8B) showing transition of clay to micrite in the obliquely drilled well. Depths on the left are driller's depths in feet, as measured along the deviated wellbore.*

Let us examine the depth log of the deviated well, also taken from Cuadrilla's fig. A01, reproduced with annotation in Figure 2. Here we see the cross-section accompanying the vertical section of Fig. 1. Referring to the lower part; the top I-micrite is taken as being at 2128 ft, as at Balcombe-1 (depths here are true vertical depth sub-sea). But there are two light blue lines indicating this

horizon running to the right. One intersects the wellbore at 2110 ft, and the other at 2138 ft., and separated by a horizontal distance of about 127 ft. The 2110 ft mark at the end of the upper blue line is the top micrite, which dips slightly to the left. The shoe, whose base is at 2138 ft, some 28 ft lower, was set where the proportion of micrite to clay was 90% (Fig. 1), and presumably increasing to 100% the farther (i.e. deeper) the wellbore was drilled. So the return to a high proportion of clay ( 40% micrite, 60% clay, discounting the cement proportion) is inexplicable.

[Figure]

*Figure 2. Cross-section of Balcombe-2z landing in the I-micrite (from Cuadrilla Balcombe Ltd 2014, fig. A01). Upper part shows the image with overlays added; lower part shows the overlays without the image, for clarity. Vertical scale is true vertical depth sub-sea. The section is compressed horizontally. The number 7 near the shoe symbol (two black triangles – see Fig. 1 above) indicates the width in inches.*

In conclusion, we have:

- Two drilling runs, of which the second was allegedly drilled straight through the shoe,

- Two conflicting blue lines indicating the top micrite in the cross-section of Figure 2, and

- An apparent repeat entry through the transition zone of clay to micrite (as measured from

drill cuttings).

It is possible that the two conflicting blue lines shown in Figure 1, both purporting to represent the top of the micrite, could be due to the second drilling run being sidetracked out of the first, to land at a slightly different angle in the micrite. But this is speculation, without enough facts to go on.

If Mr Clarke could provide a little more detail of the drilling activities, including, for example, the gamma ray logs over the interval in question (which could prove that the wellbore remained in 90-100% micrite from the shoe onwards), it would go a long way to clearing up these discrepancies. In the absence of such evidence I rest with my two alternative explanations of (a) a fault, or (b) poorly spliced logs, but shall now favour the latter.

It is noteworthy that Mr Clarke fails to provide a substantive comment on my interpretation of well logs showing that the Paddockhurst Park Fault was cut by Balcombe-2. I therefore must assume that he agrees with my interpretation.

**D. Hydraulic fractures intersecting faults, regulation**

I did not write that monitoring of fracture-fault interactions is *"not possible"*, contrary to what Mr Clarke asserts. I merely said that it was *"incomplete"*. However, I should modify my statement citing Pettitt et al. (2009), to read *"Microseismic activity does not necessarily record the passage of fracking fluid up a pre-existing fault."* [new word underlined].

Mr Clarke believes that UK oil and gas regulation is widely viewed as a global exemplar, quoting the Royal Society and Royal Academy of Engineering (2012). I refer him to my reply to Professor Younger, in which I discuss this unsatisfactory report in more detail.

**General conclusions**

The problems discussed here illustrates the insufficiency of information put into the public domain by Cuadrilla, the developer, even though the company has gone to the trouble of publishing some of its results in a peer-reviewed journal (Clarke et al. 2014). Dr Westaway [comment se-2015-134-SC2] and I are here in agreement, that we are trying to decipher elements of the geology, drilling and earthquake history based on fragmentary evidence and tiny diagrams, which is unsatisfactory. The 3D seismic will be released under UK licence regulations in 2017; I see no reason why it should not be released immediately for open study and scrutiny.

I propose to modify my discussion paper to take into account Mr Clarke's comments and my responses, where appropriate, and I thank him for taking the trouble to comment.

**References**

Clarke, H., Eisner, L., Styles, P., and Turner, P. 2014. Felt seismicity associated with shale gas hydraulic fracturing: the first documented example in Europe, Geophys. Res. Lett. 41(23), 8308-8314, doi:10.1002/2014GL062047.

Cuadrilla Balcombe Limited: Lower Stumble Hydrocarbon Exploration Site. Cuadrilla Resources Limited Planning Application to West Sussex County Council, Appendix A: Geological summary, log and cross-section. http://buildings.westsussex.gov.uk/ePlanningOPS/tabPage3.jsp?aplId=1634, 2013.

de Pater, C.J. and Baisch S. 2011. Geomechanical Study of Bowland Shale Seismicity, Synthesis Report for Cuadrilla Resources Limited. http://www.cuadrillaresources.com/wp-content/uploads/2012/02/Geomechanical-Study-of-Bowland-Shale-Seismicity_02-11-11.pdf.

Dusseault, M.B., Bruno, M.S. And Barrera, J. 2001, Casing shear: causes, cases, cures. Society of Petroleum Engineers, SPE 72060 (http://www.advancedgeomechanics.com/article/spe72060.pdf)

Guglielmi, Y., Cappa, F., Avouac, J.-P., Henry, P. and Elsworth, D. 2015. Seismicity triggered by fluid injection-induced aseismic slip, Science, 348(6240), 1224–1226, doi:10.1126/science.aab0476.

Hird, C. and Clarke, H. 2012. Preese Hall-1 end of well report LJ/06-5. Cuadrilla Resources Ltd. [included in well release package, in the public domain].

Kaiser, T. 2014. Wellbore deformation in unconventional resources. Society of Petroleum Engineers Distinguished Lecture Program, www.spe.org/dl/docs/2014/Kaiser.pdf.

Pettit, W., Reyes-Montes, J., Hemmings, B., and Hughes, E. 2009. Using continuous microseismic records for hydrofracture diagnostics and mechanics, Soc. Explor. Geophys. Ann. Meeting Expanded Abstracts.

Royal Society and Royal Academy of Engineering 2012. Shale gas extraction in the UK: a review of hydraulic fracturing, Issued: June 2012, DES2597, royalsociety.org/policy/projects/shale-gas-extraction.

Smythe, D.K., 2016. Hydraulic fracturing in thick shale basins: problems in identifying faults in the Bowland and Weald basins, UK. Solid Earth Discussion; doi: 10.5194/se-2015-134, 45 pp.

---

## Referee Comment (RC1) · A. Aplin (Referee) · 29 Mar 2016

**Smythe: Hydraulic fracturing in thick shale basins: problems in identifying faults in the Bowland and Weald Basins, UK**

I have chosen here to focus this review on the main aspects of the paper.

The title of the paper is promising, suggesting a focussed and scientific study of issues related to the identification and accurate location of faults in shale-rich sequences; this would have been relevant to discussions surrounding the safe and effective exploitation of shale gas. Indeed, the vertical migration of water, $CO_2$ and hydrocarbons is a subject of general interest to those interested in the rates and mechanisms by which fluids move in sedimentary basins.

Agreeing to review this paper, I had hoped that it would either be a scientific study or a review of the state of the art. It is neither, in that (a) no new data are presented with which to test a hypothesis and (b) the relevant literature is not reviewed in a helpful, insightful and unbiased way. It turns out that the title of the paper is wholly misleading; one part of it does indeed comment on the difficulties associated with the accurate location of faults using poor quality seismic data, but much is an invective-strewn commentary on other issues, loosely drawn together as a general discussion of potential shale gas exploitation in the UK.

The starting point of the paper is that in terms of risking faults as fluid conduits from target shales to surface, US shale gas experience cannot be extrapolated to the UK (and indeed Europe) since UK basins are extensional (but reactivated) basins, have thick shale sequences and are more likely to have (a) more faults generally and (b) more faults which cut from target shale to surface. This key assertion – which would be interesting if proven - is based on an unpublished desk study and no references are given. So, the underlying premise of the paper is missing.

Section 1.1 reviews "faulting in relation to fracking" (what does this mean – that faults result from hydraulic fracturing?) but primarily questions whether or not a key industry dataset on the locations of induced earthquakes is complete or has had "inconvenient results" omitted". The author does not present evidence for this and appears not to have looked at the supplementary literature which relates frack fluid volumes to the likely maximum height of hydraulic fractures. Nor does he comment on Hammack et al.'s (2014) field study of the fracture growth and fluid migration in the Marcellus– perhaps a study by the US DoE is less likely to have been redacted? Whilst none of these studies can conclusively prove that induced hydraulic fractures absolutely cannot penetrate from target shale to shallow aquifers, it is essential to present a balanced picture of what the currently amassed data indicate and what their true implications are. In the context of where this paper leads, which is the risk of fluid flow from shale to fault and then along faults, it would have been more useful to review the extent to which microseismic data reflect frack-related changes to shale structure and also how accurately they indicate changes to the small- to large-scale permeability structure of target horizons. Das and Zoback's (2013) work is also interesting in this respect, since microseismic data do not tell the whole story.

Section 2 is basically irrelevant to the paper and should be removed.

Section 3 discusses the Bowland Basin: its general structure, a very basic introduction to its stratigraphy, a review of available seismic data, its hydrogeology and then a reinterpretation of the

location of the fault along which an earthquake was triggered, relative to the position of the wellbore. Much of the background material is unfocussed and thus unhelpful. Going back to the title of the paper, very little information is given about (a) the distributions of faults which can be seen, (b) faults which are likely to be missing on a seismic dataset of a given vintage/quality and (c) those which are can never be seen on seismic. The geophysical community is well aware of the limitations of seismic data and there is a very significant literature on the relation between fault density and length/throw. Both these ideas are relevant to discussions about the safe and effective exploitation of shale resources, but could be condensed into a couple of sentences – or could be expanded into a sound and useful review. The work presented in this paper does neither effectively.

Section 3.2 concerns the hydrogeology of the region. However, it contains almost no hydrogeology (e.g. hydraulic head data, fluid flow measurements), just some rather random salinity data which alone are not relevant to discussions on regional fluid flow or the possibility for upwards fluid flow, which is a central issue in this paper.

The location of the key fault, its proximity to the drilled well and the relation to the induced earthquake is one of the stronger and more focussed parts of the paper (Sections 3.3-3.6). Nevertheless, the subject has been looked at previously and similar material has been published recently by e.g. Westaway (2015). The conclusion that faults can be difficult to observe in thick shale sequences may indeed be valid but does not derive from the analysis in this paper.

Section 4 concerns the identification of faults in the Weald Basin. Key conclusions are that even faults with significant throws may not be visible on old, low quality 2D seismic lines, but that they *can* be interpreted from detailed log and stratigraphic data. Sound - but hardly novel. Section 4.2, which relates to Celtique's drilling application, contains no useful or scientifically relevant material.

Section 5 concerns the potential for faults to act as sub-vertical conduits for fluids – frack fluids, gas or water. This is a topic which has exercised geoscientists for decades, as expressed in standard hydrogeology textbooks and in copious primary literature – which is not reviewed. In this paper, certain modelling studies have been selected to paint an overall picture that contamination of drinking water aquifers is likely as a result of flow along faults – see Table 2. A much more critical approach is needed and to a large extent this has already been covered in papers such as Birdsell et al. (2015), who not only reviewed previous work but also undertook a series of new 3D flow models incorporating robust estimates of permeability and a range of key processes including capillary imbibition. Birdsell et al.'s basic conclusion was that a "permeable pathway" from fracked zone to near-surface was needed in order for any gas/frack fluid to reach e.g. a drinking water aquifer. No surprise there but note that the permeable pathway in Birdsell et al's model extended continuously from fracked unit to surface and had a permeability throughout its length of $8 \times 10^{-11}$ m$^2$, i.e. that of a sand. What is the evidence that faults through shale-rich sequences have anything like such permeabilities?

Predicting the flow (rate and volume) of gas or frack fluid to and then along a fault requires a set of data which are difficult to obtain. Issues include: shale matrix (relative) permeability; imbibition of water; the permeability of fractured shale; the volume of fractured shale in hydraulic connection to the fault on the appropriate timescale; the permeability of the fault throughout its length (and how it changes if there is slip); the volume of fluid transmitted to the fault; overpressure prior to fracking;

pressure changes as a result of fluid injection. In a thoughtful review, all of this needs to be covered critically; most has been in e.g. Birdsell et al., but not in the current paper – a very fundamental flaw.

Other comments have covered the discussion regarding the drinking water contamination at Bradford County but to use it to support the idea that faults are important leakage routes is disingenuous. The concluding statement in Llewellyn is that "the data released here do not implicate upward flowing fluids along fractures from the target shale as the source of contaminants but rather implicate fluids flowing vertically along gas well boreholes and through intersecting shallow to intermediate flow paths via bedrock fractures". That is, an uncased borehole is implicated for most of the leakage pathway, and it is unclear whether frack fluids derived from the fracking process.

The final section relates to the regulatory regime in the UK. I doubt that this is of broad interest to the readership of Solid Earth Discussions, and I suggest that essentially all of this material is removed; some material comes over as rant, whereas a brief and reasoned comment on how fault properties might be reasonably assessed prior to drilling would be a more useful contribution.

Finally, there are a considerable number of personal and contentious comments regarding the integrity and understanding of both individuals and organisations. None are substantiated and all should be removed.

Andrew Aplin

---

## Short Comment (SC14) · 29 Mar 2016

R. Westaway

robert.westaway@gla.ac.uk

Smythe's (2016a) latest output, combining thuggery with erudition – hurling abuse at a distinguished colleague while simultaneously parading his knowledge of philosophy of science – reminded me of the scene in Monty Python's Life of Brian in which the Roman centurion threatens to cut the hero's throat with a sword because of his unfamiliarity with Latin grammar (Jones, 1979; YouTube, 2006). I doubt, when this classic satire was being produced, if anyone ever envisaged that life would one day so closely imitate art.

Smythe (2016a) has argued that Gödel's (1931) incompleteness theorems in mathematics have no relevance to physical science. In his view, the nature of scientific method was established by Popper, who pointed out that scientific hypotheses must be

testable, so are always open to the possibility of falsification and can therefore never be regarded as true as a matter of certainty. Gödel (1931) established two different though related incompleteness theorems, usually called his first and second incompleteness theorems, as Smythe (2016a) has stated. However, as Raatikainen (2015) has pointed out, the phrase 'Gödel's theorem' is routinely used to refer to the conjunction of these two theorems, but may refer to either—usually the first—individually. Smythe (2016a) was therefore incorrect to criticise Engelder (2016a) for using this expression.

I note in passing that Popper's great work on scientific method was published in 1934, not 1939 as Smythe (2016a) has stated. Had he delayed its publication until after the Nazi takeover of Austria in 1938 he would almost certainly – as a person of Jewish heritage – not have been permitted to publish it thereafter and would, thus, not have had a sufficient publication track record to obtain an academic job abroad in time to ensure his personal safety from the Holocaust. Furthermore, Smythe's (2016a) statement that 'if anyone can claim to be unbiased (but well informed) in the scientific debate about fracking it is I' conveys a degree of self-righteous certainty that does not fit well with Popper's (1934) deduction that one can never prove scientific results, only falsify them.

Gödel's (1931) first incompleteness theorem holds that, for any system of expressing arithmetic, not every statement that is true is provable within the system. Many workers (e.g., Charlesworth, 1981) have shown that this theorem is equivalent to a statement that no computer program can correctly determine whether it will eventually reach a result and halt, when run with a particular input, or whether it will run indefinitely, a method of proof first developed by Turing (1937). In principle, one might envisage writing a hypothetical computer program that attempts to calculate some particular numerical result by iteratively summing an ever-increasing number of algebraic terms until a particular level of accuracy has been achieved. However, because of its requirement for a deterministic halt condition, the existence of such a computer program is precluded by Gödel's (1931) first incompleteness theorem. Computer algorithms indeed tend to work differently, often determining results of calculations by carrying out a predetermined number of iterations that is sufficiently large for the estimated accuracy of the result to be as good as, or better than, one needs (e.g., better than the number of significant figures at which the resulting information is stored). An example is the CORDIC algorithm, originally developed for iteratively calculating trigonometric functions (as part of the first computer navigation system, used in the B-58 'Hustler' supersonic bomber, built by Convair for the U.S. Air Force) and nowadays widely used to calculate a range of mathematical functions (e.g., Volder, 1956, 1959, 2000; Meher et al., 2009), for example in math co-processors of PCs; it converges to solutions far more rapidly than would a standard power series solution. One is indeed used to the situation of computer calculations being to such high degrees of accuracy that the implications of Gödel's (1931) first incompleteness theorem are immaterial, extreme examples being provided by attempts to set world records (currently 13.3 trillion) for the number of decimal digits of accuracy for the value of pi (Yee, 2016). Enterprises of this type likewise utilize rapidly convergent iterative series solutions, in this case the 'Chudnovsky algorithm' (Chudnovsky and Chudnovsky, 1989).

Computer programs are of course widely used in the modelling of scientific data but not all the mathematical functions needed have analogously benefitted from inputs through large-scale funding by defence contractors or from expert mathematicians attempting world records; the accuracy available can be significantly limited. As a result, if one undertakes such a numerical prediction, and it does not agree with observation, one can have no way of knowing whether the hypothesis on which the calculation as based was incorrect, on the one hand, or whether the computer program did not lead to an accurate calculation, on the other. The first of these scenarios represents the case discussed by Popper (1934), of falsifying an existing hypothesis; the second scenario represents another form of ambiguity, not addressed by Popper (1934) (unsurprisingly, since his work pre-dated the use of computers in science), over which anyone involved in numerical modelling must always exercise great care. It follows that Popper's (1934) point, that one can never prove any scientific hypothesis but can only falsify it, is a

subset, within the domain of science undertaken through numerical modelling, of the fundamental ambiguity inherent in Gödel's (1931) first incompleteness theorem.

As a practical geophysical example, one might consider the flow of heat across the surface of a cylinder of a given radius, embedded in rock of uniform thermal properties, where this surface is held at a constant temperature that differs from the initial temperature of its surroundings. This problem was fist solved by Jaeger (1942); its solution depends on functions, now known as 'Jaeger integrals', that can be written as integrals of combinations of exponential and Bessel functions. Such mathematical functions are relevant to geothermics, relating for example to heat extraction using borehole heat exchangers, but also bear upon related problems in other areas of science, for example involving radial flow of fluids or electricity. Unfortunately, there is no simple way of evaluating Jaeger integrals; they can be approximated using one power series in the limit of short timescales and a different power series in the limit of long timescales (e.g., Carslaw and Jaeger, 1965, p. 336), but there is no known power series approximation that is valid at all time scales; computer calculations of these functions using any particular power series will therefore not necessarily converge with any degree of accuracy, providing an example of the issue recognized by Charlesworth (1981) as a demonstration of Gödel's (1931) first incompleteness theorem. It is, thus, difficult to test any hypothesis in any field of science that requires calculation using Jaeger integrals; as a result, efforts to develop calculation procedures that guarantee results with particular levels of accuracy continue to the present day (e.g., Peng et al., 2002; Britz et al., 2010; Phillips and Mahon, 2011). For example, the Phillips and Mahon (2011) approach does not guarantee accuracy to better than $\sim$0.2%, meaning that in some circumstances it only provides results accurate to two significant figures.

All this adds up to the recognition that one can never guarantee to prove any geological hypothesis, whether related to fracking or to any other branch of Earth science: for any given dataset there will always be observations that cannot be uniquely explained in terms of any particular hypothesis. As Engelder (2016a) has stated, one might

reduce the range of potential ambiguity by gathering more extensive datasets, including monitoring any site before fracking takes place, and one can also usefully discount hypotheses that incorporate demonstrably invalid assumptions.

Smythe's (2016a) contribution demonstrates familiarity with Popper's (1934) work (if not with its year of publication) and its implication of never being able to guarantee to prove any particular hypothesis. It should now be apparent how the ambiguity recognised by Popper (1934) relates, for practical purposes, to Gödel's (1931) first incompleteness theorem. On the basis of this undoubtedly extensive personal knowledge of the philosophy of science, and its implication that hypotheses can never be proved and there is always uncertainty in science, it is therefore inappropriate for this author to have argued, as he has done repeatedly (e.g., Smythe, 2014, 2016b), that shale developers should undertake exploration until the geology of their sites is known 'with certainty', and that the existence of any uncertainty is a basis for not proceeding with such projects.

References

Britz, D., Østerby, O., Strutwolf, J., 2010. Reference values of the chronoamperometric response at cylindrical and capped cylindrical electrodes. Electrochimica Acta, 55, 5629–5635.

Carslaw, H.S., Jaeger, J.C., 1965. Conduction of heat in solids. 2nd edition. Clarendon Press, Oxford.

Chudnovsky, D.V., Chudnovsky, G.V., 1989. The computation of classical constants. Proceedings of the National Academy of Sciences of the United States of America, 86 (21), 8178–8182.

Gödel, K., 1931. Über formal unentscheidbare Sätze der Principia Mathematica und verwandter Systeme I. Monatshefte für Mathematik und Physik, 38, 173-198. English translation (1934): On undecidable propositions of formal mathematical systems.

Notes on Lectures by Kurt Gödel. February-May, 1934. Princeton Institute for Advanced Study, Princeton, New Jersey, 30 pp.

Charlesworth, A., 1981. A proof of Gödel's theorem in terms of computer programs. Mathematics Magazine, 54 (3), 109–121.

Engelder, T., 2016. Advocacy-Based Science. Interactive Discussion item SC4, 11 pp. Available online: http://www.solid-earth-discuss.net/se-2015-134/discussion (accessed 27 March 2016)

Jaeger, J.C., 1942. Heat flow in the region bounded internally by a circular cylinder. Proceedings of the Royal Society of Edinburgh, Series A, 61, 223–228.

Jones, T., 1979. Monty Python's Life of Brian. HandMade Films Ltd., London.

Meher, P.K., Valls, J., Juang Tso-Bing, Sridharan, K., Maharatna, K., 2009. 50 years of CORDIC: Algorithms, architectures, and applications. IEEE Transactions on Circuits and Systems, 56 (9), 1893-1907.

Peng, H.Y., Yeh, H.D., Yang, S.Y., 2002. Improved numerical evaluation of the radial groundwater flow equation. Advances in Water Resources, 25, 663–675.

Phillips, W.R.C., Mahon, P.J., 2011. On approximations to a class of Jaeger integrals. Proceedings of the Royal Society, Series A, 467, 3570–3589.

Popper, K.R., 1934. Logik der Forschung. Zur Erkenntnistheorie der modernen Naturwissenschaft. Mohr Siebeck, Tübingen, Germany. English translation: Popper, K.R., 1959. The Logic of Scientific Discovery. Routledge, Abingdon.

Smythe, D.K., 2014. Planning application no. LCC/2014/0096 by Cuadrilla Bowland Limited to drill at Preston New Road, Lancashire: Objection on grounds of geology and hydrogeology. Available online: http://www.davidsmythe.org/fracking/Smythe Preston New Road objection v1.3.pdf (accessed 9 March 2016)

Smythe, D.K., 2016a. Conjecture and refutation; author's response to Dr Engelder.

Interactive Discussion item AC2, 7 pp. Available online: http://www.solid-earth-discuss.net/se-2015-134/discussion (accessed 27 March 2016)

Smythe, D.K., 2016b. Hydraulic fracturing in thick shale basins: problems in identifying faults in the Bowland and Weald basins, UK. Solid Earth Discussion; doi: 10.5194/se-2015-134, 45 pp.

Raatikainen, P., 2015. Gödel's Incompleteness Theorems. Stanford Encyclopedia of Philosophy. http://plato.stanford.edu/entries/goedel-incompleteness/ (accessed 27 March 2016)

Turing, A., 1937. On computable numbers, with an application to the Entscheidungsproblem. Proceedings of the London Mathematical Society, Series 2, 42, 230–265 (with 1938 correction: Proceedings of the London Mathematical Society, Series 2, 43, 544–546).

Volder, J.E., 1956. Binary computation algorithms for coordinate rotation and function generation. Convair Aeroelectronics Group internal report IAR-1.148. Convair Inc., San Diego, California.

Volder, J.E., 1959. The CORDIC Trigonometric Computing Technique. IRE Transactions on Electronic Computers, EC-8 (3), 330–334.

Volder, J.E., 2000. The birth of CORDIC. Journal of VLSI Signal Processing, 25, 101–105.

Yee, A.J., 2016. Y-cruncher - A multi-threaded pi-program. http://www.numberworld.org/y-cruncher/ (accessed 28 March 2016)

YouTube, 2006. Life of Brian - Romanes Eunt Domus. Available online: https://www.youtube.com/watch?v=IIAdHEwiAy8 (accessed 27 March 2016)

---

## Short Comment (SC15) · 29 Mar 2016

DISCUSSION OF FAULTS AND EARTHQUAKES IN THE VICINITY OF THE PREESE HALL WELL

The presence of faults within the Bowland Shale and overlying Carboniferous successions in the United Kingdom is not controversial. The Preese Hall well in the Flyde, Lancashire was subject to a number of small earthquakes in 2011 in the vicinity of the wells related to hydraulic fracturing ("fracking") operations, and deformation of the actual well bore. Having raised concerns about the accuracy and apparent ambiguity in locations between interpretations of the various faults in the vicinity of the Preese Hall well, the author proposes an alternative interpretation of the fault that moved causing these earthquakes. There are number of issues with the author's methodology for achieving this.

MANIPULATION OF FIGURE 4B

Smythe makes clear that his Figure 4B (originally from Clarke et al., 2014) has been digitally altered to facilitate his interpretation. However, by choosing to cover an arrow in the original figure with red a potentially misleading impression of the continuity of reflectors may occur. It is impossible to identify from this diagram as currently presented whether his preferred fault orientation is a valid interpretation or whether this is an artefact of the removal of the text box. In the form that this figure is currently presented, it is too ambiguous to support his hypothesis.

INTERPRETATION OF A FAULT CROSSING THE PREESE HALL WELL.

The author is highly critical of the interpretation of borehole deformation described in De Paiter and Baisch (2011) who noted that "the $5\frac{1}{2}$ in production casing was ovalized over a considerable distance of hundreds of ft. This ovalization is possibly related to the fault slip, but in view of the large interval of deformation it is most likely that the wellbore deformation is caused by shear slip on bedding planes, which is possibly associated with the fault slip." The author's alternative hypothesis is that this ovalisation was caused by intersection of the well bore with a fault at a previously unidentified orientation so that the fault was directly injected with the fracking fluids. Thereby he questions both the way that the original fracturing operation was conducted and also questions the subsequent investigation of the cause of the deformation. Smythe's interpretation of this presumed fault is based on two sources, conventional petrophysical data and the interpretation of ovalisation identified in the well by De Paiter and Baisch (2011), using their figure (shown in this paper as figure 6) as a study of borehole ovalisation. He then hypothesises that the maximum deformation zones are indicative of one or more fault strands intersecting with this wellbore in the interval 8500 to 8640'. Borehole imaging allows high resolution interpretation of the majority of the circumference of the borehole wall. This technique dates in primitive form (borehole camera and televiewers) from the 1960s and has been extensively applied to well interpretations and published on since. Resistivity borehole imaging data was collected by Weatherford using their CMI tool for an extensive interval of the Preese Hall well. The imaging data collected in the Preese Hall well are of high quality providing a clear view of the state of the borehole wall. The original De Paiter and Baisch report includes figures showing a number of sections of borehole imaging. Their Figure 31, which highlights the borehole ovalisation, includes interpreted dip data directly derived from the CMI imaging tool; this figure is included in Smythe's paper (his version included as figure 6), though with this dip data omitted. Therefore the author must have been aware of the availability of this borehole imaging data. (As the author describes in section 2) Data for all UK hydrocarbon wells are available from DECC appointed data release agents. Data typically includes composite plots, conventional geophysical logs and well data reports; Preese Hall data was released in April 2015. It would therefore have removed ambiguity from the author's interpretations if he had chosen to publish this original data, rather than reusing other author's figures. Examination of the borehole imaging for Preese Hall, collected prior to the fracking operation, showed unambiguous evidence of faults in this borehole in other sections of the well (for example at 2015 mbKB / 6613 fbkb showing a 40cm displacement fault). However, borehole imaging in the interval 8500 to 8640' shows consistent low angle bedding but no evidence of a fault intersection. The author could have accessed the borehole imaging data as it was available from release agents with all other data during April 2015. (IHS 2016, personal communication). It is unfortunate that the author chose not to examine the borehole imaging as this shows without ambiguity that his hypothesised fault does not intersect the wellbore in this interval.

INTERPRETATION OF THE FAULTING AT BALCOMBE

A further apparent manipulation of existing published data can be seen in section 4, dealing with faulting in the Weald Basin of Southeast England, which critiques the interpretations of Andrews, (2014). After examination of geophysical logs he hypotheses a 10m fault that crosses the Balcombe 1 well (some of these logs were included in the Andrews interpretation) which is adjacent to the recently drilled and still confidential Balcombe 2 well. He therefore postulates a fault to account for a hypothesised missing section. Examination of the log data used to demonstrate this 10m fault in Balcombe 1 highlights that some degree of exaggeration is apparent as about 3m of the raw log data has just been omitted. Adding that back in, the fault should have a thrown of about 6m. Whilst this may seem pedantic, this constitutes a 60% exaggeration.

CONCLUSIONS

The borehole imaging from Preese Hall well shows that Smythe's interpreted fault is not present. Given that a key element of his hypothesis can be disproven using freely available data I would recommend that this version of this paper should be withdrawn from publication by the author.

REFERENCES

Andrews, I. J., 2013: The Carboniferous Bowland Shale gas study: geology and resource estimation, British Geological Survey for Department of Energy and Climate Change, London, UK, https://www.gov.uk/government/publications/bowland-shale-gas-study, 2013. Andrews, I.J., 2014 : The Jurassic shales of the Weald Basin: geology and shale oil and shale gas resource estimation, British Geological Survey for Department of Energy and Climate Change, London, UK, de Pater, C.J. and Baisch S., 2011: Geomechanical Study of Bowland Shale Seismicity, Synthesis Report for Cuadrilla Resources Limited. http://www.cuadrillaresources.com/wp-content/uploads/2012/02/Geomechanical-Study-of-Bowland-Shale-Seismicity_02-11-11.pdf

[Figure]

**WELL TITLE: Preese Hall 1**

**Fig. 1.**

---

## Short Comment (SC16) · 29 Mar 2016

Andrew Kingdon. Petrophysicist British Geological Survey

---

## Short Comment (SC17) · 31 Mar 2016

Smythe's (2016) latest input to this thread concerns dip data derived from logging the Preese Hall-1 well using a CMI imaging tool. His main emphasis is to criticize Kingdon (2016) for citing the wrong figure: Kingdon (2016) cited Fig. 31 of de Pater and Baisch (2011) as the published source of these data, when he should have said Fig. 33. These data had previously been reported in Fig. 6.2 of Harper (2011), which was repeated as Fig. 33 of de Pater and Baisch (2011).

The excerpt from the raw imaging dataset for the part of the well that experienced deformation shows bedding oriented sub-perpendicular to the wellbore, as Smythe (2016) has said, the latter deviated at an angle of $\sim30°$ towards an azimuth of $\sim110°$ or

[Figure]

~S70°E. This orientation of the wellbore is consistent with published depictions of the well track (such as those by Clarke et al., 2014, and Westaway, 2016a) and confirms that Harper (2011) interpreted this imaging tool dataset correctly. The version reported by de Pater and Baisch (2011) is likewise correct: according to this dataset the bedding dips at ~30° towards an azimuth of ~290° or ~N70°W. Publication of this excerpt from the imaging tool dataset therefore does not add anything to the information already available, as Smythe (2016) has claimed; it confirms the existing interpretation of these data.

Conversely, the seismic section that Clarke et al. (2014) published, excerpted from their unpublished 3-D seismic dataset, shows bedding in the vicinity of this part of the wellbore with dip no greater than ~10°. I already reported the mismatching depictions of the bedding dip between these two datasets (Westaway, 2016b); its cause remains unclear, one possibility being that Clarke et al. (2014) have plotted the well track on the seismic section in the wrong place. Until this discrepancy is resolved, it is difficult to have confidence with any proposed geomechanical interpretation of this wellbore deformation.

Smythe (2016) has also reported that when he requested access to the Preese Hall-1 dataset the release package did not include this CMI imaging dataset. I note in passing that I have likewise been unable to obtain access to this dataset; the only part of it that I have seen is the excerpt just published by Kingdon (2016). Furthermore, had this dataset been released to me, it would have been under conditions that would have prohibited publication of any part of it, as Kingdon (2016) has now done. Something is clearly fundamentally wrong with the present arrangements for implementing the UK government's publically stated commitment to open disclosure and discussion of data pertaining to shale gas development.

References

Clarke, H., Eisner, L., Styles, P., Turner, P., 2014. Felt seismicity associated with shale

gas hydraulic fracturing: The first documented example in Europe. Geophysical Research Letters, 41, 8308–8314.

de Pater, C.J., Baisch, S., 2011. Geomechanical study of Bowland Shale seismicity: synthesis report. Cuadrilla Resources Ltd., Lichfield, 71 pp. Available online: http://www.rijksoverheid.nl/bestanden/documenten-en-publicaties/rapporten/2011/11/04/rapport-geomechanical-study-of-bowland-shale-seismicity/rapport-geomechanical-study-of-bowland-shale-seismicity.pdf (accessed 3 March 2016)

Harper, T.R., 2011. Well Preese Hall-1: The mechanism of induced seismicity. Geosphere Ltd., Beaworthy, Devon, 67 pp. Available online: http://www.cuadrillaresources.com/wp-content/uploads/2012/06/Geosphere-Final-Report.pdf (accessed 3 March 2016)

Kingdon, A., 2016. Comment on use of data and figures in Smythe paper. Interactive Discussion item SC15, 5 pp. Available online: http://www.solid-earth-discuss.net/se-2015-134/discussion (accessed 31 March 2016)

Smythe, D.K., 2016. Thanks for new PH-1 image data; faulting on 3D seismic not ambiguous; nothing missing from Balcombe logs. Interactive Discussion item AC6, 6 pp. Available online: http://www.solid-earth-discuss.net/se-2015-134/discussion (accessed 31 March 2016)

Westaway R., 2016a. The importance of characterizing uncertainty in controversial geoscience applications: induced seismicity associated with hydraulic fracturing for shale gas in northwest England. Proceedings of the Geologists' Association. doi: 10.1016/j.pgeola.2015.11.011, 17 pp.

Westaway, R., 2016b. Some additional thoughts on Preese Hall. Interactive Discussion item SC10, 9 pp. Available online: http://www.solid-earth-discuss.net/se-2015-134/discussion (accessed 9 March 2016)

SED

Interactive
comment

---

## Author Comment (AC6) · 31 Mar 2016

**Reply to Andrew Kingdon (se-2015-134-SC15)**

Mr Kingdon is a geophysicist by training and works in mathematical modelling, digital imaging, and statistics at the British Geological Survey. He has commented on two aspects of my paper.

**1. Preese Hall-1 faulting**

Firstly, Mr Kingdon discusses the CMI borehole imaging data, referring to figure 31 of de Pater and Baisch (2011), reproduced here as Figure 1 for convenience.

[Figure]

*Figure 1. Reproduction of de Pater and Baisch fig 31, for Preese Hall-1, showing well log azimuths (lilac) and interpreted bedding strike from CMI imaging (blue).*

Mr Kingdon states:

> *"Their Figure 31, which highlights the borehole ovalisation, includes interpreted **dip data** directly derived from the CMI imaging tool; this figure is included in Smythe's paper (his version included as figure 6), though with this **dip data** omitted. Therefore the author must have been aware of the availability of this borehole imaging data."* [my emphasis].

The blue line shows bedding **strike**, not dip as Mr Kingdon states. The blue bedding strike line as shown in Figure 1 is uninformative regarding the presence or otherwise of faults.

I am aware of the existence of the CMI imaging data, because two samples were reproduced in figure 19 of de Pater and Baisch, but as I have pointed out in my reply to Mr Huw Clarke of Cuadrilla (AC5) I do not have these data. They were not included in the Preese Hall-1 released data package which I obtained in April 2015. However Mr Kingdon has kindly published the image data for the section of interest (his figure 1; log from 8500 to 8640 feet). The image clearly shows bedding in the Lower Bowland Shale, apparently flat-lying relative to the wellbore, but the true dip is slightly to the west because the borehole is inclined steeply to the east in this section. There is no evidence of cross-cutting faults, nor of break-outs or fractures. I shall discuss this valuable new evidence in my reply to Dr Westaway.

I disagree with Mr Kingdon that my digital erasure of the narrow V-shaped labelling feature indicating frack stage 2 in the wellbore leads to ambiguity in the interpretation of the 3D seismic image. There cannot be any sudden disruption or alteration in the seismic data at this locality, because of the very band-limited nature and spatial sampling of the data. In addition, the erased V-shape is sub-parallel to the gently east-dipping seismic layering below the deviated portion of the well. Figure 2 shows a new version of the detail of interest, in which I have also removed the thick black line indicating the wellbore (I originally prepared this figure for my reply, AC5, to Mr Huw Clarke of Cuadrilla).

Any interpretation of such seismic data is, of course, imprecise, but within the limits of the data as seen objectively in Figure 2, there is a clear and important difference between the Clarke et al. (2014) fault location XX and my version YY. The general point is that my version YY separates the two different packages of reflectors - west-dipping to the east of the wellbore and east-dipping under the wellbore – where one might expect the fault to lie, whereas the Cuadrilla interpretation XX crosses the seismic layering, and is therefore incompatible with the presence of any fault with a throw of more than a few metres, i.e. below the resolution of the seismic. Both interpretations assume that there is a fault passing somewhere through the earthquake hypocentre. Imprecision is

due to the vertical component of wavelength of the seismic, of the order of 60-80 m, and the horizontal sampling (trace spacing), which appears to be 25 m.

[Figure]

*Figure 2. A. Clarke et al. (2014, fig. 4) sample of east-west 3D seismic through the Preese Hall-1 well. B. Line drawing of reflectors with alternative fault positions noted. The semicircle on the lowermost blocked-out area is the upper half of the hypocentral location.*

One can argue about the precise location of a fault (or other geological feature) intended to separate the two main seismic sequences. My version YY cuts the wellbore around two seismic cycles below the X shown in the central part of Figure 2B. But there is no merit in trying to locate the fault any more precisely than I have done, not least because there is an inconsistency in the superimposition of the wellbore onto the seismic image.

**2. Balcombe-1 faulting**

On the question of my demonstration that the Paddockhurst Park Fault, identified from BGS surface mapping, can be inferred in the Balcombe-1 well log, Mr Kingdon states:

> *"He therefore postulates a fault to account for a hypothesised missing section. Examination of the log data used to demonstrate this 10m fault in Balcombe 1 highlights that some degree of*

*exaggeration is apparent as about 3m of the raw log data has just been omitted. Adding that back in, the fault should have a thrown of about 6m. Whilst this may seem pedantic, this constitutes a 60% exaggeration."*

His assertion of a supposed 60% exaggeration is 100% wrong, as I now demonstrate.

Figure 3 shows the BGS version of part of the Balcombe-1 log in the area of interest, within the Kimmeridge Clay above the I-micrite. The red line shows where I cut the log to match it to that of Southwater-1. Figure 4 shows detail from my discussion paper, figure 8A, in which I matched up the Balcombe-1 wireline logs to those of Southwater-1 by cutting and separating vertically the Balcombe-1 logs.

[Figure]

*Figure 3. Reproduction of detail of litholog (centre) and wireline logs from Andrews (2014). White on litholog is Kimmeridge Clay, green is the I-micrite. Depth on metres subsea. Red line shows where the cut has been made to match the logs to the Southwater-1 well (Fig. 4).*

The cut was made in Balcombe-1 at 620 m. Figure 4 shows the near-perfect match of the logs once a gap of 10 m has been introduced. This represents the missing secion due to the fault. In Figure 4 I have added a second, uncut, version of the Balcombe-1 log, to show the match above 620 m (upper figure) and below 620 m (lower figure). In the latter case the log has been displaced downwards by 10 m.

[Figure]

Figure 4. Detail from Smythe discussion paper figure 8A to demonstrate the process of matching up Balcombe-1 (logs in black), with its missing section, to Southwater-1, with its presumed complete section (logs in red).

Nothing is missing from the wireline log images, as wrongly implied by Mr Kingdon. Any vertical imprecision in the logs will be of the order 1 m, which is approximately the thickness of the scale bar lines on the original BGS image.

**Conclusions**

I thank Mr Kingdon for supplying the Preese Hall-1 CMI image data, which is not, as he states, freely available, although it is nominally in the public domain. I shall take the new data into account in modifying my fault interpretation. Regarding the alleged missing log data in my interpretation of Balcombe-1, Mr Kingdon is wrong; no data have been cut from the logs. He should withdraw his comment.

**References**

Andrews, I.J. 2014. The Jurassic shales of the Weald Basin: geology and shale oil and shale gas resource estimation, British Geological Survey for Department of Energy and Climate Change, London, UK.

de Pater, C.J. and Baisch S. 2011. Geomechanical Study of Bowland Shale Seismicity, Synthesis Report for Cuadrilla Resources Limited. http://www.cuadrillaresources.com/wp-content/uploads/2012/02/Geomechanical- Study-of-Bowland-Shale-Seismicity_02-11-11.pdf

---

## Author Comment (AC7) · 31 Mar 2016

**Reply to Dr James Verdon (SC8)**

I respond under headings following  as far as possible the structure and order of his comments.

**General comments**

The logical, scientific presentation of my thesis – that there is a problem of faulting in the UK shale basins if shale exploitation is to proceed – follows a perfectly orthodox logical pattern:

- US shale play geology is simple and unfaulted ('foreland' basins).

- UK basins are highly faulted ('extensional' basins).

- Therefore US shale exploitation experience cannot be emulated in the UK.

- Fault identification in the Fylde, Lancashire.

- Hydrogeological risks in the Fylde.

- Inadequacy of the developer's technical work in the Fylde.

- Faulting in the Weald Basin, SE England.

- Inadequacy of technical work by two developers in the Weald.

- Importance of faults – review of quantitative modelling studies to date.

- Case history at Bradford County, Pennsylvania.

- Failings of UK regulation.

- Discussion - new exploration techniques and tighter regulation required in the UK.

Dr Verdon's view that my discussion paper is "*not really a scientific paper at all, in any normal sense of the term*" perhaps says more about his own prejudices than it says about my work. In his own blog Frack-land he has also published potentially libellous *ad hominem* comments about me; for example, for the last 18 months he has permitted an anonymous comment to appear online, castigating my alleged shortcomings in teaching some twenty years ago. The comment, claiming to be by a Glasgow graduate, is factually incorrect, and therefore worthless. Its continuing presence on his blog says more about Dr Verdon than it does about myself.

**Faulting in US shale plays**

I have thought long and hard about how (and whether) to publish a review of faulting in the US shale basins. It is a desk study, limited to public domain sources, although many of the latter are

incomplete and/or partial samples of proprietary seismic and well data. My study took about two months' full-time work.

The two principal problems about how to publish such a review are firstly, that it would depend on very many copyright sources which cannot simply be reproduced without permission, and secondly, that the main result – that there is practically no faulting in areas where shale oil/gas exploitation has been undertaken, is a negative result. Am I to be expected to publish a series of blank structure maps? There is no *Journal of Negative Results in the Earth Sciences*, although I was astonished to discover that at least two such journals do exist in other fields.

Dr Verdon quotes Younger and Westaway (2014), who argue that because the USA is so much larger than the UK, it is unlikely to have been mapped geologically to the same degree. So mapping quality may have affected the outcome. I accept that parts of the UK have been exceptionally well mapped, particularly where there have been coalmine workings, but the main problem is that sedimentary basins, whether in the UK or the US, are generally poorly exposed. But I cannot accept that a US field-mapping geologist, whether employed by the State or by the USGS, will on average have missed 99% or 99.9% of faults compared to his/her BGS counterpart, in terrain which is otherwise similar. The answer, in my view, is that there is a real difference in fault density, and that this real difference is explicable by the simple words 'foreland' *vs.* 'extensional'. Furthermore, the geometric differences in area and depth of the respective shale plays, of one to two orders of magnitude (discussion paper fig. 1 and table 1), which I presume these critics do accept as real, also tie in with the fault density difference of 2 to 3 orders of magnitude.

It is noteworthy that neither Dr Engelder nor any other American earth scientist has sought to question my conclusion. But there is a solution; if Drs Verdon, Westaway and Younger (three commentators who have either here or elsewhere criticised my review as being unpublished, and, worse! 'un-peer-reviewed') then they are free simply to search for and offer up one or two examples to refute my assertion.

If the Editor insists, I could add a supplement listing all the sources consulted for my US basin fault study, arranged by shale play, and with a few notes appended where required.

**The Fisher and Warpinski study of frack height growth**

I agree with Dr Verdon (and with Dr Westaway) that partial disclosure of the Halliburton dataset is better than non-disclosure. It is a common problem in earth science to be aware of expensive and valuable industry data, which for often sound commercial reasons cannot be published. We

academics sometimes have to beg for samples to be released.

But when the industry itself chooses to publish, we are right to ask, what is the agenda? A typical industry paper, peer-reviewed or not, almost certainly has an agenda, which is normally to promote the excellence of the company's products or services. In the case of Fisher and Warpinski (2012) the agenda is clearly the promotion of fracking, by demonstrating (and I accept their views, within the limits of what they have chosen to release) that fracture growth is highly unlikely to propagate upwards so as to contaminate groundwater resources. I do not impugn the integrity of Messrs Fisher and Warpinski, but it is legitimate to ask, as I did, commenting:

> *"This uncritical attitude towards an industry publication is surprising, as well as naïve, given that:*
>
> * *Halliburton's database remains confidential.*
>
> * *Wells are located only to county level.*
>
> * *Individual wells cannot be identified on the four main graphs presented.*
>
> * *We do not know whether inconvenient results have been omitted.*
>
> * *We do not know how complete is the database.*
>
> * *There are no wells in areas where complex geology (faults or tight folds) at the shale horizon extends to the surface."*

I also asked some other valid and pertinent questions about their database. For example, there is no reason why they could not have released their well locations, since well tops (and in some states, bottoms) are, in principle, in the public domain. Why do more academics not take a more critical, sceptical view?

In case the reader retains a belief in the benevolent nature of large companies in the hydrocarbon sector, this list of the top fines (year and amount in millions of $US) for corrupt practices imposed by the US Department of Justice and by the Securities and Exchange Commission might make them reconsider (hydrocarbon-related industries in bold):

1. Siemens            2008      800

2. Alstom             2014      772

3. **Halliburton**      **2009**      **579**

4. BAE               2010      400

| | | | |
|---|---|---|---|
| 5. | Total | 2013 | 398 |
| 6. | Eni + Snamprogetti | 2010 | 365 |
| 7. | Technip | 2010 | 338 |

[source: *Le Monde*, 20 January 2015].

**Microseismic data, pathways to faults**

Dr Verdon claims that my example of a stealth zone (fluid moving along a path not marked by microseismic activity) is self-contradictory. I disagree; the use of the word 'stealth' shows that the fluid went somewhere silently. The presence of a fault had to be inferred by the fact that microseismic activity had jumped to a new location. I conclude correctly that the deployment of a microseismic array, although essential, is no guarantee that all faults will be identified.

I agree that Hammack et al. (2014) is a useful paper, which I read first in October 2014 during my search for faults in US shale basins. It has a nice example of seismic data illustrating faulting at Marcellus level in western Pennsylvania, and also depicts many examples of microseismic activity. However, I am not so concerned with fracture growth, or even interaction with faults down at the shale level. This study demonstrates that upward growth, although penetrating beyond the Tully Limestone, nevertheless stopped well below Upper Devonian reservoirs and 2 km below any potable aquifers. But there are no through-penetrating faults in the monitored area; therefore it has no direct relevance to the main thrust of my discussion, which is that through-going faults from a fracked shale to the near-surface can put water resources at risk.

**Development of hydraulic fracturing in the UK**

Dr Verdon complains that in my section 2 I am seeking to 'settle scores' with other UK academics. I am certainly not, but when I read of senior academics presenting misleading or erroneous information to public bodies, whether by slide-show or not, critical comment is required, if only in order to lay to rest the kind of misinformation which these academics often accuse the anti-fracking lobby of promoting. I have had the whole Wytch Farm 3D seismic dataset, all the surrounding 2D data for the Bournemouth Bay and onshore area, and most of the wells on my computer interpretation database for the last eight years, for the purpose of conventional oil prospecting. I am fed up with reading about Wytch Farm in the misleading context of fracking, environmentally safe though it may be. This includes a newspaper report quoting Dr Verdon.

**Re-interpretation of Preese Hall-1 faulting**

I like Dr Verdon's assessment that 10 geophysicists, given the (minor) earthquake seismic data for the Preese Hall events, would come up with 10 different interpretations. The issue of the location of the hypocentre(s) and the fault that slipped has certainly not yet been resolved satisfactorily, but in my view my proposal that the fault was intersected by the wellbore is both novel and worth examining. I discuss this in more detail in my response to Dr Westaway, who has recently published on the same subject.

The history of the interpretation of the earthquake triggering and attempts to locate the fault serves to illustrate that understanding of the Bowland Shale is still very incomplete. I have never promulgated the view that earthquakes triggered by fracking is a problem to worry about; my concern has always been the risk of upward fluid flow *via* faults. A sensible approach would be for baseline monitoring to be set up in the Fylde, to run for two years or more, and for the 3D seismic dataset to be released now rather than in a year's time, when it nominally falls into the public domain. Perhaps then the various interested parties could then converge on a consensus view.

**Fracking in the Weald Basin**

Dr Verdon states:

> *"This section discusses **conventional** oilfield activities in the Weald Basin at Balcombe. The UK's 2015 Infrastructure Bill precludes hydraulic stimulation taking place at depths of less than 1,000m. The wells in question are at depths of approximately 800m. It is therefore difficult to see the relevance of this section to the development of hydraulic fracturing in the UK."* [my emphasis].

*The statement is disingenuous.* Dr Verdon maintains that the aborted well drilling in the Weald was "*conventional*" in nature. But he knows that Cuadrilla originally intended the drill horizontally and frack the limestone at Balcombe, as described in its successful planning application of 2010. Cuadrilla later assured the county council that it would not frack either Balcombe-2 or the deviated Balcombe-2z at those stages in the appraisal, but as Dr Verdon noted in his blog post dated 31 July 2013:

> *"In the last few weeks I've ...*[been] *deploying seismometers around Cuadrilla's planned Balcombe well. ... the current Cuadrilla plan is to drill into limestone for conventional oil, with **no intention of hydraulic fracturing at this stage**, but we wanted to get some experience deploying seismometers for this sort of situation."* [my underlining].

Dr Verdon explained that the deployment of seismometers in 2013 was intended to obtain baseline, or background, data on environmental noise levels. But I can only assume that this (sensible) experiment was in preparation for a future phase of fracking, which in the event never took place. If no fracking was ever intended, then there would be no risk of triggering earthquakes, and therefore the *raison d'être* of the baseline survey collapses (one does not need earthquake monitoring for conventional drilling).

Similarly, Celtique Energie said in its two applications to drill horizontally that it would not frack the wells in appraisal, but reserved the right to do so at a later stage.

*The statement is irrelevant.* His reference to the Infrastructure Act (which he wrongly calls a Bill) is irrelevant, as it only came into force in February 2015. I discussed the case histories of Cuadrilla drilling at Balcombe and the proposed drilling by Celtique Energie at two other sites in West Sussex to demonstrate the technical incompetence of the operators – wells misplaced on maps; misleading seismic data, faults ignored, and licence boundaries grossly in error. At the time that these proposals or actions happened there was no legal limit in place regarding a minimum depth for fracking. The 1000 m depth limit would now rule out fracking of the two Kimmeridgian micrites at Balcombe, and also at the currently controversial Horse Hill well north of Crawley, but would permit fracking in the Lower Kimmeridge Clay and in deeper targets such as the Oxford Clay and Lias. It would not have prevented fracking of the Kimmeridgian micrite at either of the two Celtique Energie sites.

**Modelling studies**

I did not cite the Hammack et al. (2014) paper here because it does not concern the possibility of fluid migration up to the near-surface *via* faults. The observations, interesting though they may be, were in an area which, as the authors demonstrate, the faulting is confined to sub-Tully Limestone levels.

I did allude to the Flewelling et al. (2013) paper by way of my organogram (fig. 9), in the context of a critique of Myers (2012), but forgot to add it to the reference list. Incidentally, Flewelling et al. repeat the error of Fisher and Warpinski (2012) in arguing that permeable faults cannot exist where hydrocarbons are present; an argument which conflates conventional and unconventional methods.

**Bradford County study**

I have provided a substantial re-interpretation of this important case history in my second response to Dr Engelder (AC3).

**Regulation**

Criticism of the UK unconventional oil and gas regulatory system might appear superficially to be out of place in an earth science journal, but if the criticism entails detailed technical analysis, understandable only by earth scientists, then an earth science journal is the only appropriate forum. How could a social science or legal journal handle such evidence?

Dr Verdon agrees with two other commenters (Younger and Westaway) that such comments are not appropriate, but I disagree with all three. It should be the duty of disinterested academics to comment upon perceived failings in the regulatory system, and if that extends to 'discrediting' certain agencies or government departments, as Dr Verdon implies, then so be it. I never identify individuals for criticism, unless it is completely unavoidable, because I know from my own experience of 14 years as a public servant with the BGS that such people have to work under certain constraints. But Dr Verdon and I are free of such constraints.

**Declaration of interest in acknowledgments**

I am flattered that Dr Verdon takes a close interest in my work, to the extent of guessing (albeit incorrectly) how much funding I allegedly get from objectors' groups. I am happy to declare the work I have been asked to undertake for such groups, in challenging unconventional exploration planning applications, or for appearing at local planning inquiries. I have also given several (unfunded) talks about fracking to the general public in the UK and in France.

Unfortunately, contrary to Dr Verdon's beliefs (expressed elsewhere), I make practically no money out of these ventures. I declare my consultancy income, which includes the small honoraria I have been given, to the UK tax authorities. From the tax year 2013-2014 onwards, my UK tax liability has been zero. In the last three or four years my research into fracking has been remunerated by these groups at an average rate of well under £1 per hour. However, my travel expenses have usually been reimbursed when I appear before county councils or planning inquiries.

I propose to declare my interests as follows:

> 'Declaration of interest: I work from time to time *pro bono publico* to assist groups of objectors in challenging unconventional hydrocarbon planning applications. These groups usually fund travel expenses, and I have sometimes been paid small honoraria.'

**References**

Fisher, K. and Warpinski, N. 2012. Hydraulic-fracture-height growth: real data. Society of

Petroleum Engineers Annual Conference Paper SPE 145949, Denver 2011. SPE Production & Operations, February 2012, pp 8-19, 2012.

Flewelling S.A., Tymchak M.P., Warpinski N., 2013. Hydraulic fracturing height limits and fault interactions in tight oil and gas formations: Geophysical Research Letters 40, 1-5.

Hammack R., Harbert W., Sharma S., Stewart B., Capo R., Wall A., Wells A., Diehl R., Blaushild D., Sams J., Veloski G., 2014. An Evaluation of Fracture Growth and Gas/Fluid Migration as Horizontal Marcellus Shale Gas Wells are Hydraulically Fractured in Greene County, Pennsylvania: EPAct Technical Report Series, U.S. Department of Energy, National Energy Technology Laboratory, Pittsburgh, PA, NETL-TRS-3-2014.

Myers, T.: Potential contaminant pathways from hydraulically fractured shales to aquifers. Ground Water 50, no. 6: 872–882. DOI: 10.1111/j.1745-6584.2012.00933.x, 2012.

Younger P.L. and Westaway R., 2014. Review of the inputs of Professor David Smythe in relation to planning applications for shale gas development in Lancashire (planning applications LCC/2014/0096 /0097 /0101 and /0102) and associated recommendations. http://eprints.gla.ac.uk/108343/1/108343.pdf

---

## Author Comment (AC9) · 31 Mar 2016

**Reply to Dr Rob Westaway (SC2)**

I am responding to Dr Westaway's comments using his own headings. This is an interim reply, since interactive comments close on 1 April 2016, and the Preese Hall fault problem deserves more detailed consideration following the release of the CMI log image by Mr Kingdon (SC15) on 29 March 2016.

**Introduction**

Dr Westaway's introductory comments begin in an antagonistic manner, particularly as in his initial comment SC1 dated 3 February 2016 he tried to present, in the guise of a standard Harvard-type citation, 'Seamark 2014'. This 'paper' was nothing less than a smear piece published in the UK tabloid press quoting some precipitate and abusive comments made about me by another UK academic. That version of his comment was removed by the editor at my request, as it fell well below the standards required of the journal discussion. However, once the discussion turns to the science, particularly of Preese Hall, it appears that Dr Westaway and I do share a lot of common ground.

**Geometry of the Preese Hall induced seismicity**

Dr Westaway starts this important topic with a reading list:

> *"my own outputs on this topic (none of which are cited by Smythe, 2016) include the Westaway and Younger (2014) and Westaway (2015, 2016) publications and the Younger and Westaway (2014) report"*

Westaway and Younger (2014) deals with the limitations of the 'traffic-light system of earthquake monitoring using local magnitude $M_L$. Although I have some sympathy with the authors' views, I did not discuss this paper, because it is peripheral to the main problem. The commentary (Younger and Westaway (2014) on my submissions to Lancashire County Council is nominally in the public domain, as all submissions to LCC should have been, but I only became aware of its contents in summer 2015. As the new frontispiece intimates, it is now somewhat of historic interest, as the authors' more recent work has superseded their views expressed in that report.

I could hardly have cited his latest paper (Westaway 2016) in a paper submitted on 22 December 2015, when his own paper only went online on 7 January 2016. However, I am grateful for the chance to discuss it here. He and I evidently share some common ground.

Like Dr Westaway, I too was confused by the mismatch of the illustration of the focal plane solutions (Clarke et al. 2014, fig. 4) with the geometry of the fault orientation. This is given by Clarke et al. in supplementary publication B as "*strike, dip and rake of 40°; 70°; -150° (resp. 299°;*

*62º; -23º)*". However, the second triplet of figures matches the illustration well if one assumes that white quadrants represent compression.

The uncertainty in the fault planes is stated to be about 20º. Clarke et al. concluded that all three earthquake studies have very similar locations and mechanisms, being no more than 170 m apart (supplementary publication A). For that reason I accepted their hypocentral location.

Westaway accuses me of basing my interpretation on "*uncritical acceptance of the accuracy of this Clarke et al. (2014) hypocentre*". I do not intend to enter the hypocentral location discussion here, suffice to say that the seismological data are rather poor. For some arrivals there is even doubt about the polarity as well as the onset time. Dr Verdon (comment SC8) shares my misgivings.

In my view the most important data (for which, unfortunately, we only get a tiny sample) are the 3D seismic data and its interpretation. We do not even need a hypocentral location to identify a fault or fault zone. I reproduce here the line drawing I prepared for my response to Huw Clarke (Figure 1 below), but this time with the oblique view of the time slice at 2930 m visible.

[Figure]

Figure 1. A. Clarke et al. (2014, fig. 4) sample of east-west 3D seismic through the Preese Hall-1 well. The grey-shaded lower part is an oblique view of the seismic timeslice at 2930 m depth, with a bifurcating fault picked by them shown as a dashed line.

B. Line drawing of reflectors with alternative fault positions noted. The semicircle on the lowermost blocked-out area is the upper half of the hypocentral location.

Figure 2 is an attempt to portray the 2930 m depth timeslice on a map. The bifurcating fault shown on the timeslice (white dashed lines) follows a light event on the timeslice. The interpreted fault (Clarke et al. 2014, fig. 1) seems to be picked (red line) through the middle of what could be interpreted as being more of a complex fault zone than the discrete line mapped by Clarke et al.

[Figure]

*Figure 2. Timeslice of Clarke et al. (2014) anamorphically stretched to fit a map of the Preese Hall-1 locality. The fault interpretation of Clarke et al. (2014, fig.1) from their map is shown in red, and the fault(s) as marked on the timeslice are shown by white dashed lines. Dotted lilac lines are other possible fault structures*

In my view the 3D seismic evidence of a fault zone is stronger than any inference from hypocentral locations and focal plane solutions.

The fault zone passes through the lower part of the wellbore, but there is another problem in locating the wellbore on the seismic image. The wellbore deviation shown in Clarke et al. is steeper by a few degrees than the shape implied by the now-released well coordinates, implying that the well as depicted on the 3D seismic image has not been accurately placed. Dr Westaway has just pointed this out independently (SC17). The seismic image cannot be located precisely either to the

OS grid system or to the well top. In short, there are too many unknowns either for myself or for Dr Westaway to attempt to pinpoint precisely where the fault lies.

Another important piece of information is the CMI image spanning the deformed wellbore section. This was only made available by Mr Kingdon of the BGS in his comment (SC15) on 29 March 2016. Clearly the well casing deformation must have been by bedding plane slip, as first recognised by de Pater and Baisch (2011), so the problem now is how to reconcile bedding plane slip over 160 m of the wellbore with the fault zone passing through the well.

**Selective referencing**

Contrary to what Dr Westaway states, I did not "*praise*" the Myers (2012) paper. I cited (briefly) the various critics (see my organogram, fig. 9), and did, however, point out that although unsuitable for Appalachian geology the model Myers used might have some applicability in a UK setting.

**Regulation**

I have commented in my response to Dr Verdon on this subject, and also provided in my reply to Professor Younger a lot more new evidence explaining why I am concerned; firstly, about the EA's view on the non-potability of groundwater at 300 m below the Fylde, and secondly, the still poorly-known location of the important Woodsfold Fault, which separates the largest groundwater resource in NW England from the Fylde.

But there is one other point of discussion that Dr Westaway has raised, the subject of regulating drilling through faults. He believes that I consider that this should be prohibited. But I did not imply that; explorationists can hardly avoid faults, and, indeed, often penetrate them unknowingly. Water drillers target shallow faults because they have the best hydaulic conductivity. What I wish to see better regulated in the UK is drilling through or near to faults at or above a shale volume to be fracked. We are in some measure of agreement here, except that I would never entrust a decision to a developer, to decide whether or not to drill or avoid a fault, on the Panglossian premise that the develop will make a sound economic judgment.

**References**

Clarke, H., Eisner, L., Styles, P., Turner, P., 2014. Felt seismicity associated with shale gas hydraulic fracturing: The first documented example in Europe. Geophysical Research Letters, 41, 8308–8314.

Westaway, R., 2015. Induced Seismicity. In: Kaden, D., Rose, T.L. (eds.), Environmental and Health Issues in Unconventional Oil and Gas Development. Elsevier, Amsterdam, pp. 175-210.

Westaway R., 2016. The importance of characterizing uncertainty in controversial geoscience applications: induced seismicity associated with hydraulic fracturing for shale gas in northwest

England. Proceedings of the Geologists' Association. Doi: 10.1016/j.pgeola.2015.11.011, 17 pp.

Westaway, R., Younger, P.L., 2014. Quantification of potential macroseismic effects of the induced seismicity that might result from hydraulic fracturing for shale gas exploitation in the UK. Quarterly Journal of Engineering Geology and Hydrogeology, 47,333–350.

Younger, P.L., Westaway, R., 2014. Review of the Inputs of Professor David Smythe in Relation to Planning Applications for Shale Gas Development in Lancashire (Planning Applications LCC/2014/0096 /0097 /0101 and /0102) and Associated Recommendations. Report to Lancashire County Council, 12 pp. + 1 p. preface. University of Glasgow; available online: http://eprints.gla.ac.uk/108343/

---

## Short Comment (SC18) · 1 Apr 2016

Mercifully, the 'discussion' phase for Smythe's (2016a) manuscript is now almost at an end. In my view this journal has given him far too much latitude, first, to allow his manuscript to be posted online in the first place, containing as it does so much innuendo directed against members of the UK Earth Science community; he should confine such allegations to his notorious website. As the reviewer has made clear (Aplin, 2016), this manuscript contains no content that might justify publication, yet it has been allowed to appear online as a permanent record within the public domain, associated with and thus given credibility by a leading scholarly society. This author has also been allowed by the journal to include in his postings on this discussion phase

much material that can only be described as personal attacks on his critics, which will likewise persist indefinitely within the public domain. It would be appreciated if the journal can ensure no repetition of this episode.

I shall deal briefly with each of the substantive points mentioned by Smythe (2016b) and not covered in previous postings in the order they are set out.

(1) Smythe's (2016b) first comment concerns the fact that, in response to a complaint from him, the journal editor asked me to remove a citation of a newspaper article by an investigative reporter, which had been highly critical of his past conduct opposing shale gas development in the UK. The editor indeed stated 'I am thus writing this email after having received a complaint from Dr. Smythe who felt offended by the first part (the first three lines and the corresponding references) of your comment, where his scientific reputation is doubted by citing "non scientific" press articles.' Although I complied with this request, I had been surprised to receive it, given that the Smythe (2016a) manuscript cites a great deal of similar material, not just newspaper content but also website postings by environmental activist groups. In response to this, I removed the citation of the one newspaper article that I had cited. I am most surprised that Smythe (2016b) has now aired this issue over again, and has thereby been allowed to place in the public domain his own spin on the matter. It therefore now seems reasonable to provide the reference details for the newspaper article he is alluding to, by Seamark (2014), so readers can judge the matter for themselves.

(2) His second comment concerns the Westaway and Younger (2014) article regarding the strength of ground vibrations from induced seismicity caused by fracking. He now states that Smythe (2016a) 'did not discuss this paper, because it is peripheral to the main problem'. Nonetheless, much of the content of Smythe (2016a) concerned the adequacy or otherwise of existing UK regulations for any future shale gas industry. As part of this, Smythe (2016a) gave the impression that induced seismicity is in his view one of few aspects that are currently adequately regulated. On the contrary, the gist of Westaway and Younger (2014) is that the current 'red traffic light' limits for such

regulation, of magnitude 0.5, is so low as to be ridiculous: in particular, it makes no sense to regulate ground vibrations from fracking much more stringently than ground vibrations from other forms of human activity, such as from quarry blasting, construction activity, or indeed from use of vehicles on local roads.

(3) Smythe (2016b) claims to have independently spotted the inconsistency in the documentation by Clarke et al. (2014) of the focal mechanism orientation for the Preese Hall-1 microseismicity, even though he did not mention this until after it had been pointed out in my publications and postings in this thread (e.g., Westaway, 2015, 2016a, 2016b). His solution to resolving this inconsistency is to presume that the un-shaded (white) quadrants of this focal mechanism are compressional and the shaded (red) quadrants are dilatational, the opposite of how such diagrams are usually drawn. Nonetheless, this cannot possibly be the correct explanation for a number of reasons, for example: (a) if it were so, the downward vertical would lie in a compressional quadrant, so the focal mechanism would involve strike-slip and reverse faulting, rather than strike-slip and normal faulting. However, the key issue requiring resolution has been the meaning of the negative rake angle reported by Clarke et al. (2014); if there were a component of reverse faulting to the focal mechanism, then the rake angle would be positive, by definition. In addition (b) if the red quadrants of the published focal mechanism were dilatational, the minimum principal stress (which is roughly east-west in this vicinity; e.g., Westaway, 2016a) would lie within a dilatational quadrant, a physical impossibility (cf. McKenzie, 1969). Smythe's (2016b) claim that the white quadrants of this published focal mechanism are compressional and the red quadrants dilatational therefore makes no sense, and joins the list of pieces of demonstrably wrong 'information' that he has been allowed to place in the scientific literature.

(4) Smythe (2016b) challenges the statement by Westaway (2016b) regarding his "uncritical acceptance of the accuracy of this Clarke et al. (2014) hypocentre" for the Preese Hall induced seismicity. He says that the underlying data are poor and the precise location of the activity is immaterial, but he has nonetheless used the Clarke et al.

(2014) hypocentre in his construction of the geometry of the seismogenic fault and has illustrated this hypocentre, and no other candidate locations, in several of his diagrams. As already noted, the problems with the Clarke et al. (2014) location procedure, including their use of a seismic velocity model that is too fast, exaggerating the depth of the hypocentre, and not factoring in the clear lateral variations in seismic velocity structure that result from the dipping stratigraphy (e.g., Westaway, 2016a), cast strong doubt on this particular hypocentre.

(5) Smythe (2016b) now also claims to have independently realised that the likely explanation for various contradictory aspects of the published parts of the Preese Hall-1 dataset is that the well track is not marked in the correct place on the Clarke et al. (2014) seismic section, even though this is another 'fact' that he never mentioned until after I had pointed it out (cf. Westaway, 2016c, 2016d).

(6) Smythe (2016b) claims that I exaggerated the support provided for the Myers (2012) paper in the Smythe (2016a) manuscript. On the contrary, Smythe (2016a) provides multiple favourable citations of this flawed paper, including stating on the basis of its results that 'when fracking occurs the transport times of contaminated fluid from the fracked shale to the near surface can be reduced to a few tens of years "or less"'. Engelder (2016) has covered these aspects thoroughly and no further deliberation is needed.

(7) Finally, Smythe (2016b) criticises me (Westaway, 2016b) for claiming that he had advocated that drilling through faults (in relation to shale gas development) should be prohibited. However, he has repeatedly argued this in his submissions to local authorities in the UK. For example, when objecting to one proposal for shale gas development, Smythe (2014) wrote 'Cuadrilla has defined so-called 'regional' faults, which will be avoided by the fracking operations, and 'local' faults, through which drilling and fracking may take place. Its definitions are inconsistent and illogical. All faults should be avoided, whatever the scale; if this results in the Bowland Basin being unexploitable for shale gas, then so be it.' Smythe (2016a) likewise states 'In the UK there is as

yet neither legislation nor guidance on the what should be the minimum ('respect' or stand-off) distances from faults, vertically and horizontally, of both the wellbores and the fracked shale volumes', implying that when this author wrote that a few months ago he still favoured some form of legal prohibition on drilling through faults, even if he has changed his mind since.

References

Aplin, A., 2016. Smythe se-2015-134 Review. Interactive Discussion item RC1, 3 pp. Available online: http://www.solid-earth-discuss.net/se-2015-134/discussion (accessed 31 March 2016)

Clarke, H., Eisner, L., Styles, P., Turner, P., 2014. Felt seismicity associated with shale gas hydraulic fracturing: The first documented example in Europe. Geophysical Research Letters, 41, 8308–8314.

Engelder, T., 2016. Advocacy-Based Science. Interactive Discussion item SC4, 11 pp. Available online: http://www.solid-earth-discuss.net/se-2015-134/discussion (accessed 31 March 2016)

McKenzie, D.P., 1969. The relation between fault plane solutions for earthquakes and the directions of the principal stresses. Bulletin of the Seismological Society of America, 59, 591-601.

Seamark, M., 2014. Anti-fracking 'expert' and question marks over his credentials: Ex punk rocker 'lied and peddled pseudo science'. Daily Mail Online. Available online: http://www.dailymail.co.uk/news/article-2713509/Scientist-claims-fracking-dangerous-argues-against-drilling-applications-fraud-lied-credentials.html (accessed 1 February 2016)

Smythe, D.K., 2016a. Hydraulic fracturing in thick shale basins: problems in identifying faults in the Bowland and Weald basins, UK. Solid Earth Discussion; doi: 10.5194/se-2015-134, 45 pp.

Smythe, D.K., 2016b. Interim reply to Dr Westaway. Interactive Discussion item AC9, 5 pp. Available online: http://www.solid-earth-discuss.net/se-2015-134/discussion (accessed 1 April 2016)

Westaway, R., 2015. Induced Seismicity. In: Kaden, D., Rose, T.L. (eds.), Environmental and Health Issues in Unconventional Oil and Gas Development. Elsevier, Amsterdam, pp. 175-210.

Westaway R., 2016a. The importance of characterizing uncertainty in controversial geoscience applications: induced seismicity associated with hydraulic fracturing for shale gas in northwest England. Proceedings of the Geologists' Association, in press. doi: 10.1016/j.pgeola.2015.11.011

Westaway R., 2016b. Comment on "Hydraulic fracturing in thick shale basins: problems in identifying faults in the Bowland and Weald basins, UK" by D.K. Smythe'. Interactive Discussion item SC2, 8 pp. Available online: http://www.solid-earth-discuss.net/se-2015-134/discussion (accessed 9 March 2016)

Westaway, R., 2016c. Some additional thoughts on Preese Hall. Interactive Discussion item SC10, 9 pp. Available online: http://www.solid-earth-discuss.net/se-2015-134/discussion (accessed 9 March 2016)

Westaway, R., 2016d. Preese Hall-1 bedding dip. Interactive Discussion item SC17, 4 pp. Available online: http://www.solid-earth-discuss.net/se-2015-134/discussion (accessed 31 March 2016)

Westaway, R., Younger, P.L., 2014. Quantification of potential macroseismic effects of the induced seismicity that might result from hydraulic fracturing for shale gas exploitation in the UK. Quarterly Journal of Engineering Geology and Hydrogeology, 47, 333–350.

---

## Author Comment (AC11) · 1 Apr 2016

I find it regrettable that Dr Rob Westaway, of Glasgow University, now chooses to malign the journal that dares to publish views with which he disagrees, and instead of engaging in open discussion on what is evidently a controversial topic, is forced to fall back upon the fulminations of the tabloid press.

It is even more regrettable that Dr Westaway, in reviving ad hominem attacks, risks endangering the good academic reputation of the ancient Scottish university of which we are both members.
* * *

---

## Editor Comment (EC3) · F. Rossetti (Editor) · 2 Apr 2016

In his SC18 (Solid Earth Discuss., doi:10.5194/se-2015-134-SC18, 2016) on the Discussion paper SE2015-134 by Dr. D. Smythe, Dr. Westaway stated that the SE journal has provided a scientific platform to present a biased view of UK Earth Science Community and that the journal should not host similar papers in the future. As handling editor of this SE Discussion paper, I have to point out that, despite controversial and provocative in some parts, the SE2015-134 paper deserved to be handled in the same way as all the papers submitted to SE. It is online publication has stimulated a vigorous scientific debate that I consider has somehow contributed to improve the state of knowledge on the various aspects dealing with the hydraulic fracturing technique

(fracking) and its applications. I would also remark that the open-access platform of SE ensures all the different opinions are adequately represented during the editorial workflow, providing a fair and unbiased review process. Finally, regarding point (1) of the SC18 of Dr. Westaway: following the SE editorial policy, as topical handling Editor of the manuscript I have the duty to censor "comments that are not of substantial nature or of direct relevance to the issues raised in the discussion paper or which contain personal insults, especially if Authors…notify the topical editor in case of abusive comments." I can thus just re-iterate what I wrote in my email to Dr Westaway after having received a complaint from Dr. Smythe about his SC2: "Although I am aware that there is a vigorous debate in UK on the fracking technique that is not only confined to the scientific community, I believe that comments on scientific articles should be instead confined only to scientific issues. On this regard, I would also emphasise that comments that regards interpersonal relationships and aspects of the professional reputation of the Author (or anyone else) are to be avoided, because not in discussion here.

Federico Rossetti

---

## Short Comment (SC19) · 4 Apr 2016

Dear Dr. Westaway,

the first part of your interactive comment SC18 contains criticism of the Journal's editorial policy regarding the handling of manuscript se-2015-134 authored by D.K. Smythe. I'm sorry to say that, from what you wrote, it seems to me that you are not fully familiar with the EGU open access publication policy. Actually, all manuscripts that are sent out for review are automatically published online in the Discussion Section of the journal (SED) and get a doi number. In such a way we provide the possibility for anyone, not just the reviewers, to make constructive comments suitable to help the relevant Topical Editor come to decide on the manuscript. So, what has happened so far with

manuscript se-2015-134 is fully compliant with the workflow of SE and all EGU open access journals. Publication in SED does not mean final acceptance in SE because manuscripts in SED can be rejected dependent on the reviews and interactive comments. Since everything is available for free download, the interactive comments are also part of the "permanent record within the public domain" thus allowing interested readers to understand in a comprehensive way what is the subject of the debate. I would say that a very poor manuscript, typically rejected, does not contribute to give credibility to the Author(s). To conclude, as Chief Executive Editor of Solid Earth, I have a twofold feeling on what is happening with this manuscript. On one hand, I'm very happy to see that, given the sensitive issue, the manuscript stimulated a vigorous discussion, thus utilising the editorial workflow. On the other hand, I have the impression that not all the content of the interactive comments/replies comply with the purpose of providing constructive contributions to the review process of the manuscript, and this is not what we are looking for. I fully agree with the Topical Editor that the ongoing debate is positive in helping to make further progress on such a delicate issue and, consequently, I would recommend that you, and everybody else that adds further interactive comments, deals strictly with the science of the manuscript and nothing else. Last but not least, given the societal importance of this issue, the ongoing discussion on manuscript se-2015-134 may suggest the need of a review paper on the subject. We will be glad to handle it in Solid Earth.

Best regards,

Fabrizio Storti
* * *

---

## Short Comment (SC20) · 14 Apr 2016

Smythe seeks to minimise differences in our view of the hydrogeology of faults by recourse to semantics (the definition of 'inherent'). This will not do. Our views are not close. No amount of special pleading over the interpretation one may put on the word 'inherent' will change that consensus. I cited textbooks in support of my position because these best represent the consensus view in the discipline. In support of his own position, Smythe quotes from a German text which makes a statement which is reasonable (after all, excessive water use or the usual spillages of any industrial sites can 'entail considerable environmental risk, particularly when it comes to water resource conservation' – at least in administrations with poor regulatory processes).

This point has been made repeatedly in major reviews, such as those of Mair et al. (2012) and Masters et al. (2014). However, the quotation from the German text makes no mention of faults, and thus in no way supports Smythe's central and erroneous contention in relation to them.

Having been forced by my comments to acknowledge that he misleading cited irrelevant papers concerning karstified limestones, Smythe clings to the wreckage of his argument by referring to (though not, I note, citing any literature to support) deep circulation systems that are very definitely hosted by limestones. Again, citation of some literature - any literature – that actually substantiates his minority-of-one claims about hydraulically continuous high permeability in faults traversing thick shale sequences might have helped take the argument forward. However, he has signally failed to cite any such literature, for the simple reason that no such literature (and no such phenomenon) exists. (Incidentally, Smythe does not bother to cite any of the many papers on the origin of the Bath hot springs; had he referred to the literature he might have discovered that, although the Carboniferous Limestone source and approximate minimum age of the waters ($\sim$ 1,000 years)are now reasonably well constrained (Edmunds 2004), the actual location of the recharge area has never been definitively established; while the Mendips is widely presumed (e.g. Atkinson and Davison 2002), other karst hydrogeology specialists argue convincing for a South Wales source area (Wilcock and Lowe 1999)).

Again, without bothering to quote any relevant sources, let alone engaging with the irrefutable temporal sequence of hydraulic head conditions induced by shale gas operations, Smythe simply states that "upward flow is possible; the questions which remain the subject of debate are the precise mechanisms, and the timescales". As I pointed out in my earlier comment, no one denies that upward hydraulic gradients are temporarily established during the very brief period of hydraulic fracturing. There is no doubt about "timescales" – fracking takes days, whereas depressursation of wells to allow gas to enter them persists as long as the well is in production, and thus ensures

downward gradients over periods of years to decades. Unless we are to set aside the basic laws of physics, there is no legitimate "debate" about this. Given the miniscule permeability of the unfractured shale beyond the artificially fractured zone (cf Yang and Aplin 2007), re-establishment of an upward gradient after cessation of production (and thus suspension of active depressurisation) is unlikely to occur over anything less than geological time. In any case, in the UK, the history of ongoing uplift that commenced in the early Cenozoic (e.g. Westaway 2009) means that over-pressured conditions are not anticipated in onshore unconventional reservoirs even before exploitation – let alone after years or decades of sustained depressurisation. Undaunted, Smythe goes on to say "conclusion, [sic] upflow can happen, driven by several different forces". No. Anyone who has read and understood the mainstream hydrogeological textbooks I quoted in my previous comment will know that there is only one "force" that drives groundwater flow, and that is hydraulic head - a summation of pore pressure and relative elevation (a proxy for the effect of gravity). No other forces drive upward flow – or flow in any direction. This again is basic physics.

Smythe affects to suggest – without offering any argument or references in support of his position – that the examples of saline springs which I referenced do not constitute an appropriate analogy. Why not? Until some argument to the contrary is proffered, the analogy stands unchallenged. Furthermore, to argue that the analogy "evidently excludes gas (especially methane) migration" is to miss the point: these springs do not emit methane. In fact methane migration through groundwater is a very slow process – unless it is at very high bubble pressures (which is not to be expected in the UK, given the history of Cenozoic uplift mentioned above). Other than that the only mechanism of transport is by diffusion through the groundwater, which is invariably far slower than even the slowest advection (again see the standard groundwater textbooks I cited in my earlier comment). For this very reason, the flooding of methane-emitting mines in the UK invariably leads to an abrupt cessation of methane release (Jardine et al. 2009; Younger 2014). This, it should be noted, regularly puts abandoned mine methane capture systems out of business – and this in a hydrogeological environment that is

many orders of magnitude more permeable than even the most well-fracked shale gas reservoirs (Younger 2014). Smythe indulges in a nostalgic reflection on the bad old days of dilute-and-disperse approaches to management of contaminants. As it has long been accepted that it is the loading (dimensions of M T-1) rather the concentration (M L-3) that controls the dosage affecting sensitive biological receptors, dilute-and-disperse has been out of regulatory fashion for decades, and has no relevance to this debate. At no point in my text did I advocate a return to that approach. I was simply arguing that undetectably low concentrations effectively equal zero pollution. It is not "invalid" to argue as I did. My arguments instinctively follow the common approach to first-tier hydrogeological risk assessment (e.g. Banwart et al.2002) in which a breakthrough concentration of 100% of source strength is routinely assumed – not merely the 90% of initial concentration which Smythe triumphantly quotes (finally citing a source, Gassiatt et al. 2013). Assuming 100% is routine because it defines the worst-case scenario. At no point in my comment did I argue for any lesser percentage. Nevertheless, my paragraph began with the words 'let us suspend disbelief'; if we decline to do so, my arguments become even less assailable.

Smythe appears to assume that the dissolution of halite is the principal source of salinity in the Permo-Triassic sandstone aquifers of the UK. This is erroneous. Sources of salinity at depth in UK aquifers are many and varied, including trapping of ancient sea waters, dissolution of silicate minerals over very long time-scales, possible 'membrane filtration' through shales subject to high hydraulic gradients, and sub-permafrost solute exclusion during the Quaternary cold periods (see Younger et al. (2015) and extensive literature cited therein). Localised production of saline water by evaporite dissolution within the Mercia Mudstone Group has long been recognised, of course, and diffusional transfer into adjoining aquifers over geological time has been documented (see, e.g. Younger 1995). The lack of rapid circulation at depth in deeply confined aquifers means that such diffusion can indeed contribute to the salinity in deeper aquifers, though such evaporite sources are by no means the principal explanation of the ubiquitous saline groundwaters encountered at depth in UK aquifers (see, e.g., Bottrell et al. 2006).

Smythe then proceeds to "examine the Fylde evidence" – though, breathtakingly, this examination does not engage with a single one of the many hydrogeological studies of the Fylde Aquifer published over the decades (including Allen et al. 1997; Barker and Worthington 1973a, 1973b; Brereton and Skinner 1974; Lovelock 1977; Oakes and Skinner 1975; Seymour et al. 2006; Tellam and Barker 2006; and Worthington 1977). Had Smythe taken the time to read these (and many other) publications on the Fylde Aquifer, he would have had the opportunity to understand that the distribution and movement of water in the aquifer is very well understood, and leaves no room for his uninformed speculation. He would also have had the opportunity to appreciate that historical pumping of formerly freshwater boreholes in the aquifer has led to saline intrusion – principally from saline waters down-dip within the Sherwood Sandstones. Being unsupported by any engagement with the extensive literature on the Fylde Aquifer, Smythe's special pleading DOES NOT make the case for the persistence of freshwater to great depths – on the contrary, the loss of freshwater status in the boreholes he mentions is but another example of the worldwide phenomenon of saline intrusion, which demonstrates the opposite of what Smythe claims. A basic understanding of the tendency for higher-density saline waters to sink beneath lower-density freshwaters, and the delicate balance between them (as per the well-known Ghyben-Herzberg relationship that is documented in all groundwater textbooks) might have spared Smythe from making his rather embarrassing suggestion of the occurrence of freshwater beneath the saline water within the aquifer. Hence Smythe's claim that "the EA has written off a past and future potential groundwater resource in the Fylde" is not in the least justified by the brief notes from his own partial study, which has resolutely ignored the work of the bona fide hydrogeologists I cite above.

Other commentators have dealt with Smythe's defamatory pronouncements on what he alleges to be a systemic lack of professionalism, honesty and integrity on the part of the Environment Agency. He has yet to undertake (let alone cite, for none exists) any sociological investigation that would remotely support his scurrilous remarks. Without rigorous evidence of a proper examination of the performance of the EA by appropriately qualified social scientists (which Smythe is not) his own anecdotal "evidence" has no place in a supposedly scientific article.

Smythe closes his response to my earlier comment by remarking that he finds my "comments to be somewhat dogmatic in tone". While I do not consider that personal taste in literary 'tone' is a legitimate point for scientific discussion, this curious remark prompts me to respond that this is the first time I have had to deal with a supposedly scientific debate in my core discipline (hydrogeology) in which I have had to expend much of my text explaining (and now re-explaining) absolutely basic points of hydrogeology that anyone could learn by spending a little time reading the various textbooks I cited in my previous comment. It is a characteristic of entry level classes in any subject that they tend to be rather 'dogmatic', since those who are not versed in the basics of any discipline are not yet at a sufficient level of understanding to engage in sophisticated debate. Accordingly, I would not presume to submit a discussion on points of deep seismic surveys (in which Smythe is expert and I am not); but were I rash enough to do so, I would not be in the least surprised if my arrogance were to be met by somewhat dogmatic answers.

References

Allen, D J, Brewerton, L J, Coleby, L M, Gibbs, B R, Lewis, M A, MacDonald, A M, Wagstaff, S J, and Williams, A T. 1997. The physical properties of major aquifers in England and Wales. British Geological Survey Technical Report WD/97/34. 312pp. Environment Agency R&D Publication 8.

Atkinson, T.C., and Davison, R.M., 2002, Is the water still hot? Sustainability and the thermal springs at Bath, England. In: Hiscock, K. M., Rivett, M.O. & Davison, R. M. (eds) Sustainable Groundwater Development. Geological Society, London, Special Publications, 193: 15-40.

Banwart, S.A., Evans, K.A., and Croxford, S., 2002, , Predicting mineral weathering rates at field scale for mine water risk assessment. Geological Society, London, Special

Publications, 198:137-157 (doi:10.1144/GSL.SP.2002.198.01.10).

Barker, R D, and Worthington, P F. 1973a. The hydrological and electrical anisotropy of the Bunter Sandstone of Northwest Lancashire. Quarterly Journal of Engineering Geology, 6, 169–175.

Barker, R D, and Worthington, P F. 1973b. Some hydrogeophysical properties of the Bunter Sandstone of northwest England. Geoexploration, 11, 151–170.

Bottrell, S.H., West L.J., & Yoshida, K., 2006, Combined isotopic and modelling approach to determining the source of saline groundwaters in the Selby Triassic sandstone aquifer, UK. Geological Society, London, Special Publications 263: 325-338. (doi: 10.1144/GSL.SP.2006.263.01.19).

Brereton, N R, and Skinner, A C. 1974. Groundwater flow characteristics in the Triassic Sandstone in the Fylde area of Lancashire, Water Services, 78, 275–279.

Edmunds, W.M., 2004, Bath thermal waters: 400 years in the history of geochemistry and hydrogeology. In Mather, J.D. (Ed.), 200 Years of British Hydrogeolog. Geological Society of London Special Publication 225: 193–199.

Gassiat, C., Gleeson, T., Lefebvre, R., and McKenzie, J. 2013. Hydraulic fracturing in faulted sedimentary basins: Numerical simulation of potential long term contamination of shallow aquifers, Water Resources Research, 49(12): 8310-8327, doi:10.1002/2013WR014287.

Jardine, C.N., Boardman, B., Osman, A., Vowles, J., and Palmer, J., 2009, Chapter 8: Coal Mine Methane. In: Jardine, C.N., Boardman, B., Osman, A., Vowles, J., and Palmer, J. (editors) Methane UK. Environmental Change Institute, University of Oxford. pp. 64-71. (available on-line at: http://tinyurl.com/lcryduw; last accessed 19-8-2013).

Lovelock, P E R. 1977. Aquifer properties of the Permo-Triassic sandstones of the United Kingdom. Bulletin of the Geological Survey of Great Britain, No. 56, 50.

[Figure]

Mair, R., Bickle, M., Goodman, D., Koppelman, B., Roberts, J., Selley, R., Shipton, Z., Thomas, H., Walker, A., Woods, E., & Younger, P.L., 2012, Shale gas extraction in the UK: a review of hydraulic fracturing. Royal Society and Royal Academy of Engineering, London. 76pp.

Masters, C., Shipton, Z., Gatliff, R., Haszeldine, R.S., Sorbie, K., Stuart, F., Waldron, S., Younger, P.L., and Curran, J., 2014, Independent Expert Scientific Panel – Report on Unconventional Oil and Gas. Scottish Government, Edinburgh. 102pp.

Oakes, D B, and Skinner, A B. 1975. Lancashire Conjunctive Use Scheme Groundwater Model. Special Report, Water Research Centre, TR12. Seymour, K. J., Ingram, J. A., and Gebbett S. J., 2006, Structural controls on groundwater flow in the Permo-Triassic sandstones of NW England. Geological Society, London, Special Publications, v. 263:169-185,doi:10.1144/GSL.SP.2006.263.01.09

Tellam, J.H., and Barker, R.D., 2006, Towards prediction of saturated-zone pollutant movement in groundwaters in fractured permeable-matrix aquifers: the case of the UK Permo-Triassic sandstones. Geological Society, London, Special Publications, v. 263:1-48,doi:10.1144/GSL.SP.2006.263.01.01

Yang, Y., and Aplin, A.C. (2007) Permeability and petrophysical properties of 30 natural mudstones. Journal of Geophysical Research, 112, B03206 (doi:10.1029/2005JB004243). Westaway, R. (2009) Quaternary uplift of northern England. Global and Planetary Change, 68(4), pp. 357-382. (doi:10.1016/j.gloplacha.2009.03.005)

Wilcock, J., and Lowe, D.J., 1999, On the origin of the thermal waters at Bath, United Kingdom: A sub-Severn hypothesis. Cave and karst Science 26 (2): 69 – 80.

Worthington, P F. 1977. Permeation properties of the Bunter Sandstone of Northwest Lancashire. Journal of Hydrology, No. 32, 295–303.

Younger, P.L., 1995, Hydrogeology. Chapter 11. In Johnson, G.A.L., (Editor). Robson's

Geology of North East England. (The Geology of North East England, Second Edition). Transactions of the Natural History Society of Northumbria, Vol 56, (5). pp 353 - 359.

Younger, P.L., 2014, Hydrogeological challenges in a low-carbon economy. (The 22nd Ineson Lecture). Quarterly Journal of Engineering Geology and Hydrogeology, 47 (1): 7 – 27. (doi 10.1144/qjegh2013-063).

Younger, P.L., Boyce, A.J., and Waring, A.J., 2015, Chloride waters of Great Britain revisited: from subsea formation waters to onshore geothermal fluids. Proceedings of the Geologists' Association, 126: 453–465. (DOI: 10.1016/j.pgeola.2015.04.001)

---

## Referee Comment (RC2) · S. Haszeldine (Referee) · 15 Apr 2016

The comment was uploaded in the form of a supplement:
http://www.solid-earth-discuss.net/se-2015-134/se-2015-134-RC2-supplement.pdf

---

## Referee Comment (RC3) · Anonymous Referee #3 · 10 May 2016

Review of

**Hydraulic fracturing in thick shale basins: problems in identifying faults in the Bowland and Weald Basins, UK**

David K. Smythe

**General comments**

My review deals with the overall structure, style and presentation of the article. I am not an expert on seismic interpretation so I leave comments on the scientific quality of this part of the article to other reviewers.

What I can say is that the article is a mixture of science, insinuation and review – most of it the latter. The new research is minimal. The article isn't about what the title says; and the conclusions don't reflect the content of the article either. The piece is not a research or science article, more a detailed magazine article. However it is well written and interesting and probably would be fine as an opinion piece in a specialist magazine, or with suitable editing, in a newspaper.

I recommend rejection.

**Detailed comments**

My first reservation is simply that the paper isn't about what the title says it's about: 'Hydraulic fracturing in thick shale basins: problems in identifying faults in the Bowland and Weald Basins, UK.'

Much of the paper is about regulation and other aspects that are irrelevant. The paper contains many unsubstantiated assertions and irrelevant statements - and comments completely inconsistent with the style of an academic paper. Here are but a few:

*'…In addition, the CBL that was run on the production casing proved that the cementing was inadequate, and that DECC knew this but tried to hide it…'*

*'…EA funding was cut by more than 20% between 2011 and 2015…'*

*'…Celtique also submitted a misleading diagram in each application...'*

*'…However, below the Mercia Mudstone Group the SSG is alleged by the UK Environment Agency to be hypersaline, but may in fact be fresh down to 500 m depth...'*

*'…The Minister of State at the Department of Energy and Climate Change, the UK licensor, wrote to Lord Browne, Chairman of Cuadrilla Resources Ltd., the operator, on 11 May 2012 (Hendry, 2012) to express his concern that the wellbore deformation, which might be linked to the fracking, had been concealed from his officials…'*

*'…Regulation of unconventional energy is split across four main separate agencies or authorities…'*

*'…Guidance for councils on unconventional hydrocarbon planning applications was provided, not by any of the agencies mentioned above, but by the Department for Communities and Local Government (2013). This document was only available on the government website from July 2013 until March 2014. It was then archived with a website redirection to a new website; but the latter (http://planningguidance.communities.gov.uk/) contains no equivalent information…'*

*'…The committee recommended that the disparate regulatory bodies be brought under one central overseeing body, but this has not come to pass…'*

*'…The current UK regulatory system is over-complex and not fit for purpose. Its government has adopted a laissez-faire approach…'*

*'…Central government is currently calling in applications, i.e. to make the final decision itself, on the basis that shale development is part of the so-called 'National Infrastructure', and therefore too important to be left to county councils…'*

All these comments may or may not be true, but they have no place in a scientific paper which should recount research, data, hypotheses and conclusions. With these statements the paper reads more like a piece of investigative journalism or an opinion piece. It might be perfectly good as such. The article would perhaps be better published in a specialist magazine, or with suitable editing, in a newspaper.

Second. The amount of new research or data presented is minimal. With a paper purporting to be about something as important as faults and leakage, some new data and conclusions following modelling or research would really be welcome. The paper is almost entirely a review, and a partial review at that. It could easily be about a quarter of its present length and concentrate only on faults as the title suggests.

Third. The author is right to consider the paper by Llewellyn et al. (2015). In the paper he says '…Llewellyn et al. (2015) prove beyond reasonable doubt that contamination of drinking water was caused by passage of frack fluid and/or produced water in part through the geology'. In fact Llewellyn et al. (2015) were much more circumspect than this in their conclusions. Here is a paragraph from the paper:

*If HVHF fluids did contaminate the water wells, it would be surprising if such contamination were due to fluids returning upward from deep strata, given that (i) this has never been reported (6), (ii) the time required to travel 2 km up from the Marcellus along natural fractures is likely to be thousands to millions of years (31), and (iii) Fig. 6 shows that the Cl:Br ratios in the drinking waters indicate the absence of salts that would be diagnostic of fluids from the Marcellus Shale (e.g., flowback/ production waters). The most likely way for HVHF fluids to contaminate the shallow aquifers would therefore be through surface spillage of HVHF fluids before injection or by shallow subsurface leakage during injection.*

(p. 6329 of Llewellyn, G.T., Dorman, F., Westland, J.L., Yoxtheimer, D. Grieve, P., Sowers, T., Humstone-Fulmer, E. Brantley, S.L. 2015. Evaluating a groundwater supply contamination incident attributed to Marcellus Shale gas development. Proceedings of the National Academy of Sciences).

Finally, in the same way that the title does not reflect the contents of the paper, the conclusion does not reflect what has been examined in the paper either. Taking the first few sentences of the conclusions, for example….

Sentence one. *The USA experience of fracking in shale basins cannot be applied to the UK shale basins, or, for that matter, shale basins anywhere in western Europe, because the geometry of the basins is completely different*. There is no examination of the *geometry of the basins* in the paper at all beyond a few sentences. An examination of this would be interesting and worthy of a paper – and such a paper would have to be substantial to cover

such a lot of ground. But the present paper can't pretend to have shown that the '…geometry of the basins is completely different…'

Sentence two. *The major normal faults which cut through the shale to the surface, a universal feature of the UK extensional basins, but absent in the US shale basins*. There is no serious examination of this topic in the paper. It may be true, but has the author really surveyed the huge US basins enough to establish such a point?  Where is all the evidence?

There are other statements and sentences in the conclusion that could not be said to have been established through the research or discussion presented in the paper.

End of review

---

## Referee Comment (RC4) · Anonymous Referee #4 · 12 May 2016

The manuscript:

Hydraulic fracturing in thick shale basins: problems in identifying faults in the Bowland and Weald Basins, UK by David K. Smythe

is clearly a journalistic/information text on the serious problematis around fracking activities. Its scientific content/contribution is really negligible and the entire manuscript is strongly biased by a rather dogmatic opinion of the author. Furthermore, faults are treated in different ways along the manuscript and this shows the not-complete (to be kind) competence of the author on the subject as well as on the subject (e.g. faults are treated as an ideal element, all equals, independently from their capability to re-activate, change in permeability, or to trigger earthquakes – that the author missed

to consider the magnitude. . ., the assumed straight correspondence between surface mapped faults and their propagation at depth, the real reason why faults are missing in some maps –just compare a 19th century geological map and the equivalent from the 20th century. . .). In this way my opinion is that the manuscript is not acceptable, and I do not see an effective way to suggest a revision: too many different subjects are mixed and trivially treated. An entire book will be necessary to provide an objective and competent discussion in this important environmental task (this might be a reasonable suggestion to the author). Let me (I consider myself an environment protector) enter directly in the subject of environment-fracking. It is really a big environmental problematic. And deserve a serious and competent discussion. Manuscript like this one, with scientific flakes, would just produce the opposite than pretended: It would be very easy, for the companies, to overtake the unproved and dogmatic opinions in this manuscript. This would leave them to continue to operate without control. I am confident that this is not the aim of the author, so my opinion is just a warning: we must be environment protector BUT on a very competent scientific basis to avoid inducing the opposite effect resulting from the unreliability of the environment-protector statements.

---

## Editor Comment (EC4) · F. Rossetti (Editor) · 14 May 2016

Dear Dr. Smythe,

your ms. has been now evaluated by 4 independent reviewers.

Based on the resulting reports (3 negative over 4), I regret to say that we can not go forward with this ms., since too much work is needed to render it potentially suitable for final publication on SE.

My decision is therefore to discourage submission of a revised manuscript.

Yours sincerely, federico rossetti

---

## Author Comment (AC12) · 7 Jun 2016

**Response to Professor Paul Younger (SC20) and Professor Andrew Aplin (RC1) on Fylde groundwater salinity**

**David Smythe**

**Introduction**

The scientific issue here is whether or not the confined aquifer of the Sherwood Sandstone Group (SSG) below the Mercia Mudstone Group (MMG) in the Fylde west of the Woodsfold Fault is saline or not.

**Alleged non-citation of previous research**

Professor Younger writes:

> "*Smythe then proceeds to "examine the Fylde evidence" – though, breathtakingly, this examination does not engage with a single one of the many hydrogeological studies of the Fylde Aquifer published over the decades (including Allen et al. 1997; Barker and Worthington 1973a, 1973b; Brereton and Skinner 1974; Lovelock 1977; Oakes and Skinner 1975; Seymour et al. 2006; Tellam and Barker 2006; and Worthington 1977).*"

I shall first summarise in turn the relevance of the nine papers cited above by Professor Younger.

***Allen et al. 1997.*** I downloaded this paper on 16 Mar 2015, and cited it in my submission to Nottinghamshire County Council (Smythe 2015). It is also present as an orphan reference in my SED paper (but there is no discussion of it), because I had removed several paragraphs from an earlier draft in an effort to reduce the length of the paper. I shall re-insert these paragraphs discussing Cai and Ofterdinger (2014) in my new paper, and quote them here in Appendix A for the record.

***Brereton and Skinner 1974***. I do not have access to this paper, but it is discussed in Sage and Lloyd (1978) and Tellam and Barker (2006), both of which I have consulted. All three papers confine their attention to the SSG aquifer at outcrop east of the Woodsfold Fault (Fig. 1).

***Barker and Worthington 1973a, 1973b; Worthington 1977***.  I have now downloaded the first two of these papers, to confirm their non-relevance to the issue;  the last I had downloaded on 4 February 2014. All three papers discuss the physical properties of the SSG. Figure 1 in each shows that all the samples come from the outcrop east of the Woodsfold Fault. These papers are cited in

Tellam and Barker (2006). The green dashed line (Fig. 1) encompasses all the boreholes discussed by Worthington (1977).

[Figure]

*Fig. 1. Outcrop of the Sherwood Sandstone Group (tan shading) in the Fylde. Two versions of the Woodsfold Fault are shown; the EA 1997 in blue and the earlier BGS map version just to the east in red. Other faults (red lines, teeth on downthrown side where known) are from the BGS sheets. Green dotted line encompasses all the boreholes discussed by Seymour et al. (2006); green dashed line those by Worthington (1977). Light green dots are the observation boreholes used in the Fylde conjunctive groundwater studies (Mott MacDonald 1997, 2009). Orange dots – existing Cuadrilla wells; red dots – proposed Cuadrilla wells. National Grid displayed at 2 km interval.*

**Lovelock 1977**. I do not currently have access to this BGS bulletin, but it is presumably a version of the author's thesis (Lovelock 1972), which I downloaded on 2 August 2013 from the UCL website (http://discovery.ucl.ac.uk/1381922/). Field samples of the SSG (known then in the Fylde area as the Bunter Sandstone) were collected from outcrop between Preston and Garstang (i.e.east of the Woodsfold Fault, Lovelock 1972, fig. 14), but no core was studied from boreholes in the Fylde area (Lovelock, op. cit. table 1). Once again, the thesis has no relevance to the issue discussed here.

**Oakes and Skinner 1975**. I do not have this is a forty-year old reference (slightly mis-cited by Professor Younger), to early mathematical modelling of the Fylde reservoir, but it can hardly be relevant, since I have consulted the much more extensive study which superseded the 1975 model (Mott MacDonald 1997), together with its update and conversion (Mott MacDonald 2009); all downloaded on 6 August 2015.

In Mott MacDonald (1997) the Oakes and Skinner study, referred to by them as Water Research Centre (1975), states:

> *"In the early 1970's, the first extensive pumping tests of the aquifer were undertaken by the former Fylde Water Board and Lancashire River Authority. The data collected during these tests were used in the development of a groundwater model of the Fylde aquifer (Water Research Centre (WRC), 1975)."*

**Seymour et al. 2006.** I downloaded this on 4 February 2014, and discussed it in my Roseacre Wood objection submission (Smythe 2014), p.26:

> *"The ES states:*
>
> *"In the Fylde area the north-south trending faults are reported to act as barriers or partial barriers to flow in the Sherwood Sandstone" [para. 115, p.302]*
>
> *This statement is supported by reference to Seymour et al. 2006. But it is misleading. The Seymour et al. study divides the Fylde into three areas. In the northern and central areas the predicted hydrogeological model matches well the observed data without modification. These are the two area nearest to the Cuadrilla drillsite. For the third, southerly, area it was found that the numerous N-S faults (5 faults within an E-W distance of 1.5 km according to the cross section of figure 4b of Seymour et al.) did affect the flow in the observation borehole T74. A match of predicted to observed flow was achieved by lowering the flow across the faults to 0.2 m/day within the Sherwood Sandstone Group aquifer, and by setting the regional hydraulic conductivity to 3 m/day, the regional value. So the N-S faults here act as partial barriers, **but**

***not as seals.”*** [emboldening in original].

***Tellam and Barker 2006.*** I downloaded this UK-wide review of the hydrogeological properties of the SSG on 4 February 2014. The Fylde is only mentioned in passing, either in the context of discussing geophysical survey techniques, or in summarising previous work on regional flow systems where faulting may be important, citing Seymour et al. (2006).

**Discussion of the 'missing' citations**

All nine of the citations that Professor Younger accuses me of neglecting are either irrelevant and/or outdated. I was previously aware of five of them; I have downloaded two more to confirm their irrelevance, as suspected, and the two remaining, over 40 years old, have been superseded by later work. **None of the work cited by Professor Younger has any bearing on the hydrogeology of the western Fylde, west of the Woodsfold Fault (Fig. 1).** They all concern the SSG aquifer at outcrop. To demonstrate this I have plotted on Figure 1  two green-bordered polygons, each showing the areas encompassed by the boreholes studied in Worthington (1977) – dashed line, and Seymour et al. (2006) – dotted line.

Unlike Professor Younger, I do not believe in citing works which have no relevance to the question at hand. That is why I did not cite these papers.

It is noteworthy that, despite his blunderbuss approach to citations, Professor Younger fails to cite either Mott MacDonald (1997) or its update (Mott MacDonald 2009). These are much more detailed studies, but once again, these hydrogeological modelling studies have no direct relevance to the confined SSG aquifer west of the Woodsfold Fault. However, they do reveal the Environment Agency's thinking on the transmissivity of faults. This is discussed below.

In conclusion, Professor Younger has erected a facade of emphatic assertions, buttressed by numerous citations; but when one examines his story  it has no more substance than a cowboy township erected for a spaghetti western – there is nothing behind it. Is Professor Younger seeking to deceive the reader (which I do not believe) or is he lazily recycling half-remembered and mostly obsolete references from his youth? In short, it demonstrates poor scholarship in his own area of expertise – groundwater.

**Discussion**

In his second comment (SC20) ProfessorYounger has not responded to my criticism (AC8) of his first comment (SC6) under the heading 'Potable groundwater below the Mercia Mudstone Group (MMG)'. I showed that his assertion concerning the impossibility of fresh water in the confined SSG aquifer is wrong, both in logic and in the (unspecified) examples he cites.

The groundwater modelling of Mott MacDonald (1997) and its update (Mott MacDonald 2009) for the Environment Agency concern only the unconfined SSG aquifer at outcrop; although the finite-element modelling grid does locally extend west of the Woodsfold Fault by up to 3 km, the elements here are purely for boundary condition control. In the modelling a number of observation boreholes are used to constrain the modelling. These are shown as green dots in Figure 1. Note that they all, with one exception, lie east of the Woodsfold Fault. So once again, the modelling tells us nothing about the confined aquifer west of the Woodsfold Fault.

Mott MacDonald (1997) states:

> *"[1] Flow to the west, across the Woodsfold Fault to the Kirkham Basin is considered to be very low. There is no detailed piezometric evidence to suggest otherwise. [2] In addition, the overlying Mercia Mudstone has a very low vertical permeability and thus there is unlikely to be any significant vertical exchange of water between the Sherwood Sandstone and Mercia Mudstone. [3] Consequently, there is unlikely to be any significant lateral flow from the Fylde aquifer to the west since there is no obvious point for this inflow to discharge to. [4] This boundary was, therefore, defined as a no flow boundary."* [numbering added].

So from [1], the EA asserts, based on no solid evidence, that the flow across the Woodsfold Fault will be low. Next, it assumes [2] that there will be little or no vertical flow – but this assumption ignores the presence of faults cutting the MMG. These could be transmissive pathways, particularly when one considers the stress regime in the uppermost 300 m below ground level.

The EA cannot find a discharge [3] for the flow, if present, but this again ignores the presence of faults, plus the many springs and sources in the area. Lastly, since the Woodsfold Fault is *defined* as a no-flow boundary [4], the lack of westward flow in the model cannot be used as an argument to prove that there is no westward flow.

Neither Professor Younger nor Professor Aplin have commented on the false inference drawn by the EA, based on samples from the Kirkham well taken at MMG depths, that the confined aquifer west of the Woodsfold Fault is saline. The former draws instead on misleading and erroneous analogies

with other similar settings, as I have already pointed out (SC6). The latter dismisses all this as *"some rather random salinity data"* (RC1). What does Professor Aplin mean by 'rather random'? Is it that the EA has chosen borehole data at random? Have I, on the other hand, chosen the Kirkham well at random? If he wishes to debate seriously the evidence for the level of salinity in this aquifer he needs greater precision in his argument.

**Conclusions**

Neither Professor Younger nor Professor Aplin seem to be capable of grasping the point that the potability, or otherwise, of the groundwater in the confined aquifer west of the Woodsfold Fault is not well constrained, and that the historical evidence of deep boreholes abstracting from the confined SSG suggests that it may well be fresh.

Professor Younger reiterates his serious misunderstandings, combined with misquoting of the literature, including:

- The belief that fluids migrate downwards in a fracked and faulted shale setting, in the face of six independent quantitative modelling studies which suggest the opposite,

- The assertion that the confined aquifer below the western Fylde can only be saline, based on false analogies with similar UK settings,

- The quotation of irrelevant previous work, presumably with the aim of browbeating the editors or other non-specialist readers that he has a superior grasp of the problem than I.

Professor Aplin's review reveals that he too does not grasp the nub of the problem. His report cannot be considered valid or reliable on this point.

It would be constructive if, rather than relying on dogmatic and erroneous assertions, one or both of these two critics would conduct a research project to extend the EA modelling westwards, in parallel with the collection and analysis of waters from the boreholes and natural sources in the area, to test whether or not there is any significant flow into the confined aquifer across the Woodsfold Fault, and whether or not it is potable.

**References**

Allen, D. J., L. J. Brewerton, L. M. Coleby, B. R. Gibbs, M. A. Lewis, A. M. MacDonald, S. J. Wagstaff, and A. T. Williams 1997. The physical properties of major aquifers in England and Wales, *Br. Geol. Surv. Tech. Rep. WD/97/34*, Environ. Agency R&D Publ. 8, Keyworth, Nottingham.

Barker, R D, and Worthington, P F. 1973a. The hydrological and electrical anisotropy of the Bunter Sandstone of Northwest Lancashire. *Quarterly Journal of Engineering Geology*, 6, 169–175.

Barker, R D, and Worthington, P F. 1973b. Some hydrogeophysical properties of the Bunter Sandstone of northwest England. *Geoexploration*, 11, 151–170.

Brereton, N.R. & Skinner, A.C. 1974. Groundwater flow characteristics in the Triassic Sandstone in the Fylde area of Lancashire. *Water Services,* 78, 275-279.

Cai, Z. and Ofterdinger, U.: Numerical assessment of potential impacts of hydraulically fractured Bowland Shale on overlying aquifers. *Water Resources Research*, 50, 6236–6259, doi:10.1002/2013WR014943, 2014.

Griffiths, K. J., P. Shand, and J. Ingram, 2003. *Baseline Report Series: 8. The Permo-Triassic Sandstones of Manchester and East Cheshire.* British Geological Survey Commissioned Report. Ref. CR/03/265N. BGS, London, UK. 51pp.

Jones, H K, Morris, B L, Cheney, C S, Brewerton, L J, Merrin, P D, Lewis, M A, MacDonald, A M, Coleby, L M, Talbot, J C, McKenzie, A A, Bird, M J, Cunningham, J, and Robinson, V K. 2000. The physical properties of minor aquifers in England and Wales. *British Geological Survey Technical Report*, WD/00/4. 234pp. Environment Agency R&D Publication 68.

Lovelock, P.E.R. 1972. Aquifer properties of the Permo-Triassic sandstones of the United Kingdom. PhD thesis, Department of Geology, University College London, 645 pp.

Lovelock, P.E.R. 1977. *Aquifer properties of the Permo-Triassic sandstones of the United Kingdom.* Bulletin of the Geological Survey of Great Britain, No. 56, 50.

Mott MacDonald 1997. Fylde aquifer/Wyre catchment water resources study final report 38436BA01, 176 pp + 87 figs.

Mott MacDonald 2009. Fylde Model Conversion and Update Final Report 250819/01/C, 78 pp + 41 figs.

Tellam, J.H., and Barker, R.D., 2006, Towards prediction of saturated-zone pollutant movement in

groundwaters in fractured permeable-matrix aquifers: the case of the UK Permo-Triassic sandstones. *Geological Society, London, Special Publications*, v. 263:1-48, doi:10.1144/GSL.SP.2006.263.01.01.

Oakes, D. B, and Skinner, A. B. 1975. *Lancashire Conjunctive Use Scheme Groundwater Model. A Mathematical Model of the Triassic Sandstone Aquifer of the Fylde Area.* Technical Report, Water Research Centre, TR12.

Sage, R.C. And Lloyd, J.W. 1978. Drift deposit influences on the Triassic Sandstone aquifer of NW Lancashire as inferred by hydrochemistry. *Q. Jl Eng Geol.* 11, 209-218

Smythe, D.K. 2014. *Planning application no. LCC/2014/0101 by Cuadrilla Bowland Limited to drill at Roseacre Wood, Lancashire: Objection on grounds of geology and hydrogeology.* 16 September 2014, 47 pp.

Worthington, P F. 1977. Permeation properties of the Bunter Sandstone of Northwest Lancashire. *Journal of Hydrology*, 32, 295–303.

**Appendix A**

**[Draft material removed from the submitted version of the SED paper]**

There is an error in the nomenclature of the sedimentary succession, which has been taken from de Pater and Baisch (2011) and which in turn comes from Cuadrilla's interpretation of the Preese Hall-1 well log (Turner, 2012). The error has since been corrected in the released version of the well log.

The two formations immediately below the Mercia Mudstone Group (MMG), for which hydraulic conductivities and other hydrogeological properties are used by Cai and Ofterdinger, are noted as the Sherwood Sandstone Group (SSG) and the St Bees Sandstone (SBS). The one-dimensional 3000 m–thick layer-cake hydrogeological model used by Cai and Ofterdinger has an effective grouping of elevation and hydraulic conductivity as follows, from top to bottom, increasing downwards from ground level at 0 m to the top of the Bowland Shale, as shown in Table 1:

**Table 1. Hydraulic conductivity values used by Cai and Ofterdinger**

| Group/Formation | Depth (m) | Thickness (m) | $K_h$ (m s$^{-1}$) | $K_h$ (µm s$^{-1}$) |
|---|---|---|---|---|
| Mercia Mudstone Group | 0-200 | 200 | $1.0 \times 10^{-7}$ | 100 |
| Sherwood Sandstone Group | 200-400 | 200 | $1.2 \times 10^{-5}$ | 10,000 |
| St Bees Sandstone Formation | 400-1000 | 600 | $8.1 \times 10^{-7}$ | 810 |
| Manchester Marl Formation | 1000-1100 | 100 | $1.0 \times 10^{-8}$ | 10 |
| Collyhurst Sandstone | 1100-1250 | 150 | $7.9 \times 10^{-5}$ | 79,000 |
| Lower Coal Measures | 1250-1300 | 50 | $1.7 \times 10^{-9}$ | 1.7 |
| Millstone Grit Group | 1300-2000 | 700 | $7.9 \times 10^{-8}$ | 79 |

But the SBS is a formation within the SSG and, furthermore, is not recognized in the Fylde and East Irish Sea Basin. The logging tools were run only below 580 m (1900 ft), within the supposed SBS, so there is no evidence other than drill cuttings for subdividing the arenaceous succession into two parts. The well completion log as depicted by Turner (2012) has been mis-interpreted. This error in the basic basin geology by Cuadrilla, the operator and licensee, would be neither here nor there in itself, but it has led to an inappropriate hydraulic conductivity being assigned to this succession. The SBS is only found in West Cumbria, some 80-100 km to the north, where the physical property

measurements were taken (Allen et al., 1997). The two rock formations in the model should therefore have been merged as the single SBS, without the incorrect subdivision, and assigned the physical properties of that group. In the model this group should extend from 2000 to 2800 m elevation, and not just the uppermost 200 m, as wrongly labelled from the original well log. This is important, because the hydraulic conductivity of the SSG is 15 times higher than that of the SBS.

The sources of the hydraulic conductivity values assumed in the modeling deserve discussion. Firstly, the "*hydrogeological properties*" of the Bowland Shale are taken from Marcellus Shale. This presumably includes the hydraulic conductivity, although the source of the value used is not specified in table 1 of Cai and Ofterdinger. The modelled value of the Millstone Grit ($7.9 \ 10^{-8}$ m s$^{-1}$, after correcting the misprint in Cai and Ofterdinger's table 1, row 7, column 4) is close to the mean value from drill stem tests in the East Midlands (Jones et al., 2000), but the range of measurements quoted by Jones et al. spans over three orders of magnitude, if one includes the nearest samples taken from outcrop in the West Pennines, some 20 km or more east of the Bowland Basin. The value adopted for the SSG of $1.2 \ 10^{-5}$ m s$^{-1}$ from core measurements is somewhat lower than the mean from borehole pumping tests in the Fylde (Allen et al., 1997); the latter value of $6.1 \ 10^{-5}$ m s$^{-1}$) should preferably have been used as it reflects more accurately the bulk fracture flow rather than the intergranular flow measured in core.

The 100 m of Manchester Marl Formation is the relatively low-permeability formation which is supposed to provide the main barrier to upward flow; however, it is clear from Figure 2 that this barrier is cut by faults which can have a throw greater than the layer thickness.

---

## Author Comment (AC13) · 7 Jun 2016

**Reply to anonymous referee RC3**

**David Smythe**

Here is my response to the detailed comments provided by referee no. 3 on 10 May 2016.

The referee writes:

> *"the paper isn't about what the title says it's about: 'Hydraulic fracturing in thick shale basins: problems in identifying faults in the Bowland and Weald Basins, UK.'"*

I disagree; pages 5 to 23 are concerned in detail with the problems of identifying faults in the two basins. But the referee presumably skipped over this part – the heart of the paper – because, as this referee admits, he/she is not an expert in seismic interpretation. The heart of the paper is wrapped up in the context in which the faults are studied, that is, 'hydraulic fracturing in thick shale basins', which is the first part of my title. Therefore, in my view, the title correctly reflects the content.

The referee states, followed by ten quotations taken from my paper:

> *"Much of the paper is about regulation and other aspects that are irrelevant. The paper contains many unsubstantiated assertions and irrelevant statements - and comments completely inconsistent with the style of an academic paper."*

It is a moot point as to whether discussion of the (perceived) failures of regulation and/or distortion of technical evidence by exploration companies should find a place in a scientific journal like SED or SE. Does this mean that such shortcomings should never be admitted or discussed? Should such a discussion be left to sociologists? Should they be left to the arena of planning inquiries and the courtroom? How could highly technical criticisms be satisfactorily dealt with in, say a newspaper or magazine article? My view is that we earth scientists have a duty to point out these failures; if not, the exploration companies may obtain licences and planning permissions based upon inadequate or even false data.

To take as an example the Celtique Energie case histories I quoted (section 4.3); I provided numerous instances where the company's two planning applications in West Sussex were misleading both the county council and the general public. Is such criticism, highly technical in places, to be omitted? For example, I stated how and why the faults in the seismic section published by Celtique appeared to have been removed by reprocessing. I included it as an example of the 'problems in identifying faults', which is part of the title of my paper. If the referee can suggest a more appropriate peer-reviewed scientific forum for the debating of such points I shall be pleased to take

his/her advice.

The referee mentions my discussion of Llewellyn et al. (2015), agreeing that it is an appropriate topic for review. But he/she has not read my long reply (AC3) to Dr Engelder (SC4) in which I have taken the interpretation much further with, *inter alia*, the use of new data on the location of the offending horizontal well. My new interpretation is, in effect, a paper in its own right, and I shall be submitting this discussion and reinterpretation for publication elsewhere.

Lastly, the referee accuses me of failing to describe the geometry of the US shale basins, and the absence of through-going normal faults therein. I have thought long and hard about how and whether to publish this information in orthodox peer-reviewed form. The referee concludes:

> *"It may be true, but has the author really surveyed the huge US basins enough to establish such a point? Where is all the evidence?"*

Yes, I have indeed surveyed (by desk study) the US basins. I have provided a fuller explanation in my response (AC7) to Dr Verdon's comment (SC7). Here is a summary. I spent the equivalent of two months' full-time research on this topic, reaching the conclusion that such faults do not occur in the areas of the shale basins that are being exploited. But most of the relevant information is to be found online, in informal sources like company reports. In practice it would be impossible to seek out all these sources to ask permission to reproduce their maps, cross-sections and seismic reflection examples. I now propose to make my findings public in an informal way, such that the data may be periodically updated as new information emerges. The data sources will be acknowledged, and if any third party objects to the acknowledged re-use of their images then the offending examples can be removed.

I encapsulated my US study, which comprises an essentially negative result (am I expected to publish blank maps showing an absence of outcropping faults?), in the phrase 'foreland basins', which I trust this referee understands. Built in to this observation is the style of faulting, which contrasts fundamentally with that in extensional basins.

In conclusion, this referee appears to prefer an uncontroversial, purist approach to academic publishing. My contention, in contrast, is that we earth scientists – at any rate in basin researches – cannot and should not ignore the real world of grey literature, exploration licences, commercial and/or confidential data, and so on.

---

## Author Comment (AC14) · 7 Jun 2016

**Response to Professor Paul Younger (SC20) on faulted limestone systems**

**David Smythe**

**Introduction**

The issue here concerns two case histories of potential targets for fracking of shale in faulted karst and/or limestone terrains. In my SED paper I wrote:

"In 2011 the University of Montpellier-2 published two explanatory documents on the risks of potential fracking in the south of France (Arnaud et al, 2011; Séranne et al., 2011), following the granting of shale exploration permits in the region a year earlier. They drew attention to the crucial role that faults play in the groundwater circulation system (Bicalho, 2010; Bicalho et al., 2012)."

Professor Younger initially commented (SC6):

"... the two papers he cites in support of his claims over fault permeability (Bilcalho 2010; Bilcalho et al. 2012) both relate to karstified limestones – the most extremely permeable of all natural hydrogeological systems, in which fault apertures are widened by dissolution of the soluble wall-rocks!" [NB the author cited by me is Bicalho, not Bilcalho]

I responded as follows (AC8)

"I am well aware that the French work I cited concerns – in part - karstified limestones, an extreme kind of rock formation, hydrogeologically speaking. However, it is demonstrated there that deep pathways down to greater than 2 km depth involving faults do exist, and limestone plays little or no part in the flow systems at depths greater than 1 km. The studies show that upward fault pathways exist through Lias shales, which were the target of a Total exploration licence (since annuled) for fracking."

**The Languedoc example**

Professor Younger has rejoindered (SC20):

"Having been forced by my comments to acknowledge that he misleading [sic] cited irrelevant papers concerning karstified limestones, Smythe clings to the wreckage of his argument by referring to (though not, I note, citing any literature to support) deep circulation systems that are very definitely hosted by limestones." Note that I cited four publications, not two, concerning the deep water circulation system in the eastern Languedoc region, which was formerly under licence to Total for unconventional shale exploitation. Since Professor Younger has evidently not actually read these cited works, it is worthwhile setting out here for the general reader a little more detail, to show why these French studies are important and relevant.

Figure 1 shows the stratigraphic column of the region, with the two main shale sequences highlighted (Séranne et al. 2011). The mid and light blues of Bajocian to Portlandian age (Jurassic) are the karstified limestones in the cross-section shown in Figure 2, in which these rocks are depicted by a white brick pattern (Bicalho 2010). It is differentiated thus because it is the main aquifer of the system.

---

## Author Comment (AC15) · 7 Jun 2016

**Final author comments on '*Hydraulic fracturing in thick shale basins: problems in identifying faults in the Bowland and Weald Basins, UK*'**

**David Smythe**

**Introduction**

My paper has aroused far more discussion than any other paper ever published in SED; therefore it is worth recording why this has happened, and what I propose to do with the manuscript.

In a covering letter I explained why I was submitting my manuscript to SED:

> *"I am submitting this paper to SE, rather than to a journal with a track record in applied geophysics, because I like your transparent reviewing and open access policies. However, I am aware that your publication history to date and the composition of your editorial board do not appear to include explicit expertise in applied geophysics or petroleum geoscience.*
>
> *I considered submitting to HESS, but although my paper crosses several disciplines, on balance it is more tectonic than hydrogeological. So if you feel that the paper is not for SE then I won't be offended; on the other hand it could be regarded as an opportunity for you to expand your coverage. In any case I am confident that this paper will gain a wide readership."*

**The referees**

I unwittingly gave Dr Rossetti, the topical editor, a lot of work, because he found, as I anticipated, that it was very difficult to find referees. For the record, I had written in my covering letter on submission:

> *"It is difficult to find suitably qualified referees. Few potential academic referees are familiar with the details of hydrocarbon exploration, even conventional, and in addition there is a lot of controversy around proposed fracking in the UK. There are several UK academics who consider themselves to be experts in the field of unconventional hydrocarbons, but they cannot be trusted to write an unbiased report, because they have close links to industry and are actively promoting fracking. I was even libelled in the national press by one of them 18 months ago."*

Also, for the record, I proposed five knowledgable and impartial referees, declaring my personal connections as follows (names and affiliations have been redacted to preserve their anonymity):

> *"However, I propose four academic refereees and one ex-industry referee for you to choose from, and each of whom I think I can trust to write an expert and fair report:*

*A is Professor of ... at .... He has published many papers and has obtained many industry grants. He will be familiar with the applied geology and exploration content of my paper, and is knowledgable about the stratigraphy of the basins involved.*

*B is now retired, he worked mainly in the ... oil industry, rising to become a Vice-President of ... , one of Canada's major oil companies. However, he has also published 15 papers on geophysics and tectonics. He is familiar with seismic reflection data, and with UK tectonics.*

*C is director of the ... Group and the ... . His research focuses on computer simulation of complex fracture models. As an expert on US exploration ... he will be familiar with the modelling and well logging discussion in the paper, if not with seismic reflection and UK geology.*

*D has published several papers on fracking. He is a petroleum geologist who formerly worked in the oil and gas industry. He heads the ... group on fracking, industry-funded.*

*E is Professor of Hydrogeology at ... [he/she] is involved in shale gas exploration risks ... and will be familiar with the hydrogeological aspects of the paper, if not with the details of UK geology."*

I also declared personal links to four of the five suggested reviewers proposed above:

*"P: I have talked to him by phone about faulting and fracking.*

*Q: [We] have given a joint public lecture in ... about shale gas exploration.*

*R and S: ... friend and former colleague at ... up to [n >17] years ago."*

I did not name anyone whom I did not wish to be a reviewer – an option often offered in manuscript submission to other journals. It appears that eight potential reviewers were asked, but declined. Dr Rossetti eventually found four reviewers.

**The review process and the reviews received**

In view of the conflicting verdicts of the first two reviewers (Professors Aplin and Haszeldine), Dr Rossetti solicited two more reviews. These were anonymous. I do not need to comment on Professor Haszeldine's report, which was both constructive and positive.

Professor Aplin's report raised the question of why I had not published the evidence for the lack of through-going faults in the US shale basins. I answered this in my response to Dr Verdon (AC7). He confused my discussion of the "*problem of pre-existing faults*" of section 1.1 with fracking-induced fractures – a different topic. I have already commented (AC10) on his misunderstanding of the evidence for the confined aquifer below the western Fylde being fresh or saline, and why it is significant. He did not read my revision (AC3) of the evidence for drinking water well contamination in Bradford County.

The third, anonymous, referee (RC3) concentrated on the discussion of regulation, and why he/she thought that this has no place in a science journal. That may be a valid point of view, but one with which I disagree.

I regret to say that the fourth referee, also anonymous, provided a short report (330 words) which is practically incomprehensible, even when one attempts to re-interpret the very poor English. But the gist of the review seems to be that a whole book would be needed to cover properly the topics I have tried to cover initially in 17,000 words and ten figures.

In conclusion, the three negative reports, taken together, are hardly constructive or helpful.

**Other comments received**

Comments can be constructive or destructive, but in either case can often be useful for indirectly highlighting topics or conclusions that the commentator does not discuss. Such omissions would tend to imply tacit agreement, unless the author has explicitly stated that he/she would not focus on certain topics.

In general, I conclude from the comments that the following conclusions, assertions, or statements have passed unchallenged:

- The lack of through-going faults in the US shale basins (section 1.1, but with the proviso that the evidence needs to be published).

- The orders of magnitude geometric differences in the US *vs.* the UK shale basins.

- Flaws in the Halliburton frack upward growth study.

- The UK history of long-reach conventional wells and of fracked wells.

- Geology and exploration history of the Fylde.

- Various details concerning errors or omissions by the operators.

- The Paddockhurst Park Fault cutting the Balcombe-1 well.

- Conclusions on better regulation and improved geophysical methods.

The following conclusions, assertions, or statements were challenged:

1. Re-interpretation of the triggered fault at Preese Hall-1 (sections 3.5, 3.6).

2. Hydrogeology in general, and of the Fylde in particular.

3. Recognition of a deeper fault in Balcombe-2z.

4. Parts of my fault modelling study review.

5. Bradford County case history of Llewellyn et al. 2015.

6. Image manipulation or alteration.

7. UK regulation (section 6.2).

I omit from this list some general comments, such as claims that the paper is unscientific, or that the title is misleading, and I also omit minor points of criticism. The points of contention listed above were challenged as follows.

Dr Westaway took issue with me, primarily about the significance of the fracking-induced seismicity at Preese Hall-1, Lancashire. He quoted from his paper, newly published in January 2016, which I showed was immediately obsolete (AC1) because it failed to take into account the revised stratigraphy of the well, released by DECC in April 2015. He had failed to obtain the released data himself, whereas I had used them. He has produced, over the last year or two, various interpretations of the fault that slipped. He seems to disagree with my geometrical analysis suggesting that the well was bored right through the fault; but then, latterly, claims that he thought of it first.

Dr Westaway quoted a defamatory article from the UK tabloid newspaper *The Daily Mail*, masquerading as a scientific scitation ('Seamark 2014'). The editor asked him to remove the offending reference, which was a web link, in which his colleague Professor Younger had defamed me. He did so in part, but later on re-inserted the offending pseudo-citation. He then went on to criticize the journal itself for having published my discussion paper in the first place. This earned him a public rebuke (SC19) from Professor Fabrizio Storti, the Editor in Chief.

Dr Verdon discussed (*inter alia*) the problem of the location of the earthquake-triggering fault at Preese Hall-1. He also made several *ad hominem* comments, and challenged my independence on the ground that I am funded by objectors' groups in England. My response to the latter is that I shall declare my earnings from these sources, which average out over the last three or four years at well under £1 per hour.

Mr Clarke of Cuadrilla Resources, the operator in Lancashire and at Balcombe (Weald) tried to defend his company's interpretation of the position of the earthquake fault at Preese Hall-1, in which

the fault location avoid passing through the well, as I propose. He also challenged my interpretation of the possible fault cutting the Balcombe-2z horizontal wellbore. He claims that my supposed fault is merely an artefact of drilling through a cement casing shoe. I had written:

> *"It is possible, but unlikely, to explain the repetition by assuming that two separate logging runs were made and then poorly spliced together; but an alternative and more plausible explanation is that the wellbore went through a normal fault with a downthrow to the east (well-head side)."*

I accepted Mr Clarke's explanation in part, that the drilling out of the shoe might go some way towards explaining the apparent fault, but that the data still suggest a poor splicing of two drilling runs. I invited Mr Clarke to supply some more data to resolve the point, but he has declined to do so.

Professor Younger challenged me on hydrogeological matters. But he commits no less than four fundamental errors (for details see my responses to his comments SC8 and SC20):

- The belief that fluids migrate downwards in a fracked and faulted shale setting, in the face of six independent quantitative modelling studies which suggest the opposite,

- Failure to understand that the subsidiary Lez aquifer system at 1200-3000 m depth in SE France has nothing to do with the primary shallower limestone-hosted aquifer system, but demonstrates that deep water flows upwards along faults cutting the shales which were a fracking target.

- His generalised assertion that the confined aquifer below the western Fylde can only be saline, which I showed was based on false analogies with similar UK settings.

- The quotation of irrelevant previous work on the Fylde aquifer, presumably with the aim of browbeating the editors or other non-specialist readers that he has a superior grasp of the problem than I.

Several comments were made about my mini-review of prior work on modelling fluid flow up faults in a shale setting. I omitted to give proper attention to the recent Birdsell et al. (2015) paper, and shall give it due credit when I next have reason to review the literature.

Several commentators objected to my interpretation of the important Llewellyn et al. (2015) case study in Bradford County, Pennsylvania, saying that I had misconstrued the authors' own conclusions. Indeed I had done so, believing that if well casing had not lost integrity then the methane observed to contaminate the drinking water wells had to have emerged either at the far end of the casing (at the fracked shale) or else at the wellhead. However, I pursued this problem further,

including obtaining the detailed well plans in the area of interest. My new interpretation (AC3), which is in effect a paper in its own right, suggests that the methane did indeed come from fracked Marcellus shale, passing up the fault zone identified by Llewellyn et al.

Mr Andrew Kingdon, petrophysicist at the British Geological Survey, questioned my use of certain images. He also implied that I had withheld some data from an imaging tool used at Preese Hall-1. In fact I did not have access to these data. I refuted his allegation that my interpretation of the published seismic image through the well was ambiguous. He asserted that I had exaggerated the magnitude of fault throw in my discovery of the fault cutting the higher section of Balcombe-1. Again, I showed that his allegation was false, by providing a new, more detailed image. The fault throw is indeed 10 m as stated in my manuscript, and not the 6 m claimed by Mr Kingdon. He has not withdrawn his allegation.

Professor Aplin dismisses my recognition of the faulting at Balcombe-1 that: "*even faults with significant throws may not be visible on old, low quality 2D seismic lines, but that they can be interpreted from detailed log and stratigraphic data. Sound - but hardly novel.*" Here he fails to observe two important points; (1) My precise correlation of shales on gamma ray and sonic logs between two wells nearly 15 km apart, and correlating wiggle-for-wiggle down to sub one metre resolution, has probably never been achieved before; and (2) the operator Cuadrilla failed to observe this fault, having not bothered to consult the published geology maps. Even though Professor Aplin is an expert in aspects of shales, his experience of well log interpretation and tie-in to seismic reflection data appears to be rather limited. His review suggests a reluctance on his part to criticise sub-standard technical work by UK operators such as Cuadrilla, declaring, instead, that my work is an "*invective-strewn commentary*".

Some commentators, and referee no. 3, thought that my criticisms of the failures of UK regulation were inappropriate, either because this sort of comment has no place in a scientific article, or because I was casting aspersions on the competence of scientists at the Environment Agency.

**Discussion**

In view of the three negative reviews out of the four received, Dr Rossetti had little option but to reject the manuscript "*since too much work is needed to render it potentially suitable for final publication on SE*". I concur with his view.

Even if I had been offered the opportunity to revise the paper, it would have turned out to be far too long once I had added all the necessary amendments and additional discussion. In retrospect I can see that my manuscript was over-ambitious in trying to cover five separate topics. I propose to deal

with these topics as follows, taking into account the valid review comments:

(1) *Faulting in the US shale basins*: I have explained already (reply to Dr Verdon, AC7) why it is almost impossible to publish a full accurate study in a peer-reviewed journal. I propose to put my findings into a web article, properly researched and with all the sources cited, of course. This will avoid the impracticable task of seeking permission to reproduce dozens of maps and cross-sections. Furthermore, the web page can be updated or corrected as required.

(2) *Faulting in the Bowland Basin and the Weald, England*: I shall resubmit this part of my manuscript as a new paper to SED, on the ambiguity of fault interpretation. It will cite the fault modelling studies of shale basins, while not attempting to review them in any depth.

(3) *Summary review of fault modelling studies in shale basins*: I shall leave this, updated to include the Birdsell et al. (2015) study, as a web page. A very brief version will be included in (3) above.

(4) *Bradford County case history*: My second reply (AC3) to Dr Engelder was proposed as a supplement to a revised paper for SE. However, it can now form the basis of a new stand-alone paper to be published elsewhere.

(5) *Regulation of unconventional development in the UK*: This review and discussion is probably best separated out, for submission to an environmental health journal.

**Unintended consequence of publishing the discussion paper**

I hold a lifelong Honorary Senior Research Fellowship at the University of Glasgow, which entitles me to online access *via* the to the journal and research database. About two days after I published the SED paper online, on 27 January 2016, my habitual online access, uninterrupted since my retiral in 1998, was terminated without warning or explanation. At the time of writing I can only conclude, in the absence of any credible explanation from the University Court (the governing body of the university), that my views on fracking, and in particular the publication of the present manuscript, are the reason for the termination. The Secretary of the Court had previously written to me in 2014, expressing concern that "*the views which you have expressed, particularly on the subject of shale gas, are not consistent with work which is currently being undertaken at the University.*" I am currently engaged in discussions with the University Court, and have instructed lawyers to act for me. This action by the university is clearly aimed at silencing me, for without such access I cannot properly continue my research in *any* field, not just that of unconventional oil and gas development. Professor Paul Younger, one of my critics, is a member of the Court.

Professor Younger was quoted in *The Times* (31 October 2014) as saying, *à propos* of the cancelled disinvestment in fossil fuel shares held by the University of Glasgow, that "*the new statement would*

*help to repair relationships with oil and gas companies that funded research*." So my *alma mater* and former employer seems to be more concerned about maintaining good relations with the oil and gas companies that fund its research than with permitting free and open debate. I find this attitude disappointing and regrettable.

**Conclusions**

I withdraw the paper with immediate effect. I thank Dr Rossetti, the topical editor, for his hard work and forbearance over the last four months. The discussion paper, including all the accompanying comments and replies, remains available online in perpetuity.

I believe that the review methods used in SED and SE point the way to good peer-reviewed scientific publication; that is, openly attributed discussion of a manuscript published online. The next development should be to require, in addition, that all formal reviews be signed and not anonymous, but as Dr Rossetti has pointed out to me, it is already difficult to find reviewers. I look forward to submitting to SED a new slimmed-down and better focussed paper on the ambiguities of faulting in UK shale basins.

---

## Author Comment (AC2)

**Conjecture and refutation; author's response to Dr Engelder**

**Introduction**

I thank Dr Engelder for commenting on my discussion paper, not least for what he has not written than for what he has written. He is an expert of long standing on the geology of Appalachian Plateau, and, no doubt, of many other regions of the USA. So I am pleased that he has not sought to question my summary synthesis of the structural differences between the US and the UK shale basins (Section 1 and Figure 1), and my conclusion that through-going faults are essentially absent in the former.

**Philosophy, advocacy and agendas**

Dr Engelder starts his critique by misapplying Gödel's incompleteness theorem to science in general. These are two formal theorems (not one, as Engelder quotes) in mathematical logic, with application to mathematical philosophy. Their influence may extend from arithmetic into computing science, but they do not have any relevance to the epistemology of the physical sciences, a field which I have had a fifty-year interest (I have just counted six of Karl Popper's books on my bookshelves behind where I am writing this, among the two dozen or more books I possess - and have read! - on the philosophy of science). Karl Popper did the fundamental work on conjecture, refutation, and what is meant by a testable (i.e. falsifiable) scientific hypothesis. His classic is *The Logic of Scientific Discovery* (published in German in 1939; English translation publ. 1959). I fear that Dr Engelder has confused Gödel with Popper, and so it might be wiser if he could avoid any future philosophical commentaries until such time as he has studied the field in more detail.

I am less happy that he suspects me to be an 'agenda-driven' and 'advocacy-based' scientist. Here Dr Engelder has been quoted as saying that he would "*really like to occupy the middle ground in the industry v. anti-driller scrum*", seeing himself as being above the world of agendas or of advocacy. But presumably he is, like me, an 'advocate' of sound, evidence-based science. So I am not sure that the epithet carries any useful meaning. Nor do I think that sitting on the fence, as he claims to do, is necessarily justified. Surely decisions, and therefore sides, have to be taken eventually. I note here that he has boasted of bringing in "*at least $6 million in grants from industry and $8 million from government*" ([http://www.post-gazette.com/business/businessnews/2011/03/20/The-Marcellus-Boom-Origins-the-story-of-a-professor-a-gas-driller-and-Wall-Street/stories/201103200259](http://www.post-gazette.com/business/businessnews/2011/03/20/The-Marcellus-Boom-Origins-the-story-of-a-professor-a-gas-driller-and-Wall-Street/stories/201103200259)). These are impressive figures. He

is further quoted in the same piece as saying "*There is a symbiosis between academic research, and by that I mean big-time research of the type that Penn State* [his university] *does, and industry. Industry really does benefit from this. There is a reason that industry contributes very handsomely to the academic world*".

Dr Engelder has thus been so embedded in his symbiotic link to industry for the last thirty years that I think he fails to see that this must colour his thinking; and for him to claim, notwithstanding his professional dependence on industry, that he is on the fence on the subject of fracking is quite unjustified. Therefore I could equally well call him 'agenda-driven', and an 'advocate' of fracking, but such accusations or insinuations, as he has made about me, but in the opposite direction, are unhelpful and counter-productive.

I became involved in the fracking debate because I perceived that only one side of the science was being presented, and that side was pro-industry and in favour of fracking. Ill-informed science, which Dr Engelder rightly scorns, includes the report by the academic expert group convened under the auspices of the Royal Society and Royal Society of Engineering of 2012. I have explained in detail in my paper why this report falls short. I also agree with Dr Engelder that the anti-fracking camp has promoted many misconceptions and half-baked quasi-scientific notions, but I do not fall into that camp. Indeed, I consulted for the oil and gas industry intermittently between 2002 and 2011, and have no technical objections to conventional exploration. Furthermore, I have been approached to help oppose drilling applications in the UK, but I have declined these requests, because they concerned orthodox, conventional onshore exploration projects which are very unlikely to cause environmental harm. Lastly, on the question of earthquake triggering by the fracking process itself, I have always maintained, in talks for the general public as well as in writing, that this is a side issue of little impact. Triggering by disposal of waste water by injection is, of course, another matter.

To conclude this section of the discussion, if anyone can claim to be unbiased (but well-informed) in the scientific debate about fracking it is I. To have reached an evidence-based conclusion, as I have done, about the environmental risks of faults in the fracking context is not evidence of bias. Dr Engelder can hardly claim impartiality, or lack of bias, himself; his article in the geological industry literature entitled *Truth and Lies about Hydraulic Fracturing* (Engelder 2014) implies, to me, a degree of partisanship. In contrast to Dr Engelder, I have neither monetary nor reputational advantage to be gained; nor do I have an egotistical need to give public lectures or to be cross-examined in legal enquiries, as I have done; in fact, I rather wish that I could return to other more productive areas of earth science research.

**The evidence for contamination of water wells in Bradford County, PA**

Dr Engelder claims that I have completely misconstrued the results of Llewellyn et al. by summarising the study thus: " [It] *proves beyond reasonable doubt that contamination of drinking water was caused by passage of frack fluid and/or produced water **in part through the geology***". [my emphasis]. I fail to see how the latter part of this statement is inconsistent with the authors' summary, as follows:

"*The most likely explanation of the incident is that stray natural gas and drilling or HF compounds were **driven ?1–3 km along shallow to intermediate depth fractures** to the aquifer used as a potable water source.*" [my emphasis].

Llewellyn et al. conclude:

"*The data released here do not implicate upward flowing fluids along fractures from the target shale as the source of contaminants but rather implicate fluids flowing vertically along gas well boreholes and **through intersecting shallow to intermediate flow paths via bedrock fractures**. Flow along such pathways is likely when fluids are driven by high annular gas pressure or possibly by high pressures during HVHF injection.*" [my emphasis].

For the lay person, co-author Susan Brantley has explained the geological link thus:

"*The most reasonable explanation of our findings indicated that **a highly diluted chemical mixture used in shale gas wells traveled more than 2 kilometers across natural fractures in the Earth's rocky subsurface** and entered drinking water wells.*" [my emphasis].

So Dr Brantley's quotation of 'the rocky subsurface' as being the "*most reasonable explanation*" of the pathway clearly excludes surface flow or unconsolidated sediments.

My original Figure 10 was prepared from an anamorphic version of the cross-section in Llewellyn et al.'s figure S9, squashed horizontally and simplified to show one of the Welles series of wells, not five, and with a schematic vertical fracture zone added. It has a clear error in showing the well cutting the Marcellus shale vertically, and not landing horizontally in the shale, because the original diagram on which it is based has the same error. I had also included arrows indicating direction of flow of methane and contaminated water, but removed these before submission. They showed vertical flow up the wellbore, then transmission up-dip and to the south along bedding, and then vertical transmission up the schematic fracture. But I had not appreciated that Llewellyn et al. had ruled out vertical transmission from a source as deep as the Marcellus. So apart from implying that the fugitive gas and contaminated water came from the

Marcellus shale, **my diagram and text is basically correct in summarising Llewellyn et al.'s view that the pathway was through the shallow and intermediate depth geology.**

I withdraw my claim that Birdsell et al.'s paraphrase of Llewellyn et al.'s conclusions was wrong and misleading, and will instead add a brief summary of the importance of this new paper to my modelling review section, as well as adding a new box to my organogram. Birdsell et al. themselves review the previous modelling studies of fluid flow up faults, but it is a pity that they did not include Cai and Ofterdinger (2014) in their discussion.

My partial misunderstanding of Llewellyn et al. arose from my assumption that the fracking chemical 2-BE must have come from depth. The importance of Llewellyn et al.'s paper is that it demonstrated for the first time the passage of fluids and fugitive gas through the geology – albeit at shallow to intermediate depths (the uppermost 500 m) – and not just up faulty wellbores, as is now well-documented.

I am providing separately a detailed comment which re-interprets somewhat the conclusions of Llewellyn et al. My re-analysis shows that the preferred pathway of Llewellyn et al. - travel from wellbores at shallow to intermediate depths up geological fractures – is only part of the story. The notion of surface spills as being implicated in the homeowner well contamination is also discounted. My re-interpretation uses new data, including the detailed horizontal well plans and production history for the W1 and W3 pads which were not available to Llewellyn et al. Dr Engelder will be welcome to comment on this. I want my final paper to have resolved as many problems and arguments as possible, and I don't care how many rounds of comment and reply it takes.

I propose that my sections 5.3 on this subject be completely revised, and that a Supplement be added, since my re-interpretation requires several more figures. My original Figure 10 will be scrapped and replaced by a more detailed cross-section and map. I have no agenda here; my only interest is to establish something as near to the truth as we can get, based on imperfect and incomplete data.

**Critiques of Myers 2012 and other modelling papers**

Dr Engelder devotes much attention to the controversial Myers (2012) hydrogeological modelling paper, castigating me for having given it attention. He suggests that the best thing for a flawed paper, such as this one, is to ignore it. On the contrary, I discussed it, by no means uncritically (some 250 words on page 25),and by citing the other critics, because it may indeed

have an application in a generic sense for the case of vertical or near-vertical faults. It may therefore have some validity in the UK basins. Furthermore, it may well apply to the vertical fractures postulated by Llewellyn et al., as I show in my detailed comment mentioned above. If I am guilty of not ignoring the supposedly flawed Myers paper, then so are Vidic et al. (2013) and Birdsell et al. (2014), who gave it serious attention. The lesson is that we can learn even from flawed papers.

Dr Engelder criticises my organogram (Figure 9) showing the sequential development of papers related to fluid flow up faults. I should perhaps have explicitly stated in the caption that it was *in the context of fracking*, as I made clear in my text (p23 lines 14-15). We all know, as Dr Engelder states, that "*there are a basketful of geologists and geophysicists who have contributed to the peer-reviewed literature about leaking faults long before Northrup's web posting in 2010*". Indeed, there is, in addition, a whole oil service industry devoted to differentiating between faults as seals or as leaky pathways. But if he can cite any such papers referring to leaking faults *in the context of high-volume shale fracking*, then I would be the first to add them to the organogram. If they do exist, it is surprising that they have not been cited by the early papers and reports depicted on the diagram. I also used colour to highlight the papers that reported the results of quantitative modelling of flow up faults from shale. Again, I would be pleased to include any such studies that I may have missed, if Dr Engelder would be good enough to provide the references.

My organogram differentiates between peer-reviewed papers and non-peer reviewed literature. Dr Engelder refers to one study as being from a group "*with a known agenda*". Why does he exclude the industry-produced reports (shown in yellow) from this criticism? In my view these reports might be regarded with equal suspicion, their "*agenda*" being to promote their own possibly biased point of view and thereby profit financially. But my Section 5.2 on the fault modelling studies restricted itself to being an impartial review of work to date, whether peer-reviewed or not, and from whatever source. Dr Engelder seems to be unable to accept this as being a sensible way to review the literature.

Dr Engelder, included among his twelve self-citations, quotes his own work on imbibition. These include his comment on Warner et al. (2012a), but he omits to mention either the original paper or the reply to his comment by Warner et al. (2012b). But here is not the place to develop a discussion of the relative importance of imbibition and/or well suction for reducing the flow up faults or fractures. Birdsell et al.'s results show that well suction is the dominant process in reducing the mass flow to the aquifer, and that imbibition is relatively marginal. The

significance of Birdsell et al. is that they show, using realistic generic conditions, that mass flow reaches an aquifer quickly, but in very small quantities. One somewhat unrealistic value for an important parameter chosen by Birdsell et al. is their assumed 20-year timespan for production. It would be interesting if a more realistic value of around 8 years were used (http://www.marcellus-shale.us/Marcellus-production.htm), and to model what happens once production ceases, either temporarily, perhaps due to an interruption in the supply chain, or permanently once production is deemed to be uneconomic.

**2-BE (2-n-Butoxyethanol) as an indicator of frack fluid**

Dr Engelder almost had me fooled for a moment with his homely discussion, starting with drilling his own water well, segueing into air (percussion) drilling for the 13 inch surface casing of a typical gas well, all with the aim of insinuating that AirFoam (which contains 2-BE) *might* have been used in the drilling of the Welles series of wells. Never mind that his volume calculation is way out – the actual volume of soil and bedrock that are "*disturbed*" is about 30 $m^3$ (I think he forgot the $\pi$ factor). For earthworks, this is not exactly a large figure; for example, it's about one-fifth of the water volume of my domestic swimming pool. He then goes on to review the apparently low toxicity of the chemical – an irrelevant diversion, because its identification by Llewellyn et al. was in pursuit of sourcing the household well contamination, and had no bearing on the toxicity or otherwise of the drinking water. He considers "*the possibility that the source of the very low amounts of 2-BE in local groundwater are local septic fields into which household products with 2-BE may have been flushed for years and years.*" This attempt at diverting attention from the Paradise Road homeowner well contamination omits mention either of the timing of the whole episode in relation to Chesapeake's gas drilling activities, or of the uncontaminated homeowner wells B1-3 in the locality.

No, the fact is that 2-BE was a documented component of the frack fluid in the Welles 2 to 5 pads. We do not know whether it was also used at Welles 1, but "*it is reasonable that the same nonemulsifier agent (which contained 2-BE) was likely used*" (Llewellyn et al. 2015). The composition of the frack fluid used for W 1-3H and W 1-5H has not been published on www.fracfocus.org, the voluntary industry website. Without sounding too paranoic or suspicious, it is reasonable of me to ask why the data for the Welles 1 pad has not been disclosed.

For Dr Engelder to imply that the source of the 2-BE is the vertical well air drilling, or

homeowner septic fields, and not the frack fluid, is a classic example of disinformation that he ascribes to others. Septic systems were discounted by LEA in a supplementary 'Frequently Asked Questions' paper (Llewellyn et al. 2015). Lastly, even if it were true that the air drilling used AirFoam, *in addition to* 2-BE being likely present in the frack fluid, I have shown in my re-interpretation of the case history that the explanations of (i) surface spills, leaks or vertical air drilling as a source for the 2-BE, and (ii) shallow to intermediate wellbore leaks as the source for the fugitive methane, are both unlikely.

**Additional references**

Engelder, T. 2014. Truth and lies about hydraulic fracturing. *AAPG Explorer*. Available from: http://www.aapg.org/publications/news/explorer/details/articleid/12416/truth-and-lies-about-hydraulic-fracturing.

Llewellyn et al. 2015. Frequently asked questions about the study "Evaluating a groundwater supply contamination incident attributed to Marcellus Shale gas development" by Llewellyn et al. http://www.appalachiaconsulting.com/home/whats_new/pnasarticlefaqs

Vidic, R.D., S. L. Brantley, J. M. Vandenbossche, D. Yoxtheimer, and J. D. Abad. 2013. Impact of shale gas development on regional water quality. *Science* 340, 1235009 (2013). DOI: 10.1126/science.1235009.

Warner N.R, et al. 2012a. Geochemical evidence for possible natural migration of Marcellus Formation brine to shallow aquifers in Pennsylvania. *Proc Natl Acad Sci USA*

109(30):11961–11966.

Warner N.R, et al. 2012b. Reply to Engelder: Potential for fluid migration from the Marcellus Formation remains possible. *Proc Natl Acad Sci USA* Early Edition www.pnas.org/cgi/doi/10.1073/pnas.1217974110 PNAS.

---

## Author Comment (AC3)

**Smythe (doi:10.5194/se-2015-134) Supplement S1**

**Water well contamination case history: Bradford County, Pennsylvania**

**Introduction**

I propose that this section be incorporated, together with revision of sections 5.3.1 and 5.3.2, as a supplement to my paper, since it goes into considerably more detail than I had provided in my discussion paper. The discussion section below will form the basis for the revised text in the main paper. I thank Dr Engelder for prompting me to think more clearly about this problem.

Let us first define terms in the context of the locality, relative to local ground level:

- *Surface or near-surface* :

A leak out of a pit at W1 occurred on 7 August 2009, and on 2 September the same two wells (W1-3H and W1-5H) were cited by PADEP for discharge of contaminated fluids to ground. Well W1-3H was also cited on 29 September 2011 for *"failure to control residual waste to prevent water pollution"*, but this latter event postdates, and therefore cannot be related to, the homeowner well contamination. The 2009 pit leak could either be due to a spill on the W1 pad, or to a leak from the main impoundment pond. Both these scenarios are considered next.

---

## Author Comment (AC4)

**Response to Daniel Birdsell and co-authors**

Mr Birdsell has commented under several headings, as follows:

- 1 Misunderstandings in the literature
  - 1.1 Fisher and Warpinski 2012
  - 1.2 Llewellyn et al. 2015
- 2 Numerical modelling

I thank him (and his colleagues, on behalf of whom I presume he is writing, since he uses the first person plural) for his comments, and respond below to his sections 1.1 and 2 above. I have considerably revised my analysis of Llewellyn et al. (2015), submitted separately, and trust that this revision will satisfy his concerns cited under 1.2 above. Concerning Llewellyn et al., he might also wish to read my response to Dr Engelder, again, submitted separately.

**Misunderstanding of Fisher and Warpinski**

Fisher and Warpinski (2012) wrote:

"Faults have been suggested as mechanisms for enhancing fracture growth, but this ignores the basic understanding of faults **in hydrocarbon reservoirs**. If there is an open path to the near surface through an existing fault, throughout geologic time, all of the hydrocarbons would have escaped and there would be no reason for exploiting the resource." [my emphasis].

"While faults can offer somewhat better conductive paths, it is not likely that they are conductive over sizeable fractions of the depth because any **oil or gas in the reservoir** would have escaped through such conduits and there would not be any hydrocarbon exploitation success in that area." [my emphasis].

I criticised these statements, writing: "... *if faults were conduits they would have leaked all the gas away by now. This is clearly false; the whole point of fracking is to release gas which is trapped and therefore unable to migrate.*"

Mr Birdsell has commented on my statement above as follows:

"If highly permeable faults did exist, the hydrocarbons would have leaked out of the

reservoir during the millions of years since the hydrocarbons were generated. Smythe has misunderstood the qualitative argument that the presence of hydrocarbons indicates that either there are no faults near **the shale reservoir**, or that if there are faults, they do not conduct fluids at a high rate." [my emphasis].

The confusion has arisen because Fisher and Warpinski describe (correctly) the behaviour of faults in hydrocarbon *reservoirs*, as I have emphasised in the two quotations. Conventionally, a reservoir rock will be a sandstone or limestone, but never a shale. A shale, conventionally, is a source rock or a seal, unless it is (unconventionally) fracked, at which point it becomes both the source and the reservoir. Hydrocarbons are generated within a source rock under appropriate P-T conditions, and will migrate, albeit slowly (i.e. over geological time) upwards into potential reservoirs. Hydrocarbon generation is usually a dynamic process, with ongoing generation and migration, if the 'kitchen' (the source rock at the right depth) is active. The process can even be visualised now with high-quality (offshore) 3D seismic (Aminzadeh et al. 2013). What Fisher and Warpinski say does not apply to very low permeability rocks like shale, because they are not *reservoirs* until they have been fracked.

Take the Marcellus as an example. If a hypothetical permeable fault cuts the Marcellus, only the rock in the immediate vicinity will be drained of hydrocarbons, not the entire layer. Proof of this is the fact that hydrocarbons remain trapped today in the Marcellus, even though the 'cooking' took place some 350 million years ago when it was buried to 4-5 km. If the Marcellus had sufficient permeability for faults to drain it, then all the hydrocarbons would have long ago migrated upwards - a distance of a mere 50 to 100 m - into the overlying Mahantango, even in the absence of faults.

My understanding of shale source rocks is that the gas is mostly trapped by being adsorbed onto microscopic surfaces, and only a small proportion is present as free gas in pore space. Fracking physically transforms this source/seal rock into a reservoir, desorbing the gas, and only then can permeable faults, if present, start to drain the shale.

I largely agree with Mr Birdsell's later discussion about rates of flow and the likelihood of high gas concentration remaining near a permeable fault, but I disagree with his interpretation of Fisher and Warpinski's "general concept ... [that] *if there is a high concentration of gas in a source rock, there probably is not a highly permeable fault nearby.*" That is not what they said; they twice mentioned faulting in the context of *reservoirs* (cf. the two quotations from Fisher and Warpinski cited above). Therefore I stand by my claim that Fisher and Warpinski

miscontrue the importance of faults in having the potential to drain an unfracked shale.

**Numerical modelling**

I apologise to Mr Birdsell for omitting a proper discussion of his 2015 paper in my review of the development of numerical modelling of flow up faults. I had downloaded it on 29 November 2015, and referred to it in the context of Llewellyn et al. (2015), but had not then given the attention it deserves before submitting my initial discussion paper a month later. I shall add his paper to my organogram and discuss its important contribution in a new section within section 5.2.

**References**

Aminzadeh, F., Berge, T. B., and Connolly, D. L., 2013. Hydrocarbon seepage: from source to surface, Geophysical Developments Series no. 16, Society of Exploration Geophysicists and American Association of Petroleum Geologists, Tulsa, Oklahoma, 244 pp.

Birdsell, D. T., Rajaram, H., Dempsey, D. and Viswanathan, H. S., 2015 Hydraulic fracturing fluid migration in the subsurface: A review and expanded modeling results, Water Resources Research, 51(9), 7159–7188.

Fisher, K. and Warpinski, N., 2012. Hydraulic-fracture-height growth: real data. Society of Petroleum Engineers Annual Conference Paper SPE 145949, Denver 2011. SPE Production & Operations, February 2012, pp 8-19.

Llewellyn, G.T., Dorman, F., Westland, J.L. Yoxtheimer, D., Grieve, P. Sowers, T. Humston-Fulmer, E. and Brantley, S., 2015. Evaluating a groundwater supply contamination incident attributed to Marcellus Shale gas development. Proc. Natl. Acad. Sci. PNAS Early Edition, www.pnas.org/cgi/doi/10.1073/pnas.1420279112, 2015.

---

## Author Comment (AC8)

**Reply to Professor Paul Younger (SC6)**

I am responding to Professor Younger's comments under headings which follow his own subject headings as far as possible.

**My erroneous assumption no. 1: inherent risk of groundwater resource contamination via faulting during or after unconventional resource development**

Professor Younger cites several hydrogeology textbooks to deny that faults 'inherently' act as pathways. Perhaps we are not talking the same language; to me, 'inherent' means built-in, innate, or intrinsic, qualifying adjectivally the noun 'risk', or chance, probability. The phrase 'inherent risk' does not imply that faults are necessarily permeable to flow, and I am of course aware that any particular fault may behave differently over different segments of its track, and over different geological periods. But there is a built-in risk, which needs to be assessed and, if possible, quantified.

So if the 'inherent risk' of faults acting as pathways in shale development is very low, as Professor Younger seems to imply, then why have so many quantitative modelling papers been published about the very problem in the last few years? Why is the English summary of the extensive 2012 German study called 'Hydrofracking Risk Assessment'? That document concludes, regarding groundwater:

"Hydrofracking can entail considerable environmental risk, particularly when it comes to water resource conservation, which we strongly feel absolutely must take precedence over energy production."

I am well aware that the French work I cited concerns – in part - karstified limestones, an extreme kind of rock formation, hydrogeologically speaking. However, it is demonstrated there that deep pathways down to greater than 2 km depth involving faults do exist, and limestone plays little or no part in the flow systems at depths greater than 1 km. The studies show that upward fault pathways exist through Lias shales, which were the target of a Total exploration licence (since annuled) for fracking.

Similar geological terrain was licensed for fracking by DECC in Somerset, where there is a proven deep flow system through the Carboniferous Limestone, recharged in the Mendips, flowing north at depth, and coming up to the surface along faults at Bath. The southern French and the Somerset examples, which both involve thermal springs known since at least Roman occupation times, also both show that unconventional exploration companies will simply disregard any groundwater risk if

they can get away with it. At least the French government learned fast, unlike its UK counterpart, and decided in 2011 to cancel all the licences it had previously awarded for unconventional exploration.

In conclusion, there is an 'inherent risk' in unconventional resource exploitation.

**My erroneous assumption no. 2: hydraulic gradient favouring upflow**

I refer Professor Younger to my review of the literature that has appeared since 2010 on the quantitative modelling of groundwater flow from fracked shale up fault zones. I shall be adding a brief review of Birdsell et al. (2015). All these studies concur that upward flow is possible; the questions which remain the subject of debate are the precise mechanisms, and the timescales. In conclusion, upflow can happen, driven by several different forces.

**My erroneous assumption no. 3: saline springs as an example of dilution**

Professor Younger makes an analogy with saline springs to show that even if contaminating fluids did reach shallow groundwater resources, the contaminants would be "*diluted beyond detectability*". Even if such an analogy were appropriate, it evidently excludes gas (especially methane) migration.

Such an argument is reminiscent of the days when it was thought acceptable for nuclear waste to be dumped in the oceans, justified by the so-called 'dilute and disperse' principle. It is invalid, not least because one of the modelling studies I cited (Gassiatt et al. 2013) mentions, *en passant*, that contaminated fluid reaches the near-surface *via* the specified pathway at 90% of its original concentration. It would be complacent of anyone to assume that such fluid would then somehow get "*diluted beyond detectability*" as Professor Younger hopes.

In conclusion, it seems to me that Professor Younger is taking a general stance, regarding the socalled erroneous assumptions that I have made, that is rather out of the mainstream thinking on the hydrogeology of unconventional resource exploitation. Since arriving at Glasgow nearly four years ago he will have had the resources of a research group to set up some numerical modelling studies of his own, if he disagrees with certain aspects of the mainstream.

**Royal Society and Royal Academy of Engineering report of 2012**

I took the report to task for failing to consider the fault problem properly. Professor Younger initially (in 2014) accused me of simply not reading it thoroughly; however, in trying to defend this aspect of the report, he now resorts to generalities concerning the eminence of the two societies.

I shall rephrase my summary statement (section 6.2.4) referring to the report having a "perceptible"

*pro-industry bias*"; this was alluding only to the uncritical comments the committee made about the upward growth of fracks in the Halliburton paper (Fisher and Warpinski 2012), and on re-reading the report I do not find any other instance of there having been such a bias.

**Potable groundwater below the Mercia Mudstone Group (MMG)**

My doubts about the Environment Agency (EA)'s claim (*inter alia*) that groundwater in the Sherwood Sandstone Group (SSG) at 300 m depth below the MMG in the Fylde is saline, and therefore not of concern, are dismissed by Professor Younger; however he does not offer a detailed rebuttal, because he apparently does not have time to deal with my "*unsubstantiated opinions*".

Professor Younger states, in particular:

"Smythe's claims about the possibility that fresh groundwater occurs in the Sherwood Sandstones beneath saline water in the Mercia Mudstones is **at odds with all known sites in the UK where this setting has been monitored** (e.g. in many English coalfields)." [my emphasis].

I hesitate to take issue with this strong claim by one of the UK's leading hydrogeologists. So let us first be clear what is being said:

- The "*setting*" is my claim that there could be fresh groundwater in the SSG below saline water in the MMG (in the Fylde), and
- Such a setting does not exist anywhere in the UK.

There are three other basins, apart from West Lancashire Basin within which the Fylde is situated, which have halite within the MMG. These are the Avon/Somerset, Worcester/Gloucester, and Cheshire basins (Hobbs et al. 2002). So Professor Younger is claiming that because the setting occurs in three of the four halite-bearing basins it must necessarily be true of the fourth, the Fylde. Firstly, this is a *non sequitur*, and secondly, it is not even necessarily correct in the three other basins, as I now demonstrate.

I have looked briefly, but not systematically, at the Cheshire Basin. The regional flow through the SSG is to the NW, towards the Irish Sea, and "*flow probably tends to follow peripheral routes, around the deeper area, where the permeability is better and is enhanced by fractures*" (Downing et al. 1998).

A groundwater baseline study of the sandstones of west Cheshire by Griffiths et al. (2002) concerns mainly the outcrop of the SSG of west Cheshire and the Wirral, west of the central part of the

Cheshire Basin where the MMG crops out. Griffiths et al. cite one component of the present day groundwater flow regime, as follows:

"At the centre of the basin where the SSG is covered by the MMG. Density variations in the groundwater as a result of halite dissolution and mixing of freshwater at the margins influences the groundwater flow. The division of the flow field into a number of mixing cells results in **quite large salinity variations** across the basin." [my emphasis].

Although concentrating on the western SSG outcrop, their study does include seven locations within the central outcrop of the MMG (Griffiths et al., fig. 6.7). Chloride levels within four of these boreholes is in the range 30-50 mg/l, two are in the range 50-105 mg/l, and the seventh is of the order of 200 mg/l. So they all appear, in principle, to provide potable water. Details of some of these boreholes can be found on the BGS Borehole Viewer. The two adjacent boreholes in the centre of their map (with Cl c. 30-50 mg/l) are on the outcrop of the Tarporley Siltstone (MMG), and about 1-2 km east and SE of the Helsby Sandstone Formation (SSG), formerly the Lower Keuper Sandstone. The top of the SSG is at 68 m in the borehole. The principal borehole of this pair is named Eaton Crewe Waterworks, from which it is evident the use to which the resource is (or was) put.

There are many other boreholes on the outcrop of the MMG, not necessarily penetrating to the SSG, but which are licensed for agricultural abstraction, including irrigation and golf course watering. It is unlikely that these are producing saline water. In conclusion, Professor Younger's generalisation about the 'setting' of MMG with halite over SSG does not stand up to scrutiny.

Now let us examine the Fylde evidence, without accepting Professor Younger's evidently overgeneral conclusion. For the benefit of the Editors and any other readers, let me first summarise and expand upon what I wrote about the Fylde. Firstly, I noted that the only evidence that the EA seems to consider is the hypersalinity in the Kirkham geothermal test borehole, which penetrated the SSG at 366 m. I pointed out that this evidence is invalid because two of the three hypersaline samples were taken from levels within the MMG, where the observed *hyper*salinity (and not simply salinity) can be explained by perched relict halites known to exist within the MMG. The groundwater within the SSG was never sampled. Surely this is a fundamental point which Professor Younger should have taken on board.

I then alluded to two other boreholes which penetrated to (and presumably abstracted from) the SSG, writing "[they] *suggest that potable water was formerly exploited within the SSG*". For the record, these two boreholes are Rowe's Model Dairies at Inskip, and Phoenix Mill at Kirkham. They

Professor David Smythe

are part of a group which I examined, comprising some 39 relevant wells west of the Woodsfold Fault, of depth greater than 30 m, which are available on the BGS borehole mapper website. I think it is a reasonable conclusion to draw that it is unlikely that hypersaline water was used either for the cheesemaking at the dairy or for cotton spinning at the mill. About five of these borehole records are confidential, and/or there is no information. In addition, I studied the water composition records of 56 boreholes, which I obtained from the EA.

In conclusion, my concern that the EA has written off a past and future potential groundwater resource in the Fylde is justified. Before any unconventional exploitation begins it would be prudent for the EA and/or the BGS to sample the water at SSG levels.

**My criticism of the EA**

I carried out my study of the Fylde before coming to my conclusion about the EA's potential failure to protect groundwater resources below the Fylde. I am critical of the organisation, not of any individual employees, as Professor Younger implies.

Since I submitted my discussion paper more evidence has emerged which casts the EA's views on the risk to Fylde groundwater resources in an even worse light than I had viewed it with previously. I append below as an Appendix part of a comment I submitted to the Local Planning Inquiry in March 2016. In brief, the EA never responded to my comments of April 2015 on the hypersalinity readings in the Kirkham well. Moreover, the agency tried to justify its *laissez-faire* stance by quoting a then-confidential study it had made of the SSG aquifer of NW Lancashire. It turns out that this study is twenty years old, was carried out for the EA by the BGS, and that a crucial part of the study (the basemap) has been lost. However, it highlights the fact that the location of the important Woodsfold Fault is uncertain. In conclusion, my criticism of the EA is based upon sound evidence.

**UK regulation**

Under this heading Professor Younger once again quotes the joint committee report (Mair et al. 2012) discussed above, and also cites the Scottish independent report on unconventional gas (Masters et al. 2014). I am sorry to say that I count three of the latter report's authors among my former colleagues. This report also barely mentioned fault pathways, although it did comment:

"6.78 Other pathways for leakage may also exist, such as through faulting, mine workings or other boreholes which may be some distance from the wellhead. However, this requires artesian groundwater pressures."

I have discussed the question of UK regulation further in my replies to Dr Verdon and Dr Westaway.

Professor David Smythe

**Comments and conclusions on Professor Younger's comment**

Overall, I find Professor Younger's comments to be somewhat dogmatic in tone, and although I have tried to consider them seriously – for example, spending a day investigating the Cheshire Basin - I find little reason as a result of his comments to alter and improve my discussion paper, other than in minor ways.

**References**

Downing, R.A., Edmunds, W.M. and Gale, I.N. 1998. Regional groundwater flow in sedimentary basins in the U.K. *From* Goff, J. C. & Williams, B.P. J. (eds), 1987, *Fluid Flow in Sedimentary Basins and Aquifers*, Geological Society Special Publication No. 34, pp. 105-25.

Ewen, C., Borchardt, D., Richter, S. and Hammerbacher, R. 2012. *Hydrofracking Risk Assessment Executive Summary* [English summary version of Borchardt et al., www.dialog-erdgasundfrac.de]

Gassiat, C., Gleeson, T., Lefebvre, R., and McKenzie, J. 2013. Hydraulic fracturing in faulted sedimentary basins: Numerical simulation of potential long term contamination of shallow aquifers, Water Resour. Res., 49(12), 8310-8327, doi:10.1002/2013WR014287.

Griffiths, K. J., Shand. P. and Ingram. J. 2002. Baseline Report Senes 2. The Permo-Triassic Sandstones of west Cheshire and the Wirral. British Geological Survey Commissioned Report No. CR/02/109N.

Hobbs, P.R. N. et al. 2002. Engineering geology of British rocks and soils — Mudstones of the Mercia Mudstone Group. *British Geological Survey Research Report*, RR/01/02. 106 pp.

Mair, R., Bickle, M., Goodman, D., Koppelman, B., Roberts, J., Selley, R., Shipton, Z., Thomas, H., Walker, A., Woods, E., and Younger, P.L. 2012. *Shale gas extraction in the UK: a review of hydraulic fracturing*. Royal Society and Royal Academy of Engineering, London. 76pp.

Masters, C., Shipton, Z., Gatliff, R., Haszeldine, R.S., Sorbie, K., Stuart, F., Waldron, S., Younger, P.L., and Curran, J. 2014. *Independent Expert Scientific Panel – Report on Unconventional Oil and Gas*. Scottish Government, Edinburgh. 102pp.

**Appendix**

Extract from appendix to *Comments on faulting: Appeal against refusal of planning applications by Cuadrilla Bowland Limited to drill at Preston New Road and Roseacre Wood (LCC/2014/0096, 0101)*, submitted to the Local Planning Inquiry on 10 March 2016.

**3. Approval by the Environment Agency**

On 19 May 2015 the EA responded to LCC regarding my comments of 20 April incorporating new information:

"We are satisfied that our technical assessment remains correct and that the consultation response from Professor Smythe does not alter our assessment.

•••

Points 1 to 4 relate to the geological complexity of the area being greater than that shown on published geological maps. The Environment Agency agrees with this statement. This is the reason that additional work was commissioned by us during the water resources modelling work to improve the understanding particularly in the southern area of the Fylde aquifer where modelling difficulties were encountered. This work was subsequently supplemented by the sinking of two exploratory observation boreholes which were also used in conjunction with a large number of seismic lines, hydrogeological responses and the model output to reinterpret the understanding of the geology. This work resulted in a revised understanding of the Fylde sandstone aquifer and the groundwater flow regime. As stated in the permit decision documents the water resources modelling outcomes have informed our decisions at the two sites."

In my view this response misleadingly implies that new geological work had been commissioned by the EA in the recent past, whereas it transpires that the work had been undertaken twenty years ago. The two new observation boreholes are in the Preston area, and have no direct bearing on the modelling of water flow under the Fylde.

The report in question was confidential, therefore a relevant question to ask is, did it have any bearing on the interpretation of Fylde geology in general, and of the Woodsfold Fault in particular? I asked the EA for a copy of the report on 21 June 2015. After initially refusing my request, the EA released a slightly redacted version of the report on 25 August 2015 (NEW EVIDENCE). The report had been undertaken by the British Geological Survey (BGS) for the NW Rivers Authority in 1996, and parts of it were redacted because the BGS claimed that some of it was 'Commercial in

Professor David Smythe

Confidence'. This referred to commercial seismic data reproductions contained therein.

I asked the BGS on 15 December to supply the redacted page and the map missing from the report, on the ground that the commercial data had long since been released under DECC rules. The BGS duly supplied a copy of the report, including the page previously redacted, but wrote of the map:

"Unfortunately we no longer appear to hold a copy of figure 8 and this is after exhaustive searches made at BGS. The statement on page 13 of the report, 'This is a confidential diagram held at the British Geological Survey, Keyworth, Nottingham' was contained in the original report that was delivered to the National Rivers Authority (NRA) and appears in our archived copy of the report. The actual diagram itself was shown to the NRA at the time the report was delivered but was never included in the report. We have spoken to the lead author of the report who has been unable to find the original diagram and tells us that we no longer hold it."

**So the current position is that an essential part of the geological remapping work referred to by the EA has been lost.** Fortunately the colour geological map resulting from the BGS remapping survives, although the underlying data cannot now readily be identified due to the missing map. This revised map shows that the Woodsfold Fault at outcrop (at the surface of the earth) is now placed 1 km west of its position found on the BGS Garstang 1:50,000 solid geology sheet, dating from 1990.

The BGS digital database still uses the 1990 epoch geology; no attempt has been made to update the digital database to incorporate the 1996 work. However, remapping of the faults in the region has been undertaken once again by the BGS, this time as part of the Bowland Shale study (Andrews 2013). This work results in yet another new position for the Woodsfold and other faults. The various versions of the fault outcrops are shown in Figure 1. It is clear from the inconsistencies in this map that the understanding of the major fault structures and layering (the 'architecture' of the Bowland Basin) is still inadequate. The area of the 3D seismic survey commissioned by the Appellant (not shown in Figure 1) is approximately a portrait-format rectangle aligned with the grid, just enclosing the Preston New Road site and Preese Hall-1 well to the west, just enclosing the Thistleton and Mid Elswick Fault label boxes to the north and south, but not extending quite as far east as the Roseacre Wood site.

Figure 1. Various interpretations of outcrop of major faults in the Fylde. Red with ticks – BGS solid geology maps and digital database. Green with ticks – BGS Bowland Shale study, faults at depth extrapolated to surface. Red with semicircles – interpreted by David Smythe from gravity data. Blue with ticks – BGS 1996 remapping for EA. Grid is at 1 km interval.

So the existence of the 3D survey cannot be used as a reason for justifying the Appellant's assertion that it understands the geology. The Appellant has never published the raw 3D data, so that the drilling applications depend on line-drawing interpretations of the Applicant's interpretation of the dataset. This is unacceptable. The one very small sample of the 3D data published by the Applicant (Clarke et al. 2014) has merely served to incite more argument and conflicting interpretations about faulting in the Preese Hall-1 well area (e.g. Smythe 2016, Westaway 2016).

**Appendix references**

Andrews, I. J., 2013. *The Carboniferous Bowland Shale gas study: geology and resource estimation*, British Geological Survey for Department of Energy and Climate Change, London, UK, https://www.gov.uk/government/publications/bowland-shale-gas-study

British Geological Survey 1996. *Geophysical remapping of the Fylde aquifer. BGS Technical report* WA/96/24C. Confidential report for National Rivers Authority NW Region. Environment Agency 2015. Email from Sarah Scott to Clare Phillips of LCC. 19 May 2015, 10:26.

Professor David Smythe Reply to Professor Paul Younger

Smythe, D.K., 2016. Hydraulic fracturing in thick shale basins: problems in identifying faults in the Bowland and Weald basins, UK. *Solid Earth Discussions*; doi: 10.5194/se-2015-134, 45 pp.

Westaway R., 2016. The importance of characterizing uncertainty in controversial geoscience applications: induced seismicity associated with hydraulic fracturing for shale gas in northwest England. *Proceedings of the Geologists' Association*. Doi: 10.1016/j.pgeola.2015.11.011, 17 pp.